# WEAVER: Shrinking the Generation-Verification Gap with Weak Verifiers

Jon Saad-Falcon[1][*] E. Kelly Buchanan[1][*] Mayee F. Chen[1][*]

Tzu-Heng Huang[3] Brendan McLaughlin[1] Tanvir Bhathal[1]

Shang Zhu[2] Ben Athiwaratkun[2] Frederic Sala[3]

Scott Linderman[1] Azalia Mirhoseini[1] Christopher Ré[1]

[1]Stanford University    [2]Together AI    [3]University of Wisconsin–Madison
[*]Equal contribution
{jonsaadfalcon, kelly.buchanan, mfchen}@stanford.edu

## Abstract

Verifiers can improve language model (LM) capabilities by providing feedback or selecting the best response from a pool of generated candidates. Currently, high-quality verifiers are either unscalable (e.g., humans) or limited in utility (e.g., tools like Lean for formal proofs). While LM judges and reward models have become broadly useful as general-purpose verifiers, a significant performance gap remains between them and oracle verifiers. To help close this gap, we introduce WEAVER, a framework for designing a strong verifier by combining multiple weak, imperfect verifiers. First we find that weighted ensembles of verifiers, which typically require learning from labeled data, significantly outperform unweighted combinations due to differences in the verifiers. To reduce the dependency on labeled data, WEAVER leverages weak supervision to estimate each verifier's accuracy and combines their outputs into a unified score that better reflects true response quality. However, directly applying weak supervision algorithms poses several challenges, including inconsistent verifier output formats and handling low-quality verifiers. WEAVER addresses these challenges by using dataset statistics to normalize outputs and filter specific verifiers. We study the effectiveness of WEAVER in repeated sampling settings, where a model generates multiple candidate responses at test time and a verifier is used to select the correct one. Our evaluations demonstrate that WEAVER significantly improves the $pass@1$ performance across several reasoning and math tasks, achieving o3-mini-level accuracy with Llama 3.3 70B Instruct (a much cheaper non-reasoning model) as the generator, and an ensemble of smaller judge and reward models as the verifiers (86.2% average). This gain mirrors the jump achieved between GPT-4o and o3-mini (69.0% vs. 86.7%), which required extensive finetuning and post-training interventions. To make WEAVER more efficient, we train a compact 400M cross-encoder using WEAVER's combined output scores. This distilled model retains 98.7% of WEAVER's full accuracy while reducing verification compute by up to 99.97%.

## 1 Introduction

A core challenge in deploying language models (LMs) is *verification*: determining the quality or correctness of a model's response. This problem arises across various components of the LM pipeline,

39th Conference on Neural Information Processing Systems (NeurIPS 2025).

including dataset curation, model alignment, and informed inference-time decision-making. A *verifier* is a function that scores responses. When combined with repeated sampling—generating multiple candidate responses from a LM—a strong verifier can be used to select a correct candidate response, significantly enhancing model capability on tasks such as math, code, and reasoning [5, 55, 70]. However, without a perfect verifier, a *generation-verification gap* emerges [72]: a LM can generate a correct response, but we fail to identify it.

In some settings, we have access to *oracle verifiers* that can perfectly identify correct responses. A prominent example is Lean, a formal theorem prover that can be used for problems such MiniF2F [94]. However, this is often a limited setup, as not all mathematical proofs can be processed by Lean. In contrast, more generally applicable verifiers such as LMs prompted as judges [13] and reward models [39, 45, 68] can be broadly applied to tasks like scientific reasoning, coding, and instruction-following [32, 40, 62]. However, these *weak verifiers* produce noisy, inconsistent scores, often exhibit poor calibration, and suffer from high false positive rates [73]. We ask: *to what extent can we leverage weak verifiers to improve accuracy in the repeated sampling regime?*

We explore *scaling verification*, specifically how to combine *multiple weak verifiers* to improve response selection for repeated sampling. As new pre-trained models become available, the pool of weak verifiers continues to expand and offer diverse, complementary sources of signal that could improve response selection if they can be aggregated effectively. Recent work has explored scaling verification through techniques such as self-verification or averaging LM judge scores [9, 41, 93]. However, it remains unclear how to effectively combine multiple weak verifiers. We observe three key challenges towards ensembling weak verifiers:

1. **Naively aggregating weak verifiers is insufficient for reliable verification.** Weak verifiers such as LM-based judges or reward models produce noisy, biased, and poorly calibrated scores, leading to inconsistent performance. [13, 39, 73]. While using a naive unweighted average of verifier scores is straightforward, it implicitly assumes uniform verifier quality, causing low-quality verifiers to dominate and degrade the overall accuracy [21, 76, 88]. Moreover, while previous work has hypothesized that more sophisticated weighted ensembles should perform better, this claim has not been studied [41].

2. **Effective ensembling with limited labeled data is challenging.** More sophisticated ensembling techniques typically learn verifier weights from labeled data, but such data is expensive and difficult to obtain. *Weak Supervision* (WS), a family of statistical techniques developed for data labeling, offers a potential solution through algorithms that aggregate multiple weak signals—such as crowd-worker annotations and expert-defined heuristics—while only requiring a small amount of labeled data [23, 57, 58]. In traditional WS, practitioners can design and shape each weak signal to ensure sufficient quality (i.e., iteratively tweaking program-based heuristics), and guarantees of WS hinge on a baseline level of quality. Our weak signals, however, are fixed pre-trained language model verifiers, which have wildly varying accuracy—especially when applied to out-of-distribution tasks—and can emit incompatible outputs (logits, binary scores, Likert scores) [39] that we cannot easily tweak. Due to these conditions, WS algorithms may not perform well when directly applied to verification.

3. **Verification is expensive to deploy at inference.** Verification can dominate inference-time costs [45, 68], since each verifier must process both the problem and its candidate response(s) [42], often evaluating intermediate steps [42] and multiple solution paths [70]. In fact, achieving modest gains over unverified generation can require $10\times$ to $128\times$ the inference compute per query [9, 41, 69, 93].

In this work, we introduce WEAVER, a framework for aggregating weak verifiers without supervised finetuning on ground truth labels (Figure 1). First, we demonstrate that if we have access to a large corpus of labeled training data (e.g., 50,000 query-response pairs), we can learn weighted ensembles that can outperform naive averaging by up to 11.2% points. This is because of wide variability in verifier accuracy. However, in many real-world scenarios, we do not have access to such quantities of labeled data. Second, to reduce the dependency on labeled data, we adapt Weak Supervision to the verification setting by addressing challenges around inconsistent outputs and low-accuracy verifiers. WEAVER filters out uninformative verifiers, normalizes verifier scores, and builds a latent variable model over these scores and the unknown true labels to estimate the verifier accuracies to be used as weights [26, 58].

Empirically, WEAVER improves over unweighted averaging of verifier scores by 17.1% and repeated sampling with majority voting by 13.5% (Table 1). WEAVER allows us to improve over $Pass@1$ by 17.9% for 8B models and 14.5% for 70B models across reasoning and mathematics tasks (Tables 1 and 18). *This mirrors the performance jump from GPT-4o to o3-mini* (73.9% vs. 88.2%)—but without

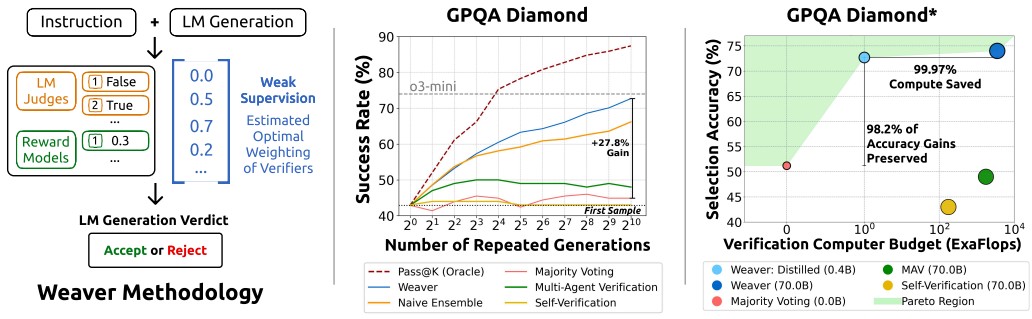

Figure 1: **WEAVER Framework**: We propose WEAVER, a framework combining multiple weak verifiers to effectively scale repeated sampling without parameter finetuning on ground truth labels **(left)**. WEAVER significantly outperforms majority voting and shrinks a model's *generation-verification gap* by 14.5%, on average, for GPQA Diamond and other datasets (Table 1) **(middle)**. By distilling WEAVER from an ensemble of 70B verifiers to a single 400M cross-encoder, we can preserve 98.2% of the accuracy gains of WEAVER while reducing inference compute cost by 99.97% **(right)**.

parameter tuning or post-training steps. We also study how WEAVER scales along different axes of test-time compute: generation, verifiers, model size, and inference budget (Section 5.2). We find that even as we increase the number of generations, many standard verification baselines (e.g. majority voting) quickly plateau (Figure 3). Naive ensembling saturates more slowly, but its gains are limited by high compute costs, and sensitivity to the choice and number of verifiers. We show that WEAVER outperforms alternative verification approaches by 13.5% while improving over $Pass@1$ by 18.3% as we scale from $2^1$ to $2^{10}$ generations (Figure 3).

Finally, as an extension of WEAVER, we train a 400M parameter cross-encoder verifier using WEAVER's selected responses to mitigate the compute costs of calling multiple weak verifiers on each response. We demonstrate that using distilled WEAVER cross-encoder as a verifier *retains 98.7% of the accuracy gains* from the learned verifier ensemble while reducing compute costs by three orders of magnitude – *saving 99.97% inference FLOPS* while still capturing an effective verification strategy (Section 6). Overall, our findings highlight that more reliable, scalable verification is possible even in the absence of ground-truth labels—paving the way for improved data filtering, model alignment, and inference-time decision-making.

## 2 Related Work

**LM Judges and Reward Models**: Both LM judges and reward models are promising approaches for evaluating language model outputs, but their high false positive rates limit their reliability [73]. LM judges can evaluate outputs without additional training [24, 47, 80], using approaches from simple prompting to chain-of-thought reasoning [47] and specialized fine-tuning [64, 75]. However, they face poor generalization across contexts [22, 59, 64] and systematic biases in position and self-preference [8, 37, 54, 96]. Similarly, while reward models have become central to model alignment [4, 14, 43], they struggle with noisy training signals from low inter-annotator agreement [3, 19, 53, 78] and learned biases favoring attributes like response length [18, 38, 67]. Recent work has improved individual verifier reliability through better data collection and chain-of-thought reasoning [83, 92], yet fundamental challenges persist [7, 20]. WEAVER advances beyond these approaches by combining multiple verification signals with adaptive weighting, thus leveraging the complementary strengths of weak verifiers while suppressing noise and reducing false positives.

**Weak Supervision**: WEAVER builds upon statistical techniques from weak supervision, which emerged as a framework for programmatically generating training labels by aggregating multiple weak sources [56, 58]. While initial work focused on classification tasks [23, 57], recent advances have expanded to handle multi-task settings [66] and structured prediction [77]. WEAVER applies this framework to answer verification, treating binary imperfect verification signals (e.g. reward models and LM judges) as weak supervision voters that classify candidate solutions as correct or incorrect.

This novel application combines predictions by converting these diverse signals into binary verdicts, enabling WEAVER to learn better verification strategies from weak but complementary verifiers.

**Verification as another compute axis and aggregation:** Recent work has explored verification as a new scaling axis [9, 41, 46, 69, 73, 93]. However this work limits their analysis to one verifier, and instead scale how many times to verify [93]. Approaches that do leverage multiple verifiers often rely on substantial amounts of labeled data for aggregation or creating specialized verifiers [36, 41]. With WEAVER, we show that it is possible to combine verifiers without ground truth labels, even when they are not specialized. Other work has focused on combining multiple verifiers for post-training the base model using RLHF [21, 81, 84]. We note that this is a direction of future work (Section 7).

# 3 Preliminaries

First, we define the problem of how to select among repeated samples. We then define verifiers and key evaluation metrics, including the generation-verification gap.

**Problem Definition** Let $q \in \mathcal{Q}$ be a input text query, and let $r \in \mathcal{R} \sim \mathcal{M}(q)$ be a corresponding response sampled from language model $\mathcal{M}$ with non-zero temperature. For a given query-response pair $(q,r)$, we define $y : \mathcal{Q} \times \mathcal{R} \to \{0,1\}$ such that $y(q,r)$ is the correctness label of $r$ for $q$. We are given an unlabeled test dataset $\mathcal{D}^{\text{test}} = \{(q_i, \boldsymbol{r_i})\}_{i=1}^n$, where $\boldsymbol{r_i} = \{r_{ij}\}_{j=1}^K$ consists of $K$ repeatedly sampled responses from $\mathcal{M}$ for each $q_i$. We do not have access to true labels $y_{ij} := y(q_i, r_{ij})$ for any $i,j$.

For each $(q_i, \boldsymbol{r_i}) \in \mathcal{D}^{\text{test}}$, our goal is to select a correct response $j^\star \in [K]$ that satisfies $y_{ij^\star} = 1$. We can broadly describe this selection rule using a scoring function $f : \mathcal{Q} \times \mathcal{R} \to \mathbb{R}$, namely $j^\star := \operatorname{argmax}_j f^\star(q_i, r_{ij})$.

**Using verifiers** A verifier, either a reward model or an LM prompted as a judge, can be expressed as a scoring function on query-response pairs $v : \mathcal{Q} \times \mathcal{R} \to \mathbb{R}$. For reward models, the verifier score is continuous, while for LM judges, the verifier score is typically discrete (for our setup, we use $[0,1]$ and $\{0,1\}$, respectively). We assume that we have access to multiple verifiers $\mathcal{V} = v_1, ..., v_m$. We apply each of the $m$ verifiers to each $(q_i, r_{ij})$, for a total of $nmK$ scores on $\mathcal{D}^{\text{test}}$ with $s_{ijk} := v_k(q_i, r_{ij})$. We aim to use $\mathcal{V}$ to construct a good verification strategy $f$.

**Evaluation metrics** The Pass@K metric is the probability that there exists a correct response among $K$ generated responses: $\text{Pass@K} = \frac{1}{n} \sum_{i=1}^n \mathbf{1}(\exists j \in [K] : y_{ij} = 1)$. This metric is independent of the verification strategy, and depends on the choice of $\mathcal{M}$, $K$, and the task dataset. The success rate of a verification strategy $\hat{f}$ is $\frac{1}{n} \sum_{i=1}^n y_{i\hat{j}}$, where $\hat{j} = \operatorname{argmax}_{j \in [k]} \hat{f}(q_i, r_{ij})$. Success rate is dependent on the verification strategy and bounded by Pass@K, and equality is obtained with oracle verification (i.e., $\hat{f} = f^\star$ can always select a correct $j$ as long as it exists).

We define the *generation-verification gap* as Pass@K - Success Rate. A large positive gap indicates that although correct answers are generated, the verification strategy fails to select them consistently. We aim to close this gap and will use it to evaluate verification strategies.

# 4 WEAVER: A Framework for Weak Verifier Aggregation

In Section 4.1, we demonstrate that naively averaging multiple verifier scores to select responses significantly underperforms weighted ensembles; however, common methods for computing weights require labeled data [65, 89]. We introduce WEAVER (Section 4.2), a method for weighted aggregation of verifier scores with minimal data that draws inspiration from Weak Supervision.

## 4.1 How to aggregate multiple verifiers: weighted vs unweighted ensembles

A straightforward approach for using multiple verifiers is a naive ensemble—selecting the response with the highest average verifier score: $f(q_i, r_{ij}) = \frac{1}{m} \sum_{k=1}^m s_{ijk}$. This approach [41] does not consider the relative accuracy of verifiers. However, we observed that there is significant variation in the success rates of individual verifiers—spanning a range of up to 37.5%—suggesting that naive ensembles could be suboptimal (Table 14).

An alternative is to use a weighted ensemble. One approach is to use a labeled dataset to identify and use the top-performing verifier, effectively assigning a weight of 0 to discarded verifiers. Other strategies

include using Logistic Regression or a Naive Bayes classifier, where the scoring function $f(q_i, r_{ij})$ is the probability $\Pr(y_{ij} = 1 | s_{ij1}, ..., s_{ijm})$, which are fit using labeled data and can be either modeled as a logistic function or factorized using Bayes' rule and independence assumptions, respectively.

In Figure 2, we compare a naive ensemble with weighted ensembles for several tasks, using Llama 3.3 70B Instruct to generate responses and using a collection of 33 7B-72B reward models and LM judges as verifiers. We see that using a weighted ensemble can achieve up to 11.2 points higher success rate than the naive ensemble. However, all weighted ensembles shown are "oracle" methods: they are computed using $y_{ij}$ for all $i \in [n], j \in [K]$, although in practice these labels are unknown for $\mathcal{D}^{\text{test}}$. In fact, when we instead use $0.01n$ labeled samples, accuracy drops by 20.1% on average (Table 15). This raises the question of how to best construct weighted ensembles with limited labeled data.

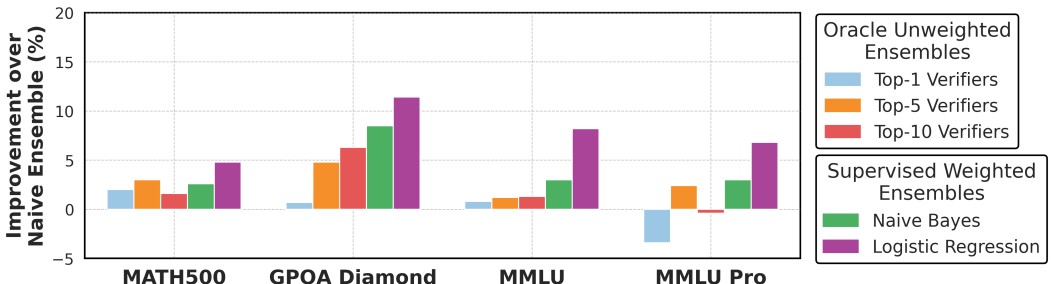

Figure 2: **Weighted Verifier Ensembles Outperform Naive Verifier Ensemble**: By using oracle data to keep the best verifiers (i.e. *top-K verifier ensembles*) or learn aggregation weights for verifiers (i.e. *supervised weighted ensembles*), we can improve beyond naive combinations of the verifiers available by 3.6% and 7.8%, on average, respectively.

## 4.2 WEAVER: weighted ensembling of verifier scores with minimal labeled data

We first describe the WS method we use in WEAVER to construct a weighted ensemble over binary verifier scores. Because verifiers often produce scores in inconsistent formats and exhibit low accuracies—challenges not typically encountered in traditional WS—we introduce a binarization and verifier discarding strategy in Appendix B.2 to discard low-quality verifiers and ensure that only sufficiently reliable binary scores are used as input to the WS method.

### 4.2.1 Weak Supervision Algorithm

In Weak Supervision, the input is an unlabeled dataset, where each entry has multiple binary "votes" on the true label. Applied to our setting, each entry is a query-response pair, forming a dataset of size $nK$, and verifier scores $s_{ijk}$ are binarized into votes $\bar{s}_{ijk} \in \{0, 1\}$ for all $i, j, k$. Our goal is to predict the probability that a response is correct $\Pr(y_{ij} = 1 | s_{ij1}, ..., s_{ijm})$ for all $i, j$.

**WS model** We can view all $y_{ij}$ as samples of an unknown random variable $Y$ and each $\bar{s}_{ijk}$ across $i, j$ as samples of a random variable $S_k$. WS then defines a latent variable graphical model over the random binary vector $\{Y, S_1, ..., S_m\}$, where $Y$ is latent while $S_1, ... S_m$ are observable. While existing WS methods assume various models, one common assumption is that $S_i \perp S_j | Y$ for each $S_i, S_j$. That is, $S_i$ and $S_j$ are conditionally independent given $Y$; intuitively, each verifier captures independent aspects of the correctness of the response. Under this assumption, we can write the posterior as the following, for some given binary verifier scores $\{\bar{s}_1, ..., \bar{s}_m\}$:

$$\Pr(Y = 1 | S_1 = \bar{s}_1, ..., S_m = \bar{s}_m) = \frac{\prod_{i=1}^{m} \Pr(S_i = \bar{s}_i | Y = 1) \Pr(Y = 1)}{\Pr(S_1 = \bar{s}_1, ..., S_m = \bar{s}_m)} \quad (1)$$

The weighted ensemble score for each query-response pair can be written in terms of: 1) $\Pr(S_1 = \bar{s}_1, ..., S_m = \bar{s}_m)$, which we compute via the law of total probability as $\sum_{y' \in \{0,1\}} \Pr(Y = y') \prod_{i=1}^{m} \Pr(S_i = \bar{s}_i | Y = y')$ using the conditional independence assumption; 2) $\Pr(Y = 1)$, which can be estimated from $\mathcal{D}^{\text{dev}}$; and 3) $\Pr(S_i = \bar{s}_i | Y = 1)$, or equivalently $\Pr(S_i = 1 | Y = 1)$, which is the verifier's "accuracy parameter"—this cannot be computed directly since we do not have access to $Y$. Next, we discuss how to estimate accuracy parameters, $\Pr(S_i = 1 | Y = 1)$, without labels.

**WS parameter estimation**    We outline a parameter estimation technique first introduced in [56]. Due to the assumption that $S_i \perp S_j | Y$, the following equation holds:

$$\Pr(S_i, S_j) = \Pr(S_i, S_j | Y = 1)\Pr(Y = 1) + \Pr(S_i, S_j | Y = 0)\Pr(Y = 0)$$
$$= \Pr(S_i | Y = 1)\Pr(S_j | Y = 1)\Pr(Y = 1) + \Pr(S_i | Y = 0)\Pr(S_j | Y = 0)\Pr(Y = 0) \quad (2)$$

Note that $\Pr(S_i, S_j)$ can be computed from the known verifier scores, and $\Pr(Y = 1)$ is estimated from $\mathcal{D}^{\mathrm{dev}}$. Then, equation 2 is a quadratic equation over the accuracy parameters. We can write this equation for every pair $S_i, S_j$, and for every pair of values $\{0,1\}^2$ they can take. Furthermore, we can write another type of equation over the accuracy parameters:

$$\Pr(S_i = 1) = \Pr(S_i = 1 | Y = 1)\Pr(Y = 1) + \Pr(S_i = 1 | Y = 0)\Pr(Y = 0) \quad (3)$$

This is a consistency property that holds regardless of the conditional independence assumption, and we can write this equation for each of the $m$ $S_i$'s. Because we know that the accuracy parameters should follow equations 2 and 3, we can construct an objective function that aims to minimize the difference between the left and right hand sides of these equations. We write this efficiently in matrix notation. Let $P \in \mathbb{R}^{2 \times 2}$ be a diagonal matrix with diagonal $[\Pr(Y = 0)\ \Pr(Y = 1)]$. Define $\mu \in \mathbb{R}^{2m \times 2}$ to be the matrix of accuracy parameters, and define $O \in \mathbb{R}^{2m \times 2m}$ to be a matrix over the joint probabilities of pairs of $S_i, S_j$; more formally:

$$\mu_{2i-1:2i,1:2} = \begin{bmatrix} \Pr(S_i = 0 | Y = 0) & \Pr(S_i = 0 | Y = 1) \\ \Pr(S_i = 1 | Y = 0) & \Pr(S_i = 1 | Y = 1) \end{bmatrix}, O_{2i-1:2i,2i-1:2i} = \begin{bmatrix} \Pr(S_i = 0) & 0 \\ 0 & \Pr(S_i = 1) \end{bmatrix} \forall i \in [m]$$

$$O_{2i-1:2i,2j-1:2j} = \begin{bmatrix} \Pr(S_i = 0, S_j = 0) & \Pr(S_i = 0, S_j = 1) \\ \Pr(S_i = 1, S_j = 0) & \Pr(S_i = 1, S_j = 1) \end{bmatrix} \forall i \neq j \in [m] \quad (4)$$

Let off-diag denote the elements of a matrix that lie outside its $2 \times 2$ block diagonal. Then, to estimate $\mu$ that satisfies both equations 2 and 3, we have the following objective:

$$\mathrm{minimize}_{\mu} \left\| O_{\mathrm{off\text{-}diag}} - (\mu P \mu^T)_{\mathrm{off\text{-}diag}} \right\|^2 + \left\| \mathrm{diag}(O) - \mu P \mathbf{1}^T \right\|^2 \quad (5)$$

We optimize 5 using gradient descent to estimate the verifier accuracy parameters. These estimates are then used in Eq. (1) to select the response with the highest estimated posterior. We provide a full derivation in Sec. B.1.

# 5    Results

In section 5.1, we provide empirical results on WEAVER's performance compared to other approaches for selecting responses in repeated sampling. In section 5.2, we study how WEAVER's performance scales along several axes: the number of responses, model size, verifier counts, and inference compute.

**Datasets, Verifiers, and Baselines**    Our reward models range in size from 8B to 72B, are all open-source, and are obtained from RewardBench [39], a popular evaluation tool for reward models. We prompt open-source language models from Chatbot Arena [13] to serve as judges. Unless specified, we use Llama 3.3 70B Instruct to generate responses and use all 33 reward models and judges. We evaluate on MATH500, GPQA Diamond, MMLU College, and MMLU Pro. See Sec. C.1 for more details.

We compare WEAVER against verifier-free baselines as well as standard verification strategies. First Sample, also known as Pass@1, only uses the first response and does not scale test-time compute or verification. Majority Voting involves repeated sampling but not verification, picking the most common final answer from the responses [5, 10, 70]. We compare against the highest scoring reward model and a naive ensemble of the top-10 reward models on RewardBench. We also evaluate two recently proposed methods that scale verification but do not use multiple verifier models or weighted ensembles: Self-Verification [93] and Multi-Agent Verification [41]. Lastly, we report the oracle Pass@K rate, which establishes an upper bound for the success rate of these verification strategies.

## 5.1    WEAVER Shrinks the Gap with Frontier LMs

In Table 1, we evaluate WEAVER along with baseline verification methods, the first sample performance of frontier LMs, and the Pass@100 metric. We use LlaMA 3.3 70B Instruct to generate $K = 100$ responses per query. We find that WEAVER's weighted ensembling of multiple verifiers allows us to outperform majority vote by $14\%$ and come within $5.7\%$ of the Pass@100 oracle metric. Furthermore, WEAVER rivals the performance of frontier reasoning models—coming within $0.5\%$ of OpenAI's o3-mini [52]—even though we use a non-reasoning model for generation.

Table 1: **WEAVER Outperforms Baseline Verification Methods and Shrinks Gap with Frontier LMs.**

| | Methodology | Generations ($K$) | Datasets | | | | |
|---|---|---|---|---|---|---|---|
| | | | MATH 500 | GPQA Diamond | MMLU College | MMLU Pro | Average |
| **Baselines** | First Sample | 1 | 78.0% | 42.9% | 82.6% | 69.9% | 68.4% |
| | Majority Voting | 100 | 83.0% | 47.4% | 84.1% | 74.4% | 72.2% |
| | Highest Scoring RM on RewardBench [39, 49] | 100 | 78.2% | 49.7% | 86.0% | 77.0% | **72.7%** |
| | Naïve Ensemble of Top-10 RMs on RewardBench [39] | 100 | 75.4% | 41.3% | 88.1% | 71.4% | 69.1% |
| | Self-Verification [93] | 100 | 78.1% | 43.1% | 82.0% | 69.5% | 66.9% |
| | Multi-Agent Verification [41] | 100 | 81.3% | 47.8% | 84.1% | 72.6% | 71.6% |
| | WEAVER | 100 | 93.4% | 66.4% | 94.9% | 90.2% | **86.2%** |
| **Frontier Approaches** | GPT-4o [51] | 1 | 77.4% | 35.9% | 87.1% | 75.4% | 69.0% |
| | Claude 3.7 Sonnet [2] | 1 | 69.2% | 48.0% | 86.1% | 78.1% | 70.4% |
| | Llama 4 Maverick [48] | 1 | 87.6% | 68.9% | 91.1% | 81.0% | 82.2% |
| | o3-mini [52] | 1 | 94.4% | 74.0% | 92.2% | 86.0% | 86.7% |
| | Oracle Verification (Pass@100) | 100 | 98.6% | 81.0% | 96.0% | 92.0% | **91.9%** |

## 5.2 WEAVER Improves Compute-Accuracy Trade-Off for Scaling

By proposing to combine multiple weak verifiers instead of one, we introduce yet another axis for test-time scaling. In this section, we study how well scaling verification with WEAVER interacts with common previously studied axes for verification, summarized in Table 2.

Table 2: **Scaling Dimensions for Generation and Verification Models**

| Scaling Dimension | Base Model | Verifier Type | Visuals |
|---|---|---|---|
| **Sample Count**: More Generations | Temperature-based sampling | Majority Vote, Weak Verifier, Top-K, WEAVER | Figure 3 |
| **Model Size**: Larger Models | Llama 8B $\rightarrow$ 70B | RM-8B $\rightarrow$ RM-70B | Table 3 |
| **Verifier Count**: More Models | Llama 8B/70B | RMs and LM Judges | Figure 4 |
| **Inference Compute**: More FLOPs for Gen./Ver. | Temp-based sampling | Weak Verifiers + WEAVER | Figure 5 |

**(1) Scaling Candidate Generations**: we study the performance of verification methods as we increase the number of repeated samples in Fig. 3. Based on prior work [4, 12], as the number of responses increases, we are more likely to see a correct response (i.e. Pass@K increases), and hence more likely to select a correct response given a good verification strategy. However, differences in verification translate into different scaling rates. We evaluate the performance of WEAVER and baselines for $K = 2^0$ to $2^{10}$, comparing to o3-mini and Pass@K as well. Across all tasks, WEAVER yields the most substantial gains when scaling the number of generations. WEAVER consistently narrows the generation-verification gap with the oracle upper bound (Pass@K) while alternative verification strategies plateau after a few generations. The effect is particularly pronounced on difficult tasks like GPQA. We detail the scaling trends observed in Fig. 3 in Sec. C.3.

**(2) Scaling Model Sizes:** In Table 3, we study how WEAVER applied on smaller models (both verifiers and for generating responses) can allow us to match the performance of larger models, enabling weak-to-strong verification. We consider an 8B setting—using LlaMA 3.1 8B to generate responses along with 8B verifiers—and compare this to a 70B setting (LlaMA 3.3 70B Instruct, 8B-72B verifiers) as well as o3-mini. We see that WEAVER applied at the 8B scale comes within 2.2% of the majority vote baseline at the 70B scale, and WEAVER at 70B comes within 0.5% of o3-mini, demonstrating a weak-to-strong verification phenomenon.

**(3) Scaling Verifier Count:** Two axes for scaling verification are **(1)** the number of verifiers used or **(2)** the number of scores sampled from each verifier. Fig. 4 shows how the performance changes as we ensemble more verifiers (up to 15) using naïve ensembling and WEAVER. We can see that aggregating verifiers continuously improves performance over using the top-1 verifier by up to 8.5%. However, these gains diminish as we continue to add models to the ensemble. These results reflect the classic ensemble bias-variance tradeoff: initial gains arise from variance reduction as independent verifier errors are averaged out, but performance plateaus as additional verifiers contribute redundant

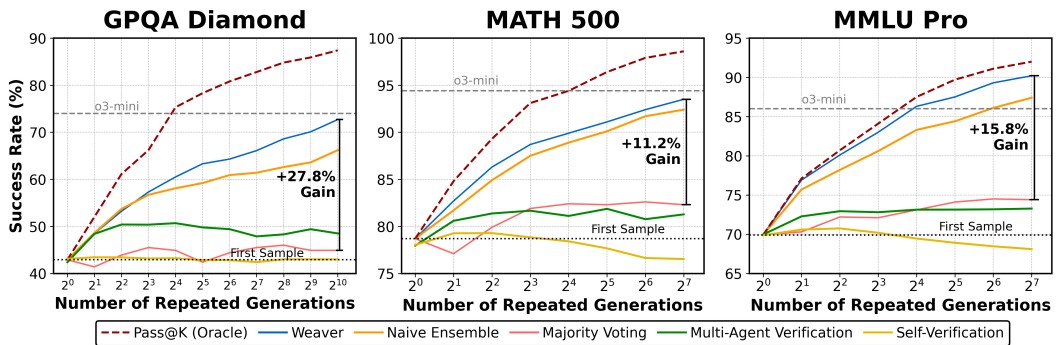

Figure 3: **Scaling Generations Boosts Performance with WEAVER**: The generation-verification gap shrinks when increasing $K$ and leveraging WEAVER, outperforming alternative verification methods by an average 18.3%.

Table 3: **WEAVER Reduces Gap between Model Classes: 8B and 70B, 70B and Frontier LM**

| Generator Model | Verifier Model | Aggregation Strategy | Datasets | | | | Average |
| --- | --- | --- | --- | --- | --- | --- | --- |
| | | | MATH | GPQA Diamond | MMLU College | MMLU Pro | |
| Llama 3.1 8B Instruct | N/A 8B and below | Majority Vote WEAVER | 69.0% 80.0% | 30.5% 47.1% | 72.7% 85.7% | 56.4% 67.2% | 57.2% **70.0%** |
| | | $\triangle$ w. WEAVER | +11.0% | +16.6% | +13.0% | +10.2% | +12.8% |
| Llama 3.3 70B Instruct | N/A 72B and below | Majority Vote WEAVER | 83.0% 93.4% | 47.4% 66.4% | 84.1% 94.9% | 74.4% 90.2% | 72.2% **86.2%** |
| | | $\triangle$ w. WEAVER | +10.4% | +9.0% | +10.8% | +15.8% | +14.0% |
| o3-mini | N/A | First Sample | 94.4% | 74.0% | 92.2% | 86.0% | **86.7%** |

information due to correlated biases on systematically difficult examples. [1]. We also include results on scaling the number of scores per verifier in Appendix C.5. We find that scaling the number of verifiers yields better performance than sampling multiple scores from the same verifier (i.e., via prompt tuning or temperature variation). However, this method is complementary and could be used in conjunction with scaling the verifier count.

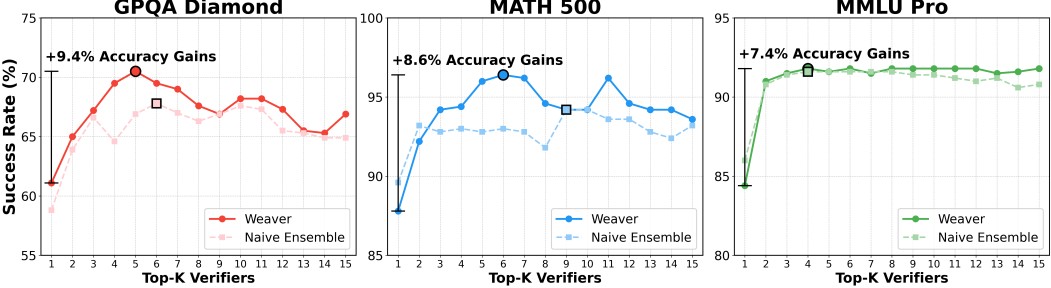

Figure 4: **WEAVER Benefits from Adding Top-$K$ Verifiers.**

**(4) Scaling Test-Time Compute:** We study how performance scales in the total compute used for both verification and repeated generations. Figure 5 shows the relationship between inference-time compute and success rate for different generation-verification systems. For each method, we scale the number of generations exponentially from 1 to 100 and plot the required inference compute for generation and verification together versus the success rate. Note that Fig. 5 differs from Fig. 3, since Majority Voting requires 0 verification inference calls while WEAVER requires 30+ calls for the weak verifiers. We find that WEAVER achieves the highest maximum success rate; notably, majority voting plateaus at around $2^2$ to $2^3$ ExaFLOPs per query while WEAVER continues scaling until 512

ExaFLOPs. However, the additional compute required for WEAVER can be prohibitive. We explore how to reduce this computational burden while retaining WEAVER's performance in the next section.

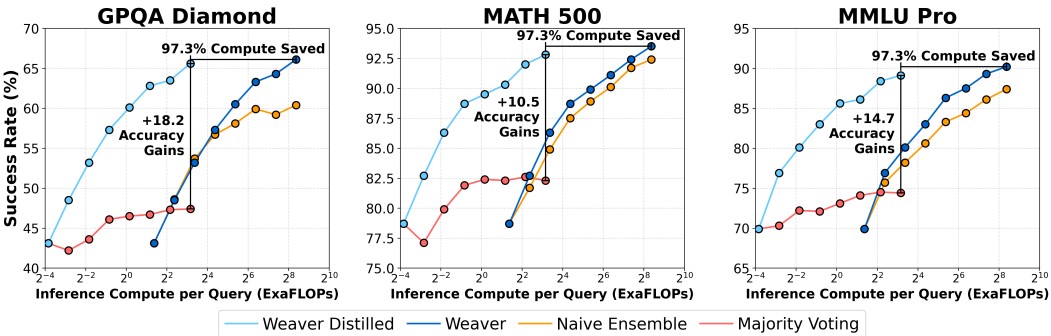

Figure 5: **WEAVER Improves the Accuracy-Compute Performance Trade-Offs.**

# 6   WEAVER Distillation: Improving Verification Efficiency at Inference

We explore distillation strategies for fine-tuning a smaller LM as a task-specific verifier. In particular, we train *cross-encoders*; the input is a concatenated query-response pair, while the output is WEAVER's pseudolabel generated from Weak Supervision, namely $\Pr(y_{ij} = 1 | s_{ij1},...,s_{ijm})$ (see Section 4). Cross-encoders are used for a variety of NLP tasks requiring comparison between two textual inputs, including search result reranking [50], question-answer matching [25], semantic textual similarity assessment [60], natural language inference [16], and response quality evaluation in dialogue systems [31]. For the model, we selected ModernBERT-Large (396M) [85]. For more details, please see Appendix C.6.

Figure 6, shows the performance of WEAVER on the Llama-70B generations against the cross-encoder on GPQA Diamond. Across tasks, we find that the distilled cross-encoder is able to capture 98.2% of the performance of WEAVER. Furthermore, while it would cost 35.35 exaFLOPs to run all the verifiers needed for preparing and evaluating WEAVER, while running a 400M cross-encoder costs 1.01 exaFLOPs for evaluating 100 samples, *this is more than three orders of magnitude in compute cost reduction, saving 99.97% of the FLOPs originally required for running the 70B verifiers*. We also outperform majority voting by 23.2% while only incurring a 0.57% increased inference cost over only generating the responses. We see similar results for additional datasets in Figure 14 (Sec. C.6).

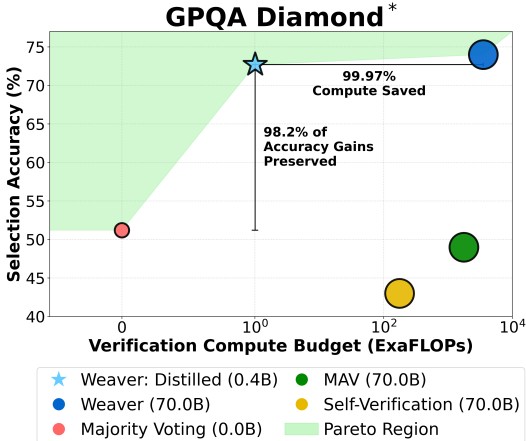

Figure 6: **Distilling WEAVER into a 400M Cross-Encoder Almost Entirely Captures the Performance of WEAVER, Yielding 99.97% Compute Savings.** *We train/evaluate on an 80:20 split.

These results suggest that, through distillation, we can capture the combined strengths of the weak verifiers used for WEAVER, and deploy generalizable and lightweight cross-encoders that use only a fraction of the parameters used for generation. This reduces our hardware constraints considerably; *rather than utilizing an 8-GPU node per 70B verifier (i.e. Nvidia H200s with 80B memory), we only require a single A100 GPU with 32GB of memory for our cross-encoder.* Our economical approach has the added benefit of leaving the original generation model *frozen* (i.e. without any parameter tuning), allowing us to preserve the original model's distribution at inference [87, 91].

# 7 Conclusion

WEAVER offers a flexible and scalable framework for aggregating weak verifiers, achieving strong performance across diverse reasoning tasks while remaining computationally efficient. Our results show that combining many weak verifier can further close the generation verification gap with a fraction of the compute. Looking forward, several directions stand to extend WEAVER's impact. First, designing specialized verifiers can further enhance task-specific accuracy. Second, WEAVER can be leveraged for RLHF for better supervision signals. Multimodal extensions of WEAVER represent another exciting frontier. Finally, the compute efficiency of WEAVER opens the door to improved on-device verification.

# 8 Acknowledgements

We thank Simran Arora, Daniel Biderman, Bradley Brown, Owen Dugan, Ryan Ehrlich, Neel Guha, Simon Guo, Jordan Juravsky, Jerry Liu, Avanika Narayan, Benjamin Spector, and Benjamin Viggiano for their constructive feedback during the composition of the paper. We would also like to thank our collaborators at the Stanford Artificial Intelligence Laboratory (SAIL) and TogetherAI.

We gratefully acknowledge the support of NIH under No. U54EB020405 (Mobilize); NSF under Nos. CCF2247015 (Hardware-Aware), CCF1763315 (Beyond Sparsity), CCF1563078 (Volume to Velocity), and 1937301 (RTML); US DEVCOM ARL under Nos. W911NF-23-2-0184 (Long-context) and W911NF-21-2-0251 (Interactive Human-AI Teaming); ONR under No. N000142312633 (Deep Signal Processing); Stanford HAI under No. 247183; Google DeepMind; Google Research; Google Cloud; NXP; Xilinx; LETI-CEA; Intel; IBM; Microsoft; NEC; Toshiba; TSMC; ARM; Hitachi; BASF; Accenture; Ericsson; Qualcomm; Analog Devices; Salesforce; Total; the HAI-GCP Cloud Credits for Research program; the Stanford Data Science Initiative (SDSI); members of the Stanford DAWN project: Meta, Google, and VMWare; and members of the Stanford SEAMS project: IBM and Felicis. The U.S. Government is authorized to reproduce and distribute reprints for Governmental purposes notwithstanding any copyright notation thereon. Any opinions, findings, and conclusions or recommendations expressed in this material are those of the authors and do not necessarily reflect the views, policies, or endorsements, either expressed or implied, of NIH, ONR, or the U.S. Government.

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

# A    Table of Contents

# B WEAVER Methodology

## B.1 Weak Supervision Model

We can construct a data generating model over response correctness $y$ and the binary verifier outputs $\bar{s}$. The model is defined as:

Response Correctness: $\quad y_{ij} \sim \text{Bernoulli}(\pi) \, \forall i \in [n], j \in [K],$

$$\text{Verifier Score:} \quad \bar{s}_{ijk} \,|\, y_{ij} \sim \begin{cases} \text{Bernoulli}(w_{k,1}), & \text{if } y_{ij} = 1, \\ \text{Bernoulli}(1 - w_{k,0}), & \text{if } y_{ij} = 0, \end{cases} \forall i \in [n], j \in [K], k \in [m]$$

where:

- $\pi$ is the probability that a response is correct.
- $w_{k,1}$ is the true positive rate (TPR) of verifier $k$, and $w_{k,0}$ is the true negative rate (TNR), which we refer to as the verifier's *accuracy parameters.*

Here, each verifier $k$ emits a binary score $\bar{s}_{ijk} \in \{0,1\}$, which is assumed to be a noisy indicator of whether response $y_{ij}$ is correct. The likelihood of the verifier's binary output $X_{ijk} \in \{0,1\}$ is:

$$\Pr(S_k = \bar{s}_{ijk} \,|\, Y = y_{ij}) = \begin{cases} w_{k,1}, & \text{if } y_{ij} = 1 \text{ and } \bar{s}_{ijk} = 1, \\ 1 - w_{k,1}, & \text{if } y_{ij} = 1 \text{ and } \bar{s}_{ijk} = 0, \\ w_{k,0}, & \text{if } y_{ij} = 0 \text{ and } \bar{s}_{ijk} = 0, \\ 1 - w_{k,0}, & \text{if } y_{ij} = 0 \text{ and } \bar{s}_{ijk} = 1. \end{cases}$$

We are interested in estimating the correctness of a response $y_{ij} \in \{0,1\}$ based on assessments from multiple verifiers $\bar{s}_{ij} = \{\bar{s}_{ij1}, \bar{s}_{ij2}, ..., \bar{s}_{ijm}\}$. Applying Bayes' Rule, we get:

$$\Pr(y_{ij} = 1 \,|\, S = \bar{s}_{ij}) = \frac{\Pr(S = \bar{s}_{ij} \,|\, y_{ij} = 1) \Pr(y_{ij} = 1)}{\Pr(\bar{s}_{ij})} \tag{6}$$

where $\Pr(\bar{s}_{ij}) = \sum_{y' \in \{0,1\}} \Pr(\bar{s}_{ij} \,|\, y_{ij} = y') \Pr(y_{ij} = y')$.

Eq. (6) requires evaluating the full conditional likelihood:

$$\Pr(S = \bar{s}_{ij} \,|\, y_{ij}) = \Pr(S_1 = \bar{s}_{ij1}, S_2 = \bar{s}_{ij2}, ..., S_m = \bar{s}_{ijm} \,|\, y_{ij}),$$

which is a joint distribution over $m$ binary random variables. Since each verifier $S_k \in \{0,1\}$ is binary, then there are $2^m$ possible verifier output configurations for $S \in \{0,1\}^m$. This results in $2^m - 1$ free parameters per class label to construct the distribution $\Pr(S = \bar{s}_{ij} \,|\, y_{ij})$.

**Conditional Independence Assumption** To avoid this exponential blowup, we can assume that the verifiers provide *conditionally independent* outputs:

$$P(S \,|\, y) = \prod_{k=1}^{m} P(S_k \,|\, y),$$

which reduces the number of parameters from $O(2^m)$ to $O(m)$ and enables efficient inference, under the assumption that each verifier provides unique information about the correctness of a response.

Then, Eq. (6) simplifies to a Naive Bayes-style estimator:

$$\begin{aligned} \Pr(y_{ij} = 1 \,|\, S = \bar{s}_{ij}) &= \frac{\Pr(S_1 = \bar{s}_{ij1}, ..., S_m = \bar{s}_{ijm} | y_{ij} = 1) \Pr(y_{ij} = 1)}{\Pr(S = \bar{s}_{ij})} \\ &= \frac{\Pr(y_{ij} = 1) \prod_{k=1}^{m} \Pr(S_k = \bar{s}_{ijk} \,|\, y_{ij} = 1)}{\sum_{y' \in \{0,1\}} \Pr(y_{ij} = y') \prod_{k=1}^{m} \Pr(S_k = \bar{s}_{ijk} \,|\, y_{ij} = y')} \end{aligned} \tag{7}$$

The parameters in Eq. (7) include:

- The prior probability of correctness $\pi = \Pr(y_{ij} = 1)$.
- The verifier-specific conditional likelihoods $P(S_k \,|\, y_{ij})$.

### B.1.1 Parameter Estimation

**Supervised Setting**   When ground-truth labels $y_{ij}$ are available, parameter estimation reduces to computing empirical frequencies. We can estimate the prior as:

$$\hat{\pi} = \frac{1}{N}\sum_{i,j}\mathbf{1}\{y_{ij}=1\}$$

For each verifier $k$, we could estimate:

$$\hat{w}_{k,1} = \frac{\sum_{i,j}\mathbf{1}\{y_{ij}=1\}\cdot\mathbf{1}\{S_k=1\}}{\sum_{i,j}\mathbf{1}\{y_{ij}=1\}},$$
$$\hat{w}_{k,0} = \frac{\sum_{i,j}\mathbf{1}\{y_{ij}=0\}\cdot\mathbf{1}\{S_k=0\}}{\sum_{i,j}\mathbf{1}\{y_{ij}=0\}}.$$

**Weak Supervised Setting**   When a few labeled $y_{ij}$ are available, we can use it to estimate $\pi$, but we still need to estimate the verifier accuracy parameters $w_{k,1}, w_{k,0}$ to compute $\prod_{k=1}^{m}\Pr(S_k=\bar{s}_{ijk}|y_{ij}=1)$. Instead of using labeled data, we estimate accuracy parameters using moment matching. In particular, we match observable second moments of verifier outputs to the model-implied moments under conditional independence assumptions, based on an approach from [27].

**Pairwise Statistics.**   For each pair of verifiers $k_1, k_2$ and binary outputs $a,b\in\{0,1\}$, we can express the joint probability of their outputs using the marginalization rule and the conditional independence assumption:

$$\Pr(S_{k_1}=a, S_{k_2}=b) \tag{8}$$
$$= \Pr(S_{k_1}=a|Y=1)\Pr(S_{k_2}=a|Y=1)\Pr(Y=1) + \Pr(S_{k_1}=b|Y=0)\Pr(S_{k_2}=b|Y=0)\Pr(Y=0)$$

where the conditional distributions for verifier $k$ are:

$$\Pr(S_k=a\,|\,y=1) = \begin{cases} w_{k,1}, & a=1, \\ 1-w_{k,1}, & a=0, \end{cases} \quad \Pr(S_k=a\,|\,y=0) = \begin{cases} 1-w_{k,0}, & a=1, \\ w_{k,0}, & a=0. \end{cases}$$

**Marginal Statistics.**   Similarly, each verifier's marginal distribution can be written as:

$$\Pr(S_k=1) = \Pr(S_k=1|Y=1)\Pr(Y=1) + \Pr(S_k=1|Y=0)\Pr(Y=0) \tag{9}$$

Note that this equation holds true regardless of the conditional independence assumption.

**Estimation method**

- Construct the second order moment matrix $O\in\mathbb{R}^{(2m)\times(2m)}$, where:

$$O_{2i-1:2i,2i-1:2i} = \begin{bmatrix} \Pr(S_i=0) & 0 \\ 0 & \Pr(S_i=1) \end{bmatrix} \forall i\in[m]$$
$$O_{2i-1:2i,2j-1:2j} = \begin{bmatrix} \Pr(S_i=0,S_j=0) & \Pr(S_i=0,S_j=1) \\ \Pr(S_i=1,S_j=0) & \Pr(S_i=1,S_j=1) \end{bmatrix} \forall i\neq j\in[m] \tag{10}$$

- Construct the conditional probability matrix $\mu\in\mathbb{R}^{(2V)\times 2}$, where each row encodes:

$$\mu_{2k+a,b} = \Pr\big(S_k=a\,|\,y=b\big)$$
$$\mu = \begin{bmatrix} w_{k_1,0} & 1-w_{k_1,1} \\ 1-w_{k_1,0} & w_{k_1,1} \\ ... & ... \\ w_{k_m,0} & 1-w_{k_m,1} \\ 1-w_{k_m,0} & w_{k_m,1} \end{bmatrix}$$

- Label prior matrix $P\in\mathbb{R}^{2\times 2}$ is a diagonal matrix:

$$P = \begin{bmatrix} \Pr(y_{ij}=0) & 0 \\ 0 & \Pr(y_{ij}=1) \end{bmatrix}.$$

Then, equation 8 is equivalent to $O = \mu P \mu^\top$ on the entries off of the $2 \times 2$ block diagonal, and equation 9 is equivalent to $\mathrm{diag}(0) = \mu P \mathbb{1}^\top$. Therefore, we optimize the following loss to compute $\mu$.

$$\text{minimize}_\mu \left\| O_{\text{off-diag}} - (\mu P \mu^T)_{\text{off-diag}} \right\|^2 + \left\| \mathrm{diag}(O) - \mu P \mathbf{1}^T \right\|^2,$$

By solving via gradient-descent, we obtain estimates of the verifier accuracy parameters $\{w_{k,1}, w_{k,0}\}$.

### B.1.2 Inference: computing response correctness probabilities

Once the accuracy parameters $\{w_{k,1}, w_{k,0}\}$ are estimated and $P(y_{ij} = 1)$ is computed from a small labeled development dataset, we can compute posterior correctness probabilities for each response:

$$\Pr(y_{ij} = 1 | S = \bar{s}_{ij}) \propto \Pr(y_{ij} = 1) \prod_{k=1}^{m} \Pr(S_k = \bar{s}_{ijk} | y_{ij} = 1)$$

$$\Pr(y_{ij} = 0 | S = \bar{s}_{ij}) \propto \Pr(y_{ij} = 0) \prod_{k=1}^{m} \Pr(S_k = \bar{s}_{ijk} | y_{ij} = 0)$$

Normalizing these, we have a full posterior $P(y_{ij} = 1 | S = \bar{s}_{ij})$, which provides a score with which we can select a response for each query.

## B.2 Adapting Weak Supervision to the Verification Setting

Given a set of verifiers, we describe two modifications to the verifiers that allow us to apply weak supervision to the verification setting. First, we discuss the motivation for how we binarize verifier outputs and filter out low-quality verifiers. These two steps before invoking Weak Supervision methods are critical, as WS assumes that verifiers are binary and are of sufficiently high signal. Second, we will describe our proposed method for binzarization and filtering, along with results.

### B.2.1 Motivation

**Binarization** The weak supervision algorithm described in the prior section requires binary verifier outputs. This is suitable for verifiers such as LM judges, which produce $\{0,1\}$ outputs, but verifiers that are reward models often produce continuous scores at different scales. In fact, naive thresholding can result in low-signal verifiers (e.g., outputting all 1s). *How do we convert reward models' continuous verifier scores into binary scores?*

**Filtering out low-quality verifiers** Beyond directly impacting the quality of an ensemble, low-quality verifiers reduce the effectiveness of Weak Supervision algorithms. *How do we discard verifiers that have low signal?*

- Skewed marginals: Consider a dataset where $\Pr(y = 1) \approx 0.5$ and we have a verifier with $\Pr(S_k = 1) \approx 0.99$. A skewed verifier with an extreme marginal (e.g., from naive thresholding for binarization) and near-constant outputs adds little information to the ensemble. It primarily increases noise in the objective in Eq. 5 and should thus be discarded. Yet, not at all verifiers that have extreme marginals add little signal; for instance, if instead $\Pr(y = 1) \approx 0.99$, a skewed verifier could be highly accurate. Therefore, the definition of a low-quality verifier depends on the distribution of correct responses.
- Breaking symmetry in the WS objective: a common assumption of Weak Supervision is that a majority of the verifiers have better-than-random accuracy [23]. Otherwise, there is a possibility that the WS algorithm can yield non-unique solutions; the terms in Eq. 5 are the joint probabilities over pairs of verifiers as well as their marginals, which do not uniquely determine if a verifier satisfies $w_{k,1}, w_{k,0} > 0.5$ or not. Therefore, it is critical to remove as many low-accuracy verifiers as possible to ensure that the optimization procedure converges to a unique solution.

### B.2.2 Adaptation method

We now describe our proposed method for binarization and filtering of verifiers, after which the Weak Supervision algorithm described in Appendix B.1.

Table 4: **Ablation of Development Set used for WEAVER Class Balance Estimation**: As our reward model threshold, we set it to the default of the 0.5.

| Approach | Model Size | Dev Set Size | Benchmarks | | | | |
|---|---|---|---|---|---|---|---|
| | | | MATH500 | GPQA Diamond | MMLU | MMLU Pro | **Average** |
| WEAVER | 70B | Naive Threshold (0.5 Threshold) | 88.1% | 52.0% | 92.4% | 83.5% | 79.0% |
| WEAVER | 70B | 1% | 90.4% | 67.1% | 91.1% | 87.0% | 84.5% |
| WEAVER | 70B | 5% | 92.4% | 72.7% | 93.5% | 90.4% | 87.5% |
| WEAVER | 70B | 20% | 92.4% | 72.7% | 93.5% | 90.4% | 87.5% |
| WEAVER | 70B | 100% | 92.4% | 72.7% | 93.5% | 90.4% | 87.5% |

Table 5: **Ablation of Verifier Selection Strategies for WEAVER**: Comparison of different strategies for dropping faulty verifiers by dataset using each verifier's marginal probability.

| Approach | Model Size | Verifier Selection | Benchmarks | | | | |
|---|---|---|---|---|---|---|---|
| | | | MATH500 | GPQA Diamond | MMLU | MMLU Pro | **Average** |
| WEAVER | 70B | No Dropped Verifiers | 90.4% | 52.0% | 91.1% | 84.2% | 79.4% |
| WEAVER | 70B | Low Marginals Dropped (Mostly Negative Verifiers) | **93.4%** | 60.6% | 91.7% | **91.0%** | 84.2% |
| WEAVER | 70B | High Marginals Dropped (Mostly Positive Verifiers) | 83.4% | 69.7% | 87.9% | 78.4% | 79.9% |
| WEAVER | 70B | Extreme Marginals Dropped (Mostly Positive or Negative) | 90.8% | **72.7%** | **92.4%** | 85.0% | **85.2%** |

Table 6: **Ablation of Adaptive Threshold Dev Set Size for WEAVER**.

| Approach | Model Size | Adaptive Threshold Dev Set Size | Benchmarks | | | | |
|---|---|---|---|---|---|---|---|
| | | | MATH500 | GPQA Diamond | MMLU | MMLU Pro | **Average** |
| WEAVER | 70B | 0.01 | 92.4% | 72.7% | 93.5% | 90.4% | 87.5% |
| WEAVER | 70B | 0.05 | 92.4% | 72.7% | 93.5% | 90.4% | 87.5% |
| WEAVER | 70B | 0.2 | 92.4% | 72.7% | 93.5% | 90.4% | 87.5% |
| WEAVER | 70B | 1.0 | 92.4% | 72.7% | 93.5% | 90.4% | 87.5% |

1. **Binarization**: We use a small amount of labeled samples (which we already are using to compute $\Pr(y=1)$) to determine a threshold for converting continuous verifier outputs to binary outputs. With only 5 to 10 labeled queries from benchmark development sets (which $\leq 1\%$ of the evaluation set), we can estimate thresholds that bolster performance by averages of 8.4% for the 70B models when compared to binary splitting along the median score of the dataset (Table 4).

2. **Filtering out low-quality verifiers**: To mitigate the impact of low-quality verifiers, we prune verifiers with extreme marginal behavior, depending on the class balance. For datasets with estimated class balance between 20% and 80%, we filter out verifiers with positive rates outside this range. If a dataset has fewer than 20% positive samples overall, we remove verifiers that predict positives more than 80% of the time. Conversely, for datasets with more than 80% positives, we drop verifiers that predict positives less than 20% of the time. We find that verifier pruning leads to 12.5% performance improvement for the 70B model setting (Table 5).

Table 7: **Performance Comparison Between Continuous and Discrete Logistic Regression**: For supervised fine-tuning on the verifier scores, the continuous model consistently outperforms the discrete variant across all datasets by avoiding the lossy conversion of floats to binary votes required for the discrete variant.

| | Continuous vs. Discrete Logistic Regression Performance (%) | | | | |
|---|---|---|---|---|---|
| **Method** | **Dataset** | | | | |
| | MATH 500 | GPQA | MMLU College | MMLU Pro | BBH |
| Discrete LR | 93.1 | 74.3 | 87.5 | 87.1 | 90.1 |
| Continuous LR | **97.2** | **78.1** | **90.4** | **92.0** | **96.5** |
| **Improvement** | +4.1 | +3.8 | +2.9 | +4.9 | +6.4 |

Table 8: **Number of Unique Extracted Answers vs. Positive:Negative Sample Ratio per Query - Correlations for Llama 3.1 Instruct Models**

| | Llama 3.1 8B Instruct Correlation Metrics | | |
|---|---|---|---|
| **Dataset** | **Metric Type** | | |
| | Pearson | Spearman | Kendall's Tau |
| MATH 500 | -0.676 | -0.745 | -0.565 |
| GPQA | -0.312 | -0.117 | -0.096 |
| MMLU College | -0.595 | -0.700 | -0.591 |
| MMLU Pro | -0.590 | -0.555 | -0.425 |
| BBH | -0.365 | -0.386 | -0.300 |

| | Llama 3.1 70B Instruct Correlation Metrics | | |
|---|---|---|---|
| **Dataset** | **Metric Type** | | |
| | Pearson | Spearman | Kendall's Tau |
| MATH 500 | -0.631 | -0.842 | -0.709 |
| GPQA | -0.148 | -0.093 | -0.089 |
| MMLU College | -0.551 | -0.862 | -0.769 |
| MMLU Pro | -0.446 | -0.693 | -0.585 |
| BBH | -0.268 | -0.594 | -0.474 |

Table 9: **Performance of WEAVER with Different Clusters Counts**

| | Clusters for WEAVER Dataset | | | | |
|---|---|---|---|---|---|
| **Dataset** | **Cluster Count** | | | | |
| | 1 | 2 | 3 | 4 | 5 |
| MATH 500 | **93.4** | 87.6 | 83.8 | 82.8 | 81.2 |
| GPQA | **66.4** | **66.4** | **66.4** | **66.4** | **66.4** |
| MMLU College | **94.9** | 91.7 | 90.1 | 89.6 | 89.8 |
| MMLU Pro | 88.4 | **90.2** | 87.1 | 84.6 | 79.8 |
| **Average** | **85.8** | 84.0 | 81.9 | 80.9 | 79.3 |

## C Experiments

### C.1 Models and Datasets

**Benchmarks**: We evaluate our models with several benchmarks for instruction-following, reasoning, mathematics, and coding: MATH500 [28], GPQA [62], MMLU [28], MMLU Pro [82], and BBH [74]. We provide an overview of each dataset in Table 10. For MMLU, we selected the college-level questions for evaluation: biology, chemistry, physics, mathematics, computer science, and medicine. For MMLU Pro, we take a random sample of 500 queries out of the 12K queries available. For BBH, we take four tasks from the dataset of 6K queries available: Penguins in a Table, Causal Judgement, Logical Deduction (Five Objects), and Tracking Shuffled Objects (Five Objects).

**Models**: We evaluate candidate generations using a range of *weak verifiers*—models with imperfect but better-than-random accuracy. Our verification system $\mathcal{V}$ includes two primary classes of weak verifiers: *Reward Models* and *LM Judges*.

- **Reward Models**: A *reward model* (RM) is a trained language model that assigns a scalar score to candidate responses based on how well they align with human preferences [39, 71]. Given a query and a candidate response, the RM outputs a value $V_{ij} \in [0,1]$ representing the estimated quality of candidate $j$ according to criteria such as correctness, helpfulness, and safety.

    – Examples of reward models include those from the RewardBench leaderboard [39], such as INF-ORM [49], QRM Gemma [17], and Skywork Reward [44]. We also include *process reward models* (PRMs), which score the reasoning process itself—emphasizing step-by-step logic and coherence—rather than just the final answer [15, 90].

    – For our study, we selected the top-20 reward models from RewardBench and the top-20 process reward models from Process Reward Bench [71] at both 8B and 70B parameter scales. We exclude any RM or PRM that fails to provide a positive learning signal—i.e., those whose rankings perform no better than random selection on benchmark train sets (Appendix C.1). The diverse training objectives and datasets used for these reward models introduce systematic biases that affect their verification capabilities [39, 71], with different loss functions—including Bradley-Terry loss for pairwise preferences [4], margin loss for fixed score differences [63], and pairwise ranking loss for relative ordering [6].

    – Previous work has noted that it is nontrivial to combine the outputs of reward models and judges as they provide logits and binary decision rules [76, 88]. Instead, we find that we can normalize all RM scores to the range $[0,1]$ using robust percentiles: the bottom 5th percentile is mapped to 0 and the top 95th percentile to 1. For models that provide multiple scoring dimensions (e.g., ArmoRM [79]), we use only their primary output. For our study, we selected the top-20 reward models from RewardBench [39].

- **LM Judges**: An *LM judge* is a language model used to assess the correctness of a candidate response by generating a binary verdict: $V_{ij} \in \{0,1\}$, where 1 indicates that the response is judged correct. These models typically apply chain-of-thought (CoT) reasoning to arrive at their decisions [86]. Each LM judge takes a query and a response as input and outputs a single binary verdict.

    – We use well-known chat models from ChatBotArena [13] as LM judges, which are known for their general-purpose reasoning capabilities. To ensure consistency and determinism, we use greedy decoding (temperature $T = 0$) when generating judgments.

Table 10: **Benchmark Overview**: Evaluation configurations for AlpacaEval 2.0, Arena-Hard-Auto, AIMO, MATH500, GPQA, MMLU, MMLU Pro, and Big-Bench Hard (BBH).

| Benchmark | Dataset Size | Scoring Type | Metric | License |
|---|---|---|---|---|
| MATH500 | 500 | Ground Truth | Pass@1 | Apache 2.0 |
| GPQA | 646 | Ground Truth | Pass@1 | CC BY 4.0 |
| MMLU College | 719 | Ground Truth | Pass@1 | MIT |
| MMLU Pro | 500 | Ground Truth | Pass@1 | MIT |

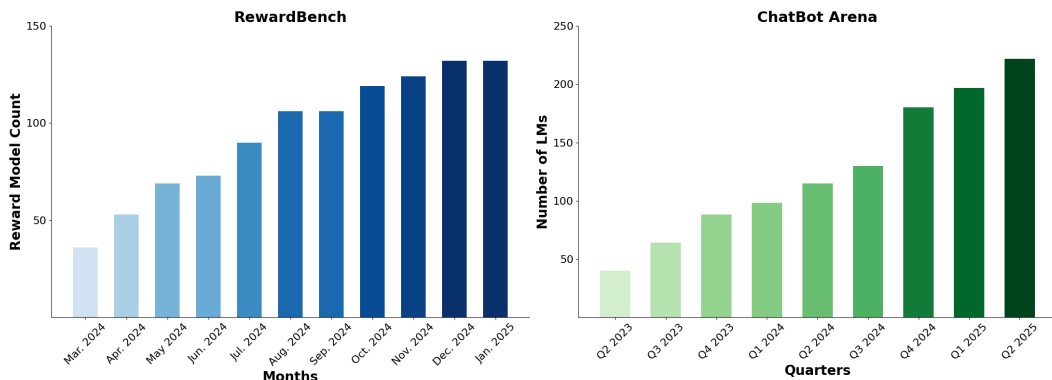

Figure 7: **Growth of Open-Source RMs and LMs**: As more and more RMs and LM judges become available, the need for better selection and utilization strategies for these models at test-time continues to grow.

Table 11: **Distribution of Generation Accuracy across Datasets for Llama 3.1 8B Instruct Model**

**Llama 3.1 8B Instruct**
**Distribution of Positive/Negative Generations**

| Dataset | Percentage of Total Dataset | | | | | | | | | | P/N Ratio |
|---|---|---|---|---|---|---|---|---|---|---|---|
| | 0.0-0.1 | 0.1-0.2 | 0.2-0.3 | 0.3-0.4 | 0.4-0.5 | 0.5-0.6 | 0.6-0.7 | 0.7-0.8 | 0.8-0.9 | 0.9-1.0 | |
| AIMO | 72.2% | 7.8% | 3.3% | 7.8% | 2.2% | 3.3% | 2.2% | 0.0% | 1.1% | 0.0% | 10.3% |
| MATH 500 | 12.2% | 11.4% | 11.6% | 8.2% | 6.4% | 8.0% | 8.4% | 6.60% | 11.2% | 15.0% | 49.9% |
| GPQA | 28.5% | 21.2% | 16.1% | 7.4% | 7.3% | 5.3% | 3.7% | 3.3% | 2.9% | 2.6% | 28.3% |
| MMLU College | 7.8% | 7.6% | 7.1% | 6.5% | 6.7% | 5.3% | 7.2% | 6.3% | 8.6% | 22.9% | 64.1% |
| MMLU Pro | 22.4% | 9.8% | 10.6% | 5.4% | 6.4% | 7.2% | 4.0% | 7.6% | 7.6% | 15.2% | 46.6% |
| BBH | 3.2% | 6.7% | 10.3% | 8.3% | 10.3% | 14.8% | 11.6% | 10.7% | 10.4% | 12.1% | 56.9% |

Table 12: **Distribution of Generation Accuracy across Datasets for Llama 3.1 70B Instruct Model**

**Llama 3.1 70B Instruct**
**Distribution of Positive/Negative Generations**

| Dataset | Percentage of Total Dataset | | | | | | | | | | P/N Ratio |
|---|---|---|---|---|---|---|---|---|---|---|---|
| | 0.0-0.1 | 0.1-0.2 | 0.2-0.3 | 0.3-0.4 | 0.4-0.5 | 0.5-0.6 | 0.6-0.7 | 0.7-0.8 | 0.8-0.9 | 0.9-1.0 | |
| MATH 500 | 7.0% | 4.2% | 3.8% | 2.0% | 2.2% | 3.8% | 4.2% | 4.0% | 7.8% | 61.0% | 78.0% |
| GPQA | 36.8% | 5.6% | 5.1% | 3.7% | 6.2% | 4.3% | 4.5% | 4.8% | 8.0% | 20.9% | 42.9% |
| MMLU College | 8.1% | 3.2% | 1.8% | 2.2% | 1.5% | 1.7% | 1.8% | 3.2% | 2.4% | 74.1% | 82.6% |
| MMLU Pro | 16.4% | 4.2% | 1.8% | 3.4% | 3.0% | 3.0% | 3.2% | 4.8% | 6.0% | 54.2% | 69.9% |

Table 14: **WEAVER Verifier Accuracies and Covariances**

| Approach | Model Size | Benchmarks | | | |
|---|---|---|---|---|---|
| | | MATH500 | GPQA | MMLU Pro | **Average** |
| Verifier Accuracy Range | 70B | 32.3% | 43.6% | 36.7% | 37.5% |
| Verifier Covariance | 70B | 0.0284 | 0.0341 | 0.0342 | 0.0322 |
| Verifier Accuracy Range | 8B | 25.6% | 31.4% | 32.1% | 29.7% |
| Verifier Covariance | 8B | 0.0199 | 0.0337 | 0.0240 | 0.0259 |

Table 13: **Models Tested for WEAVER.**

| | Model | Source Code | Parameter Count | License | Loss Function |
|---|---|---|---|---|---|
| **LM Judges** | Llama-3.1-70B-Instruct | Open-Source | 70B | Llama 3.1 Community | Cross-Entropy Loss |
| | Llama-3.1-405B-Instruct | Open-Source | 405B | Llama 3.1 Community | Cross-Entropy Loss |
| | Llama-3.3-70B-Instruct | Open-Source | 70B | Llama 3.1 Community | Cross-Entropy Loss |
| | Meta-Llama-3.1-405B-Instruct-quantized.w8a16 | Open-Source | 405B | Llama 3.1 Community | Cross-Entropy Loss |
| | DeepSeek LLM 67B Chat | Open-Source | 67B | DeepSeek License | Cross-Entropy Loss |
| | DeepSeekLlama70B | Open-Source | 70B | DeepSeek License | Cross-Entropy Loss |
| | DeepSeekQwen32B | Open-Source | 32B | DeepSeek License | Cross-Entropy Loss |
| | DeepSeekLlama8B | Open-Source | 8B | DeepSeek License | Cross-Entropy Loss |
| | DeepSeekQwen7B | Open-Source | 7B | DeepSeek License | Cross-Entropy Loss |
| | Qwen2 72B Instruct | Open-Source | 72B | Tongyi Qianwen | Cross-Entropy Loss |
| | Qwen2.5-72B-Instruct | Open-Source | 72B | Tongyi Qianwen | Cross-Entropy Loss |
| | Qwen/Qwen2.5-72B-Instruct | Open-Source | 72B | Tongyi Qianwen | Cross-Entropy Loss |
| | QwQ-32B | Open-Source | 32B | Apache 2.0 | Cross-Entropy Loss |
| | Qwen1.5 110B Chat | Open-Source | 110B | Tongyi Qianwen | Cross-Entropy Loss |
| | Qwen1.5 72B Chat | Open-Source | 72B | Tongyi Qianwen | Cross-Entropy Loss |
| | Qwen-2.5-7B-Instruct | Open-Source | 7B | Tongyi Qianwen | Cross-Entropy Loss |
| | Qwen-2.5-Math-7B-Instruct | Open-Source | 7B | Tongyi Qianwen | Cross-Entropy Loss |
| | Mixtral 8x22B v0.1 | Open-Source | 176B | Apache 2.0 | Cross-Entropy Loss |
| | Mixtral-8x22B-Instruct-v0.1 | Open-Source | 176B | Apache 2.0 | Cross-Entropy Loss |
| | WizardLM 8x22B | Open-Source | 176B | Apache 2.0 | Cross-Entropy Loss |
| | WizardLM-2-8x22B | Open-Source | 176B | Apache 2.0 | Cross-Entropy Loss |
| | dbrx-instruct | Open-Source | 132B | Databricks Open Model | Cross-Entropy Loss |
| | SkyT1 | Open-Source | 32B | Apache 2.0 | Cross-Entropy Loss |
| **RMs (8B and below)** | GRM-Llama3-8B-rewardmodel-ft | Open-Source | 8B | MIT | Pairwise Ranking Loss |
| | GRM-Llama3.2-3B-rewardmodel-ft | Open-Source | 3B | Apache 2.0 | Pairwise Ranking Loss |
| | GRM-Gemma2-2B-rewardmodel-ft | Open-Source | 2B | Apache 2.0 | Pairwise Ranking Loss |
| | Skywork-Reward-Llama-3.1-8B-v0.2 | Open-Source | 8B | Skywork License | Pairwise Ranking Loss |
| | QRM-Llama3.1-8B-v2 | Open-Source | 8B | MIT | Quantile Regression Loss |
| | URM-LLaMa-3.1-8B | Open-Source | 8B | Skywork License | Uncertainty-Aware Loss |
| | GPM-Llama-3.1-8B | Open-Source | 8B | MIT | Pairwise Ranking Loss |
| | Llama-3-OffsetBias-RM-8B | Open-Source | 8B | Llama 3.1 Community | Pairwise Ranking Loss |
| | ArmoRM-Llama3-8B-v0.1 | Open-Source | 8B | Llama 3.1 Community | Pairwise Ranking Loss |
| | Qwen2.5-Math-PRM-7B | Open-Source | 7B | Tongyi Qianwen | Cross-Entropy Loss |
| | EurusPRM-Stage1 | Open-Source | 7B | Apache 2.0 | Cross-Entropy Loss |
| | EurusPRM-Stage2 | Open-Source | 7B | Apache 2.0 | Cross-Entropy Loss |
| | internlm2-7b-reward | Open-Source | 7B | Apache 2.0 | Pairwise Ranking Loss |
| | Decision-Tree-Reward-Llama-3.1-8B | Open-Source | 8B | Skywork License | Decision Tree Loss |
| **RMs (27B–72B)** | Skywork-Reward-Gemma-2-27B-v0.2 | Open-Source | 27B | Skywork License | Pairwise Ranking Loss |
| | QRM-Gemma-2-27B | Open-Source | 27B | MIT | Quantile Regression Loss |
| | INF-ORM-Llama3.1-70B | Open-Source | 70B | Custom License | Binary Cross-Entropy Loss |
| | Qwen2.5-Math-RM-72B | Open-Source | 72B | Tongyi Qianwen | Cross-Entropy Loss |
| | Qwen2.5-Math-PRM-72B | Open-Source | 72B | Tongyi Qianwen | Cross-Entropy Loss |
| | internlm2-20b-reward | Open-Source | 20B | Apache 2.0 | Pairwise Ranking Loss |
| | Decision-Tree-Reward-Gemma-2-27B | Open-Source | 27B | Skywork License | Pairwise Ranking Loss |

## C.2 Verification Baselines

### C.2.1 Verifier-Free Approaches

**First Sample (Pass@1)**: This baseline uses only the first generated response without any verification or selection mechanism. It represents the standard approach where models generate a single response and provides a lower bound for performance comparison. This method does not scale test-time compute or employ verification.

**Majority Voting**: A verifier-free approach that generates multiple candidate responses and selects the most frequent final answer across all responses [5, 11, 70]. This method leverages repeated sampling but does not use verification models to assess response quality. Instead, it relies on the assumption that correct answers will appear more frequently than incorrect ones across multiple generations.

### C.2.2 Alternative Verification Strategies

**Naive Unweighted Aggregation**: We consider three oracle configurations using the top-1, top-5, and top-10 verifiers (ranked by their agreement with ground-truth labels). Across all datasets, these oracle ensembles substantially outperform baselines. On average, the best-performing unweighted ensembles exceed first-sample performance by 20.3% and outperform majority voting by 15.0% (see Figure 2). For more difficult benchmarks such as GPQA and MMLU Pro, the top-5 and top-10 ensembles consistently outperform top-1, suggesting that verifier diversity is especially beneficial on challenging examples. However, these oracle ensembles rely on access to ground truth to rank verifiers, limiting their use in practice and motivating the need for learned, unsupervised weighting.

**Naive Bayes**: We implement a Naive Bayes classifier that models the probability of response correctness given verifier scores: $P(y_{ij} = 1|s_{ij1}, ..., s_{ijm}) = \frac{P(s_{ij1}, ..., s_{ijm}|y_{ij}=1)P(y_{ij}=1)}{P(s_{ij1}, ..., s_{ijm})}$. Under the conditional independence assumption, this factorizes as $P(s_{ij1}, ..., s_{ijm}|y_{ij} = 1) = \prod_{k=1}^{m} P(s_{ijk}|y_{ij} = 1)$. We estimate the parameters using labeled data from the development set. This approach provides a probabilistic framework for aggregating verifier outputs but requires labeled data for parameter estimation.

**Logistic Regression**: We train a logistic regression classifier where the input features are the verifier scores $[s_{ij1}, ..., s_{ijm}]$ and the output is the correctness of each response: $P(y_{ij} = 1|s_{ij}) = \sigma(\mathbf{w}^T s_{ij} + b)$, where $\sigma$ is the sigmoid function. The weights $\mathbf{w}$ and bias $b$ are learned using labeled training data. This supervised approach can capture more complex relationships between verifier outputs than naive averaging but requires substantial labeled data for effective training.

**Multi-Agent Verification (MAV) [41]**: This approach combines multiple "Aspect Verifiers" (AVs) - off-the-shelf LLMs prompted to verify specific aspects of candidate outputs through binary True/False approvals. Unlike reward models, AVs require no additional training and can be easily combined through voting mechanisms. The MAV framework uses BoN-MAV (Best-of-N with Multi-Agent Verification), which: **(1)** samples $n$ candidate outputs from a generator LLM, **(2)** collects binary approvals from multiple aspect verifiers that vary across three dimensions (base LLM, aspect to verify, and verification strategy), and **(3)** selects the output with the most approvals. In our implementation, we use Llama 3.3 70B Instruct as the judge model rather than Gemini 1.5 Flash/Pro as used in the original paper.

**Self-Verification [93]**: This method implements a sophisticated sampling-based search approach where models verify their own responses through detailed natural language analysis. The approach goes beyond simple self-critique by using structured verification prompts that: **(1)** rewrite candidate responses in rigorous mathematical theorem-lemma-proof format, **(2)** systematically scan for errors through step-by-step analysis, and **(3)** compare responses to localize potential mistakes. The method leverages two key principles: comparing across responses provides signals about error locations (since models struggle with error recall but can identify errors when given their locations), and different output styles are optimal for different tasks (chain-of-thought for generation, rigorous mathematical format for verification). This approach differs from naive self-verification by using structured, multi-step verification protocols rather than simple correctness judgments.

Table 15: **Logistic Regression and Naive Bayes Performances across Datasets and Dev Set Sizes**

| Dataset | Approach | Model Size | Dev Set Size | | | | |
|---|---|---|---|---|---|---|---|
| | | | 0.01 | 0.05 | 0.2 | 0.5 | 1.0 |
| MATH-500 | Logistic Regression | 70B | 70.5% | 74.7% | 81.4% | 93.1% | 97.2% |
| | Naive Bayes | 70B | 67.4% | 78.1% | 85.0% | 89.2% | 92.2% |
| GPQA Diamond | Logistic Regression | 70B | 55.9% | 59.4% | 69.8% | 71.4% | 72.9% |
| | Naive Bayes | 70B | 47.2% | 49.2% | 57.6% | 62.1% | 64.3% |
| MMLU Pro | Logistic Regression | 70B | 72.1% | 81.0% | 84.6% | 86.0% | 92.0% |
| | Naive Bayes | 70B | 60.2% | 73.1% | 73.1% | 78.6% | 78.6% |

## C.3 Scaling Trends of WEAVER

Scaling laws describe how performance metrics such as accuracy, sample efficiency or compute cost change as we scale controllable resources, i.e. the number of trials $K$, model capacity. [33] showed that, for fixed-parameter Transformer language models, the cross-entropy loss decreases as a power-law in both model size and data. This framework has since been extended to explore optimal tradeoffs between model and data scaling [30], as well as inference-time scaling with multiple samples [5, 12].

First, we establish the power law scaling of the Pass@K rate. Assume the $i$-th problem has an unknown "difficulty" $p_i \in [0,1]$, the probability that one response is correct. With $K$ independent samples, the chance we get at least one correct response is

$$q_i(p_i, K) = 1 - (1-p_i)^K \approx 1 - \exp(-p_i \cdot K) \quad \text{for small } p_i$$

Define the indicator variable:

$$X_i = \begin{cases} 1, & \text{if the i-th query is solved at least once (with probability } q_i), \\ 0, & \text{otherwise,} \end{cases}$$

and let the total number of solved problems be $Y = \sum_{i=1}^{N} X_i$.

The expected coverage (Pass@K) is the expected fraction of problems solved after trying $K$ times per problem:

$$\text{Pass@K} := \mathbb{E}[Y]/N = \frac{1}{N}\sum_{i=1}^{N}(1-(1-p_i)^K)$$

To model population-level variation in problem difficulty, we assume each problem's correctness probability $p_i$ is drawn from a Beta distribution: $p_i \sim \text{Beta}(\alpha, \beta)$. This captures the idea that some problems are easier (high $p_i$) while others are harder (low $p_i$), with the overall distribution controlled by the shape parameters $\alpha$, $\beta$. Then, the fraction of problem that can be solved in $K$ attempts follows,

$$\text{Pass@K} = \mathbb{E}_{p \sim \text{Beta}(\alpha, \beta)}[1 - (1-p)^K]$$
$$= 1 - \mathbb{E}_{p \sim \text{Beta}(\alpha, \beta)}[(1-p)^K] = 1 - \frac{B(\alpha, \beta+K)}{B(\alpha, \beta)} \quad (11)$$

by the definition of the Beta function $B(\cdot,\cdot)$. Taking logarithm:

$$\log\text{Pass@K} = \log\left(1 - \frac{B(\alpha, \beta+K)}{B(\alpha, \beta)}\right) \approx -\frac{B(\alpha, \beta+K)}{B(\alpha, \beta)}$$

by $\log(1-x) \approx -x$ when $x$ is small, which holds for large $K$. Then, expressing the Beta function in terms of the Gamma function leads to:

$$\log\text{Pass@K} \approx -\frac{\Gamma(\beta+K)\Gamma(\alpha+\beta)}{\Gamma(\beta)\Gamma(\alpha+\beta+K)}$$

For large $K$, we can apply Stirling's approximation of the Gamma function $\log\Gamma(x) \approx x\log x - x + \frac{1}{2}\log(2\pi) + \frac{1}{2}\log x$:

$$\log[-\log\text{Pass@K}] = \log\Gamma(\beta+K) + \log\Gamma(\alpha+\beta) - \log\Gamma(\beta) - \log\Gamma(\alpha+\beta+K)$$

$$\approx (\beta+K)\log(\beta+K) - (\alpha+\beta+K)\log(\alpha+\beta+K) + \frac{1}{2}\log\left(\frac{\beta+K}{\alpha+\beta+K}\right)$$

$$\approx (\beta+K)\log K - (\alpha+\beta+K)\log K + \text{const}$$

$$= -\alpha\log K + \log\zeta$$

when we retain the leading term. In turn, the log of the expected coverage follows a power law in $K$, scaling as:

$$\log\text{Pass@K} = -\exp(-\alpha\log K + \log\zeta) = -\zeta K^{-\alpha} \tag{12}$$

**Verifier Success Modeling**   Now suppose we pass the $K$ candidates through a scoring model ("verifier") which selects the top-scoring answer. The verification process succeeds if (i) at least one correct answer was generated and (ii) the verifier ranks a correct answer highest.

$$\text{Selection@1}(K) := \mathbb{P}[\text{top-scoring response is correct}] \tag{13}$$

Assume the verifier assigns scores such that correct responses are drawn from a score distribution $f_1$, and the incorrect responses from a distribution $f_0$. Let $s^{(1)} = \{s_j : y_j = 1\}$ and $s^{(0)} = \{s_j : y_j = 0\}$ denote the scores of correct and incorrect responses, respectively. Then a query is successfully verified if:

$$\text{Selection@1} = \mathbb{P}\left[\max s^{(1)} > \max s^{(0)}\right]$$

Our goal is to compute the probability that the maximum of $c$ i.i.d draws from $f_1$ exceeds the maximum of $K-c$ draws from $f_0$.

To model the correctness of responses, we assume each query $i$ has a latent correctness probability $p_i \sim \text{Beta}(\alpha,\beta)$, reflecting query-specific difficulty. Given $p_i$, each of the $K$ responses is sampled independently as:

$$y_{ij} \sim \text{Bernoulli}(p_i), \quad j = 1,...,K$$

This implies the number of correct responses follows a Binomial distribution:

$$C_i = \sum_{j=1}^{K} y_{ij} \sim \text{Binomial}(K,p_i)$$

assuming (1) conditional independence of responses given $p_i$, (2) identical correctness probabilities within a query, and (3) a fixed number of responses $K$.

Because the correctness probability $p$ varies across queries, the dataset-level Selection@1 curve requires marginalizing over $p$:

$$\text{Selection@1}(K) = \mathbb{E}_{p \sim \text{Beta}(\alpha,\beta)}[\text{Selection@1}(K \,|\, p)]$$

Combined with the need to model max comparisons over verifier scores, it renders the exact calculation of Selection@1 analytically intractable.

To enable tractable, smooth modeling of Selection@1, we introduce the following parametric form:

$$\text{Selection@1}(K) \approx \exp(-\zeta K^{-\alpha}) \cdot \left(1 - (1-\pi)^{K^\gamma}\right) \tag{14}$$

- The **coverage term** $\exp(-\zeta K^{-\alpha})$ approximates the probability that at least one correct response is generated.
- The **verification term** $1 - (1-\pi)^{K^\gamma}$ approximates the chance that the top-scoring response is correct, given that at least one correct response exists. The parameter $\gamma$ controls whether verifier performance improves sublinearly or superlinearly with $K$. The parameter $\pi$ represents the effective per-response probability that a correct response is successfully selected by the verifier, conditioned on the response being correct and included in the candidate set.

To obtain practical scaling trends, we fit parametric models in Eq. (14) to the empirical averages computed from 5 independent runs for each value of $K$, across each dataset and verification strategy. Specifically, we use the L-BFGS-B algorithm to optimize a smooth approximation following [29]. To ensure numerical stability and robustness to outliers or heavy-tailed noise in the observed selection accuracies, we minimize the Huber loss between the predicted values and the empirical means. The Huber loss behaves quadratically for small residuals and linearly for large ones, making it less sensitive to outliers than mean squared error (MSE) while maintaining smooth differentiability for gradient-based optimization. It is defined as,

$$L_\delta(r) = \begin{cases} \frac{1}{2}r^2 & \text{if } |r| \leq \delta \\ \delta\left(|r| - \frac{1}{2}\delta\right) & \text{otherwise} \end{cases}$$

where $\delta > 0$ is a tunable threshold that controls the transition between the two regimes. We search over $\delta \in \{0.01, 0.05, 0.1, 0.25, 0.5\}$ to select the value that yields the best fit.

Additionally, we introduce floor and ceiling parameters to bound the predicted values and model saturation behavior. The floor accounts for the irreducible failure rate even at high $K$, while the ceiling models the upper bound on achievable performance (e.g., due to imperfect verifiers or ambiguous problems). The final fitted form is:

$$\text{Selection@1}(K) \approx \text{floor} + (\text{ceil} - \text{floor}) \cdot \exp(-\zeta K^{-\alpha}) \cdot \left(1 - (1-\pi)^{K^\gamma}\right) \tag{15}$$

We can use an unbiased estimator to evaluate best-of-$k$ selection accuracy when a fixed verifier is used to rank responses, as described in [69]. However, in the case of WEAVER, the development set constitutes 1% of the data and is itself selected based on the value of $K$. In turn, the ranking of responses is no longer independent of $K$, introducing bias into the best-of-$k$ estimate. As a result, we instead rely on Monte Carlo estimates to approximate best-of-$k$ performance, sampling $k$ responses multiple times and computing the average accuracy of the top-ranked output under the $K$-dependent verifier. We use an unbiased estimator for coverage, as described in [12].

Fig. 8 and Fig. 9 along with Table 16 illustrate how the different verification strategies scale with the number of generations and the fit to the parametric form in Eq. (14). Each method exhibits characteristic scaling behavior that aligns with Eq. (14). WEAVER demonstrates improved performance over naive ensembles and majority voting. The fitted parameters in Table 16 quantitatively capture these trends across datasets, providing evidence that the parametric from in Eq. (15) closely model empirical outcomes. Fig. 10 and Fig. 11 along with Table 17 illustrate the predictive performance of the parametric form in Eq. (15), showing that models fit on subsets of $K$ can extrapolate to unseen values of $K$.

Table 16: Fitted parameters for Scaling Trends in Fig. 8 and Fig. 9.

| Dataset | Approach | Equation | floor | ceil | $\zeta$ | $\alpha$ | $\pi$ | $\gamma$ | R2 fit | MSE fit | $\delta$ |
|---|---|---|---|---|---|---|---|---|---|---|---|
| GPQA-v2-Diamond (70B) | Pass@K | $y = \text{floor} + (\text{ceil} - \text{floor}) \cdot \exp(-\zeta \cdot K^{-\zeta})$ | 0.0000 | 0.9429 | 0.7603 | 0.3475 | X | X | 0.9999 | 0.0000 | 0.5000 |
| GPQA-v2-Diamond (70B) | Weaver | $y = \text{floor} + (\text{ceil} - \text{floor}) \cdot \exp(-\zeta \cdot K^{-\zeta})(1 - (1-\pi)^{K^\gamma})$ | 0.3958 | 0.6728 | 0.7320 | 1.5865 | 0.3250 | 0.5053 | 0.9994 | 0.0000 | 0.1000 |
| GPQA-v2-Diamond (70B) | Majority1@K | $y = \text{floor} + (\text{ceil} - \text{floor}) \cdot \exp(-\zeta \cdot K^{-\zeta})(1 - (1-\pi)^{K^\gamma})$ | 0.4283 | 0.4710 | 0.0499 | 1.0000 | 0.1217 | 1.0091 | 0.8634 | 0.0000 | 0.0100 |
| GPQA-v2-Diamond (70B) | Naive Ensemble | $y = \text{floor} + (\text{ceil} - \text{floor}) \cdot \exp(-\zeta \cdot K^{-\zeta})(1 - (1-\pi)^{K^\gamma})$ | 0.3921 | 0.6071 | 0.6553 | 1.9147 | 0.4224 | 0.5000 | 0.9975 | 0.0000 | 0.2500 |
| MATH-500-v2 (70B) | Pass@K | $y = \text{floor} + (\text{ceil} - \text{floor}) \cdot \exp(-\zeta \cdot K^{-\zeta})$ | 0.6262 | 1.0000 | 0.8394 | 0.6427 | X | X | 0.9994 | 0.0000 | 0.2500 |
| MATH-500-v2 (70B) | Weaver | $y = \text{floor} + (\text{ceil} - \text{floor}) \cdot \exp(-\zeta \cdot K^{-\zeta})(1 - (1-\pi)^{K^\gamma})$ | 0.7870 | 0.9371 | 3.3908 | 3.0000 | 0.2869 | 0.5000 | 0.9958 | 0.0000 | 0.1000 |
| MATH-500-v2 (70B) | Majority1@K | $y = \text{floor} + (\text{ceil} - \text{floor}) \cdot \exp(-\zeta \cdot K^{-\zeta})(1 - (1-\pi)^{K^\gamma})$ | 0.7747 | 0.8238 | 10.0000 | 2.1433 | 0.0885 | 2.4951 | 0.8655 | 0.0001 | 0.1000 |
| MATH-500-v2 (70B) | Naive Ensemble | $y = \text{floor} + (\text{ceil} - \text{floor}) \cdot \exp(-\zeta \cdot K^{-\zeta})(1 - (1-\pi)^{K^\gamma})$ | 0.7883 | 0.9282 | 4.0033 | 3.0000 | 0.2573 | 0.5000 | 0.9961 | 0.0000 | 0.0100 |
| MMLU-Pro-v2 (70B) | Pass@K | $y = \text{floor} + (\text{ceil} - \text{floor}) \cdot \exp(-\zeta \cdot K^{-\zeta})$ | 0.0000 | 0.9828 | 0.3303 | 0.3465 | X | X | 0.9967 | 0.0000 | 0.2500 |
| MMLU-Pro-v2 (70B) | Weaver | $y = \text{floor} + (\text{ceil} - \text{floor}) \cdot \exp(-\zeta \cdot K^{-\zeta})(1 - (1-\pi)^{K^\gamma})$ | 0.6912 | 0.9148 | 1.5284 | 3.0000 | 0.2764 | 0.5000 | 0.9987 | 0.0000 | 0.1000 |
| MMLU-Pro-v2 (70B) | Majority1@K | $y = \text{floor} + (\text{ceil} - \text{floor}) \cdot \exp(-\zeta \cdot K^{-\zeta})(1 - (1-\pi)^{K^\gamma})$ | 0.6933 | 0.7399 | 0.0498 | 1.0001 | 0.1531 | 1.0123 | 0.9451 | 0.0000 | 0.0100 |
| MMLU-Pro-v2 (70B) | Naive Ensemble | $y = \text{floor} + (\text{ceil} - \text{floor}) \cdot \exp(-\zeta \cdot K^{-\zeta})(1 - (1-\pi)^{K^\gamma})$ | 0.6969 | 0.8874 | 1.7834 | 3.0000 | 0.2403 | 0.5000 | 0.9944 | 0.0000 | 0.2500 |
| MMLU-College-v2 (70B) | Pass@K | $y = \text{floor} + (\text{ceil} - \text{floor}) \cdot \exp(-\zeta \cdot K^{-\zeta})$ | 0.5924 | 0.9744 | 0.5071 | 0.5682 | X | X | 0.9982 | 0.0000 | 0.0100 |
| MMLU-College-v2 (70B) | Weaver | $y = \text{floor} + (\text{ceil} - \text{floor}) \cdot \exp(-\zeta \cdot K^{-\zeta})(1 - (1-\pi)^{K^\gamma})$ | 0.8234 | 0.9477 | 4.4129 | 3.0000 | 0.3622 | 0.5000 | 0.9987 | 0.0000 | 0.0100 |
| MMLU-College-v2 (70B) | Majority1@K | $y = \text{floor} + (\text{ceil} - \text{floor}) \cdot \exp(-\zeta \cdot K^{-\zeta})(1 - (1-\pi)^{K^\gamma})$ | 0.8197 | 0.8412 | 0.0498 | 1.0001 | 0.2057 | 1.0173 | 0.8912 | 0.0000 | 0.0100 |
| MMLU-College-v2 (70B) | Naive Ensemble | $y = \text{floor} + (\text{ceil} - \text{floor}) \cdot \exp(-\zeta \cdot K^{-\zeta})(1 - (1-\pi)^{K^\gamma})$ | 0.8235 | 0.9266 | 3.8766 | 3.0000 | 0.3012 | 0.5000 | 0.9925 | 0.0000 | 0.0500 |
| GPQA-v2-Diamond (8B) | Pass@K | $y = \text{floor} + (\text{ceil} - \text{floor}) \cdot \exp(-\zeta \cdot K^{-\zeta})$ | 0.2262 | 0.9926 | 2.5454 | 0.8474 | X | X | 0.9996 | 0.0000 | 0.1000 |
| GPQA-v2-Diamond (8B) | Weaver | $y = \text{floor} + (\text{ceil} - \text{floor}) \cdot \exp(-\zeta \cdot K^{-\zeta})(1 - (1-\pi)^{K^\gamma})$ | 0.2463 | 0.4549 | 0.0534 | 1.0020 | 0.1953 | 0.7089 | 0.9948 | 0.0000 | 0.0500 |
| GPQA-v2-Diamond (8B) | Majority1@K | $y = \text{floor} + (\text{ceil} - \text{floor}) \cdot \exp(-\zeta \cdot K^{-\zeta})(1 - (1-\pi)^{K^\gamma})$ | 0.2783 | 0.3029 | 1.2431 | 0.6756 | 0.0630 | 2.5000 | 0.5929 | 0.0000 | 0.0500 |
| GPQA-v2-Diamond (8B) | Naive Ensemble | $y = \text{floor} + (\text{ceil} - \text{floor}) \cdot \exp(-\zeta \cdot K^{-\zeta})(1 - (1-\pi)^{K^\gamma})$ | 0.2359 | 0.3727 | 0.0701 | 1.0016 | 0.3871 | 0.5000 | 0.9408 | 0.0001 | 0.0100 |
| MATH-500-v2 (8B) | Pass@K | $y = \text{floor} + (\text{ceil} - \text{floor}) \cdot \exp(-\zeta \cdot K^{-\zeta})$ | 0.4099 | 1.0000 | 1.7182 | 0.8949 | X | X | 0.9976 | 0.0001 | 0.0100 |
| MATH-500-v2 (8B) | Weaver | $y = \text{floor} + (\text{ceil} - \text{floor}) \cdot \exp(-\zeta \cdot K^{-\zeta})(1 - (1-\pi)^{K^\gamma})$ | 0.4110 | 0.7440 | 0.0785 | 1.0033 | 0.3333 | 0.5036 | 0.9984 | 0.0000 | 0.0100 |
| MATH-500-v2 (8B) | Majority1@K | $y = \text{floor} + (\text{ceil} - \text{floor}) \cdot \exp(-\zeta \cdot K^{-\zeta})(1 - (1-\pi)^{K^\gamma})$ | 0.5058 | 0.7038 | 5.1187 | 1.0175 | 0.0666 | 2.5000 | 0.9964 | 0.0000 | 0.0500 |
| MATH-500-v2 (8B) | Naive Ensemble | $y = \text{floor} + (\text{ceil} - \text{floor}) \cdot \exp(-\zeta \cdot K^{-\zeta})(1 - (1-\pi)^{K^\gamma})$ | 0.5071 | 0.7508 | 1.8068 | 3.0000 | 0.2890 | 0.5000 | 0.9796 | 0.0001 | 0.0100 |
| MMLU-Pro-v2 (8B) | Pass@K | $y = \text{floor} + (\text{ceil} - \text{floor}) \cdot \exp(-\zeta \cdot K^{-\zeta})$ | 0.3906 | 1.0000 | 1.9045 | 0.7590 | X | X | 0.9991 | 0.0000 | 0.0100 |
| MMLU-Pro-v2 (8B) | Weaver | $y = \text{floor} + (\text{ceil} - \text{floor}) \cdot \exp(-\zeta \cdot K^{-\zeta})(1 - (1-\pi)^{K^\gamma})$ | 0.4764 | 0.6846 | 2.7876 | 3.0000 | 0.2136 | 0.6141 | 0.9985 | 0.0000 | 0.2500 |
| MMLU-Pro-v2 (8B) | Majority1@K | $y = \text{floor} + (\text{ceil} - \text{floor}) \cdot \exp(-\zeta \cdot K^{-\zeta})(1 - (1-\pi)^{K^\gamma})$ | 0.4439 | 0.5662 | 0.1136 | 1.0001 | 0.2787 | 0.6659 | 0.9084 | 0.0001 | 0.0100 |
| MMLU-Pro-v2 (8B) | Naive Ensemble | $y = \text{floor} + (\text{ceil} - \text{floor}) \cdot \exp(-\zeta \cdot K^{-\zeta})(1 - (1-\pi)^{K^\gamma})$ | 0.4771 | 0.7025 | 2.8863 | 3.0000 | 0.1728 | 0.5181 | 0.9986 | 0.0000 | 0.1000 |
| MMLU-College-v2 (8B) | Pass@K | $y = \text{floor} + (\text{ceil} - \text{floor}) \cdot \exp(-\zeta \cdot K^{-\zeta})$ | 0.4316 | 0.9924 | 0.9887 | 0.9123 | X | X | 0.9994 | 0.0000 | 0.5000 |
| MMLU-College-v2 (8B) | Weaver | $y = \text{floor} + (\text{ceil} - \text{floor}) \cdot \exp(-\zeta \cdot K^{-\zeta})(1 - (1-\pi)^{K^\gamma})$ | 0.6226 | 0.8494 | 1.2646 | 3.0000 | 0.3346 | 0.5000 | 0.9958 | 0.0000 | 0.2500 |
| MMLU-College-v2 (8B) | Majority1@K | $y = \text{floor} + (\text{ceil} - \text{floor}) \cdot \exp(-\zeta \cdot K^{-\zeta})(1 - (1-\pi)^{K^\gamma})$ | 0.6359 | 0.7368 | 1.0929 | 0.5130 | 0.0576 | 2.5000 | 0.9949 | 0.0000 | 0.0500 |
| MMLU-College-v2 (8B) | Naive Ensemble | $y = \text{floor} + (\text{ceil} - \text{floor}) \cdot \exp(-\zeta \cdot K^{-\zeta})(1 - (1-\pi)^{K^\gamma})$ | 0.6283 | 0.8085 | 1.4279 | 3.0000 | 0.3914 | 0.5000 | 0.9845 | 0.0000 | 0.0100 |

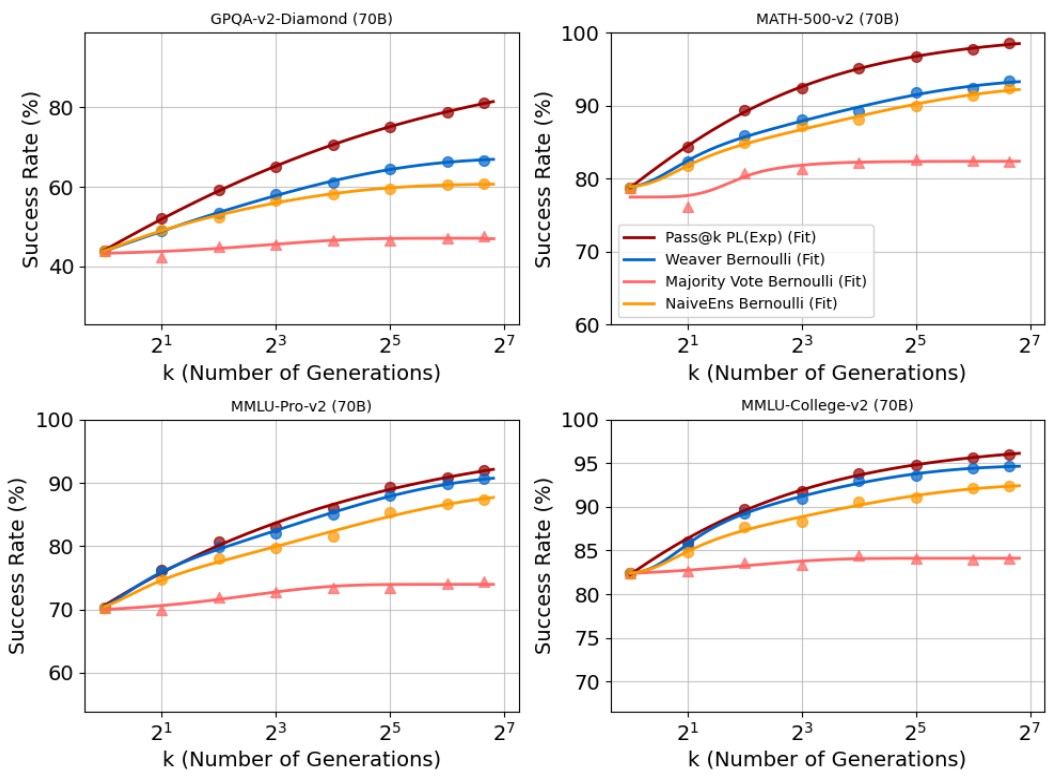

Figure 8: **WEAVER Scaling trend fit for 70B models**

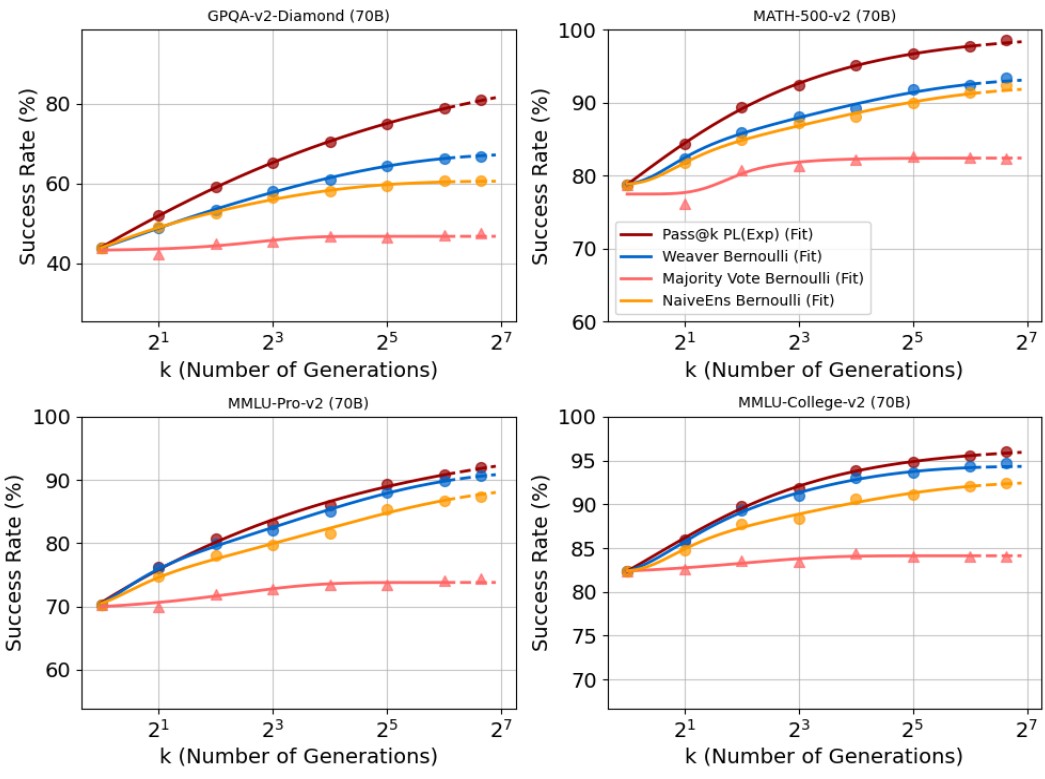

Figure 10: **WEAVER Scaling trend predicted for 70B models**

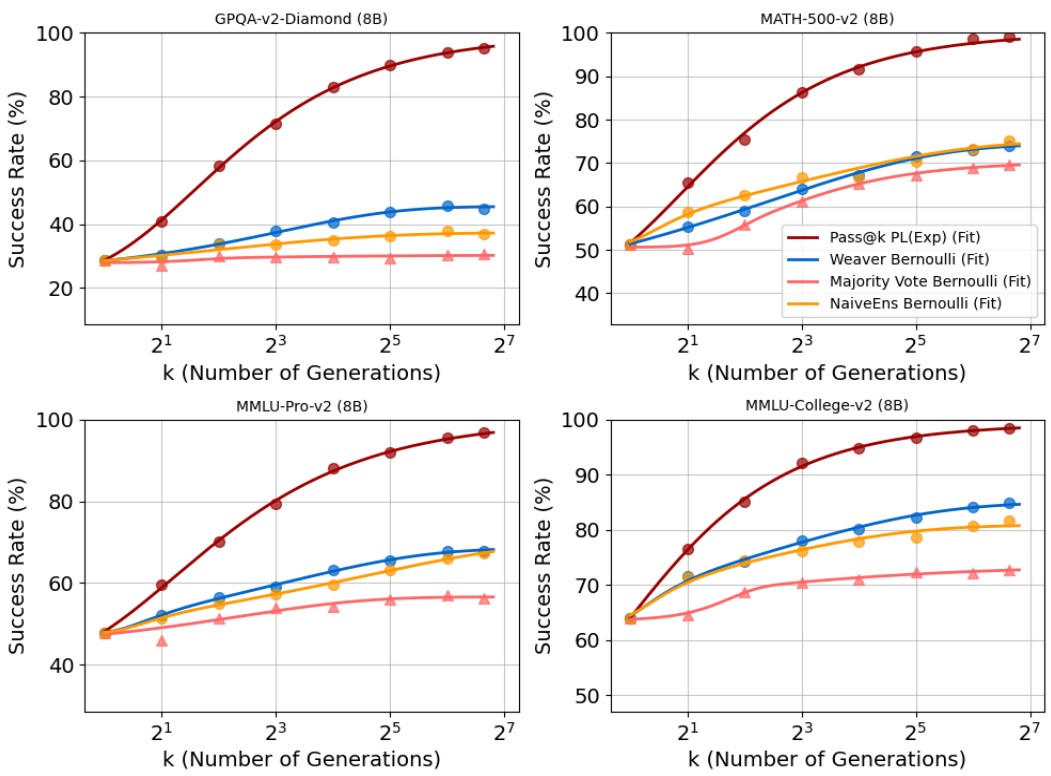

Figure 9: **WEAVER Scaling trend fit for 8B models.**

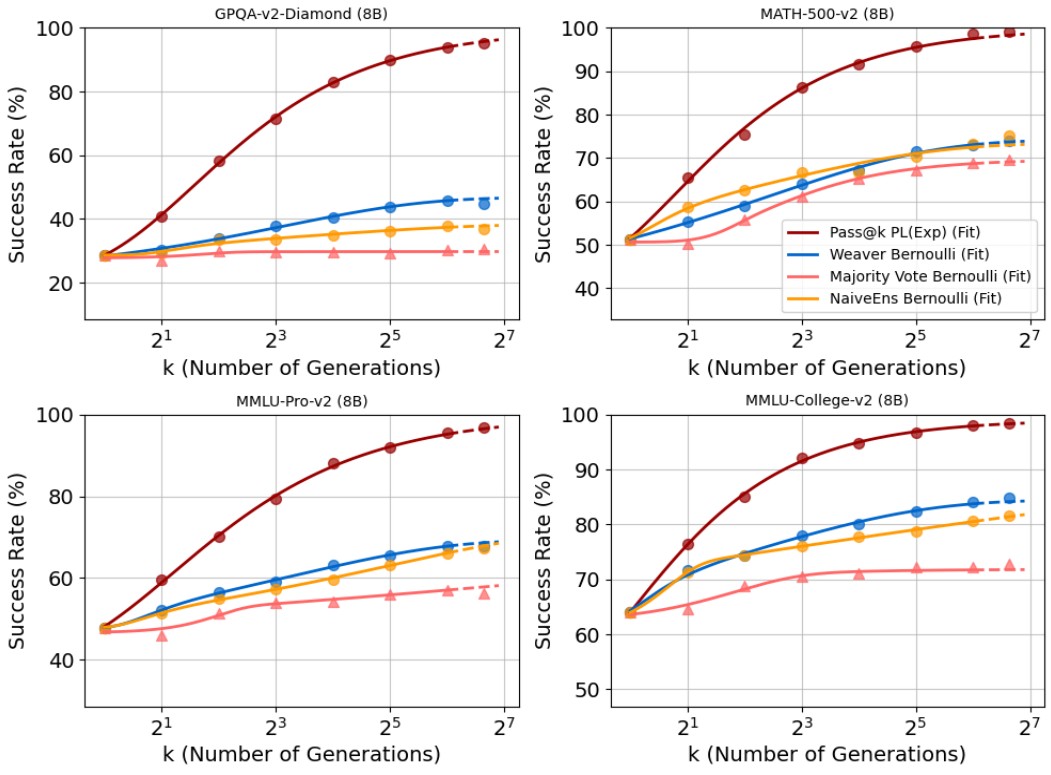

Figure 11: **WEAVER Scaling trend predicted for 8B models**

Table 17: Fitted parameters for Scaling Trends with 90% of data in Fig. 10 and Fig. 11.

| Dataset | Approach | Equation | floor | ceil | $\zeta$ | $\alpha$ | $\pi$ | $\gamma$ | R2 fit | MSE fit | MSE pred | $\delta$ |
|---|---|---|---|---|---|---|---|---|---|---|---|---|
| GPQA-v2-Diamond (70B) | Pass@K | $y=\text{floor}+(\text{ceil}-\text{floor})\cdot\exp(-\zeta\cdot K^{-}\zeta)$ | 0.0000 | 0.9357 | 0.7534 | 0.3537 | X | X | 0.9999 | 0.0000 | 0.0000 | 0.1000 |
| GPQA-v2-Diamond (70B) | Weaver | $y=\text{floor}+(\text{ceil}-\text{floor})\cdot\exp(-\zeta\cdot K^{-}\zeta)(1-(1-\pi)^{K^\gamma})$ | 0.4050 | 0.6756 | 0.9195 | 1.6227 | 0.3163 | 0.5000 | 0.9993 | 0.0000 | 0.0000 | 0.0500 |
| GPQA-v2-Diamond (70B) | Majority1@K | $y=\text{floor}+(\text{ceil}-\text{floor})\cdot\exp(-\zeta\cdot K^{-}\zeta)(1-(1-\pi)^{K^\gamma})$ | 0.4320 | 0.4678 | 0.0707 | 0.9864 | 0.0414 | 1.6541 | 0.8601 | 0.0000 | 0.0001 | 0.0100 |
| GPQA-v2-Diamond (70B) | Naive Ensemble | $y=\text{floor}+(\text{ceil}-\text{floor})\cdot\exp(-\zeta\cdot K^{-}\zeta)(1-(1-\pi)^{K^\gamma})$ | 0.3809 | 0.6061 | 0.5211 | 1.9075 | 0.4360 | 0.5000 | 0.9972 | 0.0000 | 0.0000 | 0.0500 |
| MATH-500-v2 (70B) | Pass@K | $y=\text{floor}+(\text{ceil}-\text{floor})\cdot\exp(-\zeta\cdot K^{-}\zeta)$ | 0.6639 | 0.9952 | 0.9836 | 0.6936 | X | X | 0.9995 | 0.0000 | 0.0000 | 0.0100 |
| MATH-500-v2 (70B) | Weaver | $y=\text{floor}+(\text{ceil}-\text{floor})\cdot\exp(-\zeta\cdot K^{-}\zeta)(1-(1-\pi)^{K^\gamma})$ | 0.7869 | 0.9339 | 3.5000 | 3.0000 | 0.2985 | 0.5000 | 0.9954 | 0.0000 | 0.0000 | 0.0500 |
| MATH-500-v2 (70B) | Majority1@K | $y=\text{floor}+(\text{ceil}-\text{floor})\cdot\exp(-\zeta\cdot K^{-}\zeta)(1-(1-\pi)^{K^\gamma})$ | 0.7747 | 0.8241 | 10.0000 | 2.1272 | 0.0888 | 2.4999 | 0.8560 | 0.0001 | 0.0000 | 0.0500 |
| MATH-500-v2 (70B) | Naive Ensemble | $y=\text{floor}+(\text{ceil}-\text{floor})\cdot\exp(-\zeta\cdot K^{-}\zeta)(1-(1-\pi)^{K^\gamma})$ | 0.7879 | 0.9221 | 4.2081 | 3.0000 | 0.2789 | 0.5000 | 0.9966 | 0.0000 | 0.0000 | 0.1000 |
| MMLU-Pro-v2 (70B) | Pass@K | $y=\text{floor}+(\text{ceil}-\text{floor})\cdot\exp(-\zeta\cdot K^{-}\zeta)$ | 0.0000 | 0.9785 | 0.3263 | 0.3543 | X | X | 0.9959 | 0.0000 | 0.0000 | 0.1000 |
| MMLU-Pro-v2 (70B) | Weaver | $y=\text{floor}+(\text{ceil}-\text{floor})\cdot\exp(-\zeta\cdot K^{-}\zeta)(1-(1-\pi)^{K^\gamma})$ | 0.6912 | 0.9148 | 1.5277 | 3.0000 | 0.2765 | 0.5000 | 0.9984 | 0.0000 | 0.0000 | 0.2500 |
| MMLU-Pro-v2 (70B) | Majority1@K | $y=\text{floor}+(\text{ceil}-\text{floor})\cdot\exp(-\zeta\cdot K^{-}\zeta)(1-(1-\pi)^{K^\gamma})$ | 0.6923 | 0.7379 | 0.0495 | 1.0001 | 0.1711 | 1.0148 | 0.9481 | 0.0000 | 0.0000 | 0.0500 |
| MMLU-Pro-v2 (70B) | Naive Ensemble | $y=\text{floor}+(\text{ceil}-\text{floor})\cdot\exp(-\zeta\cdot K^{-}\zeta)(1-(1-\pi)^{K^\gamma})$ | 0.6979 | 0.8907 | 1.8452 | 3.0000 | 0.2327 | 0.5000 | 0.9931 | 0.0000 | 0.0000 | 0.1000 |
| MMLU-College-v2 (70B) | Pass@K | $y=\text{floor}+(\text{ceil}-\text{floor})\cdot\exp(-\zeta\cdot K^{-}\zeta)$ | 0.7723 | 0.9655 | 1.3237 | 0.7746 | X | X | 0.9987 | 0.0000 | 0.0000 | 0.1000 |
| MMLU-College-v2 (70B) | Weaver | $y=\text{floor}+(\text{ceil}-\text{floor})\cdot\exp(-\zeta\cdot K^{-}\zeta)(1-(1-\pi)^{K^\gamma})$ | 0.8184 | 0.9436 | 2.2897 | 2.0761 | 0.4148 | 0.5000 | 0.9979 | 0.0000 | 0.0000 | 0.2500 |
| MMLU-College-v2 (70B) | Majority1@K | $y=\text{floor}+(\text{ceil}-\text{floor})\cdot\exp(-\zeta\cdot K^{-}\zeta)(1-(1-\pi)^{K^\gamma})$ | 0.8196 | 0.8413 | 0.0498 | 1.0001 | 0.2051 | 1.0153 | 0.8814 | 0.0000 | 0.0000 | 0.0100 |
| MMLU-College-v2 (70B) | Naive Ensemble | $y=\text{floor}+(\text{ceil}-\text{floor})\cdot\exp(-\zeta\cdot K^{-}\zeta)(1-(1-\pi)^{K^\gamma})$ | 0.8234 | 0.9262 | 3.9001 | 3.0000 | 0.3036 | 0.5000 | 0.9909 | 0.0000 | 0.0000 | 0.1000 |
| GPQA-v2-Diamond (8B) | Pass@K | $y=\text{floor}+(\text{ceil}-\text{floor})\cdot\exp(-\zeta\cdot K^{-}\zeta)$ | 0.2223 | 0.9976 | 2.4975 | 0.8328 | X | X | 0.9996 | 0.0000 | 0.0000 | 0.2500 |
| GPQA-v2-Diamond (8B) | Weaver | $y=\text{floor}+(\text{ceil}-\text{floor})\cdot\exp(-\zeta\cdot K^{-}\zeta)(1-(1-\pi)^{K^\gamma})$ | 0.2316 | 0.4678 | 0.0559 | 1.0027 | 0.2347 | 0.5940 | 0.9965 | 0.0000 | 0.0002 | 0.0100 |
| GPQA-v2-Diamond (8B) | Majority1@K | $y=\text{floor}+(\text{ceil}-\text{floor})\cdot\exp(-\zeta\cdot K^{-}\zeta)(1-(1-\pi)^{K^\gamma})$ | 0.2773 | 0.2979 | 0.1335 | 0.9633 | 0.0479 | 2.5000 | 0.5476 | 0.0000 | 0.0001 | 0.0500 |
| GPQA-v2-Diamond (8B) | Naive Ensemble | $y=\text{floor}+(\text{ceil}-\text{floor})\cdot\exp(-\zeta\cdot K^{-}\zeta)(1-(1-\pi)^{K^\gamma})$ | 0.2871 | 0.3845 | 8.5633 | 3.0000 | 0.2431 | 0.5000 | 0.9643 | 0.0000 | 0.0001 | 0.0100 |
| MATH-500-v2 (8B) | Pass@K | $y=\text{floor}+(\text{ceil}-\text{floor})\cdot\exp(-\zeta\cdot K^{-}\zeta)$ | 0.3986 | 1.0000 | 1.6393 | 0.8786 | X | X | 0.9976 | 0.0001 | 0.0001 | 0.0100 |
| MATH-500-v2 (8B) | Weaver | $y=\text{floor}+(\text{ceil}-\text{floor})\cdot\exp(-\zeta\cdot K^{-}\zeta)(1-(1-\pi)^{K^\gamma})$ | 0.4149 | 0.7417 | 0.0750 | 1.0042 | 0.3261 | 0.5179 | 0.9981 | 0.0000 | 0.0000 | 0.0100 |
| MATH-500-v2 (8B) | Majority1@K | $y=\text{floor}+(\text{ceil}-\text{floor})\cdot\exp(-\zeta\cdot K^{-}\zeta)(1-(1-\pi)^{K^\gamma})$ | 0.5061 | 0.6976 | 5.9901 | 1.1205 | 0.0782 | 2.5000 | 0.9960 | 0.0000 | 0.0000 | 0.0500 |
| MATH-500-v2 (8B) | Naive Ensemble | $y=\text{floor}+(\text{ceil}-\text{floor})\cdot\exp(-\zeta\cdot K^{-}\zeta)(1-(1-\pi)^{K^\gamma})$ | 0.5009 | 0.7342 | 1.6358 | 3.0000 | 0.3321 | 0.5000 | 0.9803 | 0.0001 | 0.0005 | 0.0100 |
| MMLU-Pro-v2 (8B) | Pass@K | $y=\text{floor}+(\text{ceil}-\text{floor})\cdot\exp(-\zeta\cdot K^{-}\zeta)$ | 0.3877 | 1.0000 | 1.8798 | 0.7549 | X | X | 0.9990 | 0.0000 | 0.0000 | 0.1000 |
| MMLU-Pro-v2 (8B) | Weaver | $y=\text{floor}+(\text{ceil}-\text{floor})\cdot\exp(-\zeta\cdot K^{-}\zeta)(1-(1-\pi)^{K^\gamma})$ | 0.4770 | 0.6937 | 3.1648 | 3.0000 | 0.2172 | 0.5705 | 0.9986 | 0.0000 | 0.0001 | 0.0500 |
| MMLU-Pro-v2 (8B) | Majority1@K | $y=\text{floor}+(\text{ceil}-\text{floor})\cdot\exp(-\zeta\cdot K^{-}\zeta)(1-(1-\pi)^{K^\gamma})$ | 0.4663 | 1.0000 | 2.4923 | 0.1016 | 0.0362 | 2.5000 | 0.9600 | 0.0001 | 0.0002 | 0.0500 |
| MMLU-Pro-v2 (8B) | Naive Ensemble | $y=\text{floor}+(\text{ceil}-\text{floor})\cdot\exp(-\zeta\cdot K^{-}\zeta)(1-(1-\pi)^{K^\gamma})$ | 0.4777 | 0.7207 | 3.0170 | 3.0000 | 0.1613 | 0.5000 | 0.9987 | 0.0000 | 0.0000 | 0.0100 |
| MMLU-College-v2 (8B) | Pass@K | $y=\text{floor}+(\text{ceil}-\text{floor})\cdot\exp(-\zeta\cdot K^{-}\zeta)$ | 0.4442 | 0.9912 | 1.0258 | 0.9265 | X | X | 0.9993 | 0.0000 | 0.0000 | 0.5000 |
| MMLU-College-v2 (8B) | Weaver | $y=\text{floor}+(\text{ceil}-\text{floor})\cdot\exp(-\zeta\cdot K^{-}\zeta)(1-(1-\pi)^{K^\gamma})$ | 0.6178 | 0.8447 | 1.1473 | 3.0000 | 0.3523 | 0.5000 | 0.9957 | 0.0000 | 0.0001 | 0.2500 |
| MMLU-College-v2 (8B) | Majority1@K | $y=\text{floor}+(\text{ceil}-\text{floor})\cdot\exp(-\zeta\cdot K^{-}\zeta)(1-(1-\pi)^{K^\gamma})$ | 0.6254 | 0.7186 | 0.5765 | 0.9281 | 0.2058 | 1.2287 | 0.9736 | 0.0000 | 0.0001 | 0.0100 |
| MMLU-College-v2 (8B) | Naive Ensemble | $y=\text{floor}+(\text{ceil}-\text{floor})\cdot\exp(-\zeta\cdot K^{-}\zeta)(1-(1-\pi)^{K^\gamma})$ | 0.6074 | 0.9813 | 1.2560 | 0.1636 | 0.3060 | 2.4651 | 0.9988 | 0.0000 | 0.0000 | 0.0100 |

Table 18: **WEAVER with 8B Models Exceeds Majority Voting and Naive Ensemble across All Datasets**: Candidate are generated with Llama 3.1 8B Instruct while the weak verifiers are 8B parameters or smaller in size.

| | Methodology | Generations ($K$) | MATH 500 | GPQA | MMLU College | MMLU Pro | Average |
|---|---|---|---|---|---|---|---|
| | | | **Datasets** | | | | |
| Baselines | First Sample | 1 | 49.8% | 28.3% | 64.1% | 46.6% | 47.2% |
| | Majority Voting | 100 | 69.0% | 30.5% | 72.7% | 56.4% | **57.2%** |
| | Top-Ranked RM from RewardBench [39] | 100 | 73.8% | 25.4% | 70.1% | 53.4% | 55.7% |
| | Top-10 RM Ensemble from RewardBench [39] | 100 | 70.2% | 22.1% | 73.9% | 49.4% | 53.9% |
| | Multi-Agent Verification [41] | 100 | 65.4% | 31.4% | 70.5% | 55.2% | 55.6% |
| | Self-Verification [93] | 100 | 71.4% | 32.2% | 70.4% | 53.0% | 56.8% |
| | WEAVER | 100 | 80.0% | 47.1% | 85.7% | 67.2% | **70.0%** |
| | GPT-4o-mini | 1 | 76.8% | 38.4% | 82.2% | 61.8% | 64.8% |
| | Claude 3.5 Haiku | 1 | 70.0% | 36.4% | 75.9% | 65.2% | 61.9% |
| | Oracle Verifier (Pass@100) | 100 | 99.2% | 95.2% | 98.5% | 96.8% | **97.4%** |

## C.4 Scaling Candidate Generations

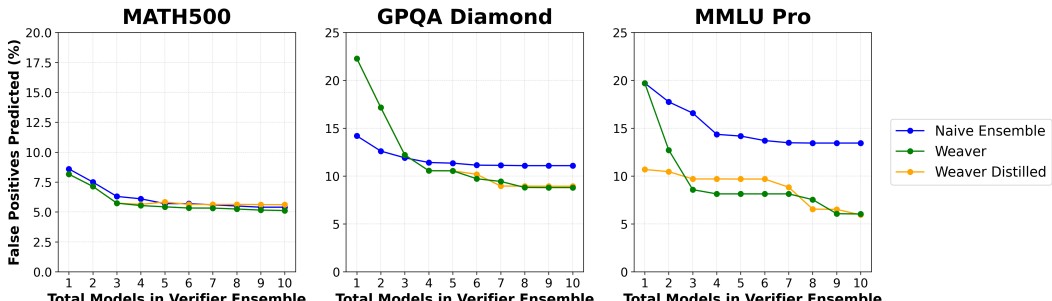

Figure 12: **False Positive Rates across Verification Systems**

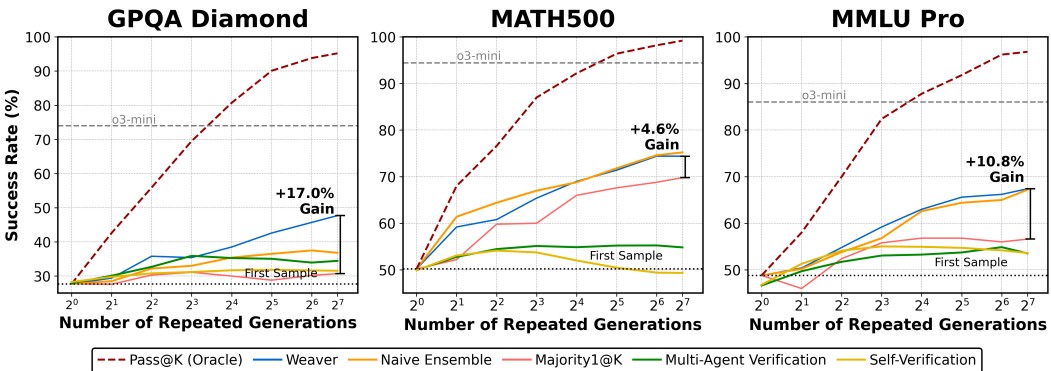

Figure 13: **WEAVER Scaling - 8B Generations and Models**

## C.5 Scaling Verifier Count

In Table 19, we include results of scaling verifier scores. We note that for reward models (RMs), which are typically deterministic [39, 71], multiple scores must be obtained by varying the prompt; for LM Judges, we can vary either the prompt or the sampling temperature to generate diverse outputs from the same model (Table 19). We find that for both types of weak verifiers, RMs and LM judges, scaling the number of models yields better performance than sampling multiple evaluations from the same model via prompt tuning or temperature variation. However, we note that these approaches are complementary.

Table 19: **Ensembling with Multiple Verifiers Outperforms Increased Sampling with Single Verifier**: Candidate responses are generated with Llama 3.3 70B Instruct while the weak verifiers range in size from 8B to 72B parameters. For details on prompting, please see Appendix C.7.

| Methodology | Benchmarks | | |
|---|---|---|---|
| | MATH500 | GPQA | MMLU Pro |
| First Sample | 78.0% | 42.9% | 69.9% |
| Majority Voting | 83.0% | 47.4% | 74.4% |
| Best Reward Model (1 Score) | 94.4% | 58.4% | 81.8% |
| Best Reward Model (5 Scores, 5 Prompts) | 93.2% | 55.3% | 82.5% |
| Top-5 Most Accurate Reward Models | **95.4%** | 64.1% | **87.3%** |
| Best LM Judge (1 Score) | 90.2% | 61.1% | 79.5% |
| Best LM Judge (5 Scores, 5 Prompts) | 88.1% | 57.2% | 80.8% |
| Top-5 Most Accurate LM Judges | 93.4% | **65.2%** | 85.4% |

**Accuracy Heatmap: MMLU-Pro (70B)**

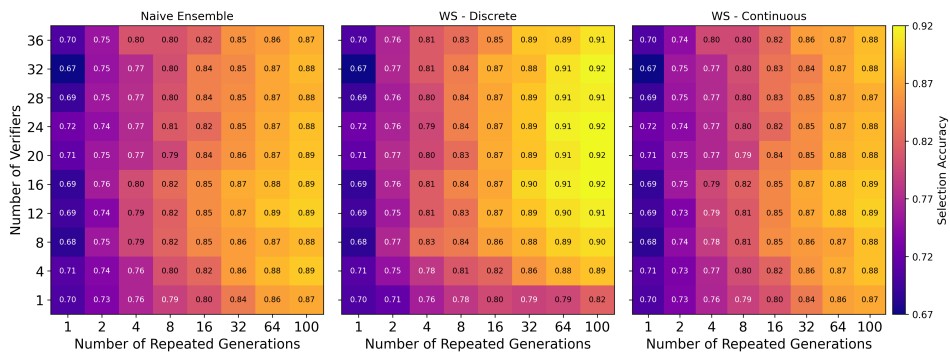

Figure 14: **WEAVER Performance Improvements from Scaling Generations and Verifiers**: Increased candidate generations and weak verifiers available generally improves performance.

When breaking down weak verifiers into RMs or LM Judges, individually, we find that additional LMs leads average gains of 5.4% and 6.1%, respectively (Table 19). In contrast, sampling additional scores from a single RM or LM judge yields only 0.8% and 1.1% gains on average. These results suggest that leveraging the complementary strengths of multiple verifiers can be more effective than eliciting multiple judgments from a single verifier. Sec. C.7 provides additional details on the verifier prompting. Finally, Figure 14 illustrates the tradeoff of scaling the number of verifiers versus increasing the number of scores from a single verifier, showing that scaling verifiers is helpful when the coverage increases as we increase sample count.

In Fig. 14 illustrates how the number of verifiers and repeated generations interact to influence success rate. We observe that increasing the number of generations tends to be more effective than increasing the number of verifiers alone—but only when paired with the right verification strategy. For example, naive ensembling of verifiers plateaus in performance even as more generations are added, whereas WEAVER continues to improve with both axes. This highlights that generation diversity is a stronger driver of performance than verifier count alone, and that weak supervision methods like WEAVER are essential to fully leverage this diversity. We illustrate the verification generation tradeoff for additional datasets in Appendix C.3.

| Model | GPQA Diamond | MATH500 | MMLU Pro | Average |
|---|---|---|---|---|
| Weaver with Llama Verifier | 59.4% | 91.1% | 87.4% | 79.3% |
| Naive Ensemble | 52.1% | 88.9% | 84.1% | 75.0% |
| Weaver with Llama Verifiers - Improvement % | +8.3% | +2.2% | +3.3% | +4.3% |

Table 20: **Performance Comparison of WEAVER with Llama Verifiers and Naive Ensemble with All Verifiers**

| Method | GPQA Diamond | MATH500 | MMLU Pro | Dev Set Size | Average |
|---|---|---|---|---|---|
| *Oracle & Frontier Models* | | | | | |
| Oracle Verification (pass@100) | 88.9 | 99.8 | 98.4 | – | 95.7 |
| o3-mini | 78.3 | 96.2 | 85.5 | – | 86.7 |
| GPT-4o | 59.1 | 74.6 | 73.3 | – | 69.0 |
| *Unsupervised Baselines* | | | | | |
| Best-of-N (random) | 48.7 | 82.8 | 70.6 | 0% | 67.4 |
| Majority Voting | 51.2 | 88.2 | 79.4 | 0% | 72.9 |
| Self-Verification | 52.3 | 86.5 | 78.1 | 0% | 72.3 |
| Naive Ensemble (mean) | 54.5 | 90.6 | 84.3 | 0% | 76.5 |
| *Supervised Baselines – 1.0x Dev Set (100% labeled)* | | | | | |
| Decision Tree | 64.5 | 87.1 | 86.3 | 100% | 79.3 |
| AdaBoost[†] | 67.1 | 90.2 | 84.1 | 100% | 80.5 |
| Logistic Regression (L1)[‡] | 73.1 | 92.5 | 90.2 | 100% | 85.3 |
| Logistic Regression (L2) | 78.1 | 97.2 | 92.0 | 100% | 89.1 |
| *Supervised Baselines – 0.01x Dev Set (1% labeled)* | | | | | |
| Decision Tree | 52.1 | 69.4 | 71.0 | 1% | 64.2 |
| AdaBoost[†] | 53.4 | 72.1 | 71.4 | 1% | 65.6 |
| Logistic Regression (L2) | 55.9 | 70.5 | 72.1 | 1% | 66.2 |
| Logistic Regression (L1)[‡] | 54.1 | 72.4 | 73.5 | 1% | 66.7 |
| XGBoost[§] | 57.5 | 75.1 | 76.3 | 1% | 69.6 |
| **Weaver (Ours)** | **71.1** | **93.4** | **90.2** | **1%** | **84.9** |
| Weaver-Distilled (400M) | 66.4 | 91.1 | 87.8 | 1% | 81.8 |

Table 21: **Comparison of Verification Methods across Reasoning Benchmarks**. WEAVER achieves strong performance with minimal labeled data (0.01x dev set) compared to supervised methods that require 100× more labels.

[†]AdaBoost: n_estimators ∈ {25, 50, 75, 100, 150}, learning_rate ∈ {0.1, 0.3, 1.0}, max_depth ∈ {1, 2}, selected via cross-validation.
[‡]L1-Regularized LR: $\lambda = 0.01$, selected via cross-validation.
[§]XGBoost: default parameters with num_rounds=10, objective=binary:logistic.

## C.6 WEAVER Distillation

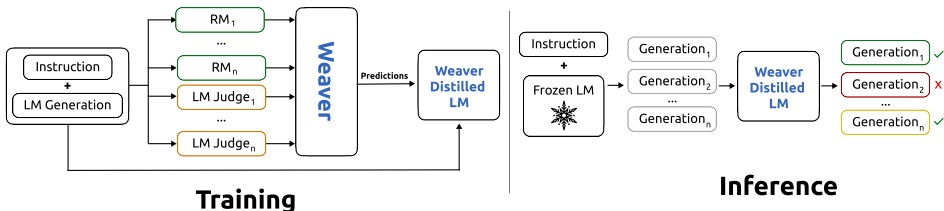

Figure 15: **Overview of WEAVER Distillation (Section 6)**

For the loss function in WEAVER distillation, we utilized cross-entropy loss with Adam [35]. Our classification architecture comprises a single linear classification layer with 0.1 dropout applied to the input, which consists of the final hidden state from the $[CLS]$ token. Regarding learning dynamics, we implemented linear warmup and linear decay via the Sentence-Transformers library [61], employing a learning rate of 5e-6 and training batch size of 64 across all experimental setups.

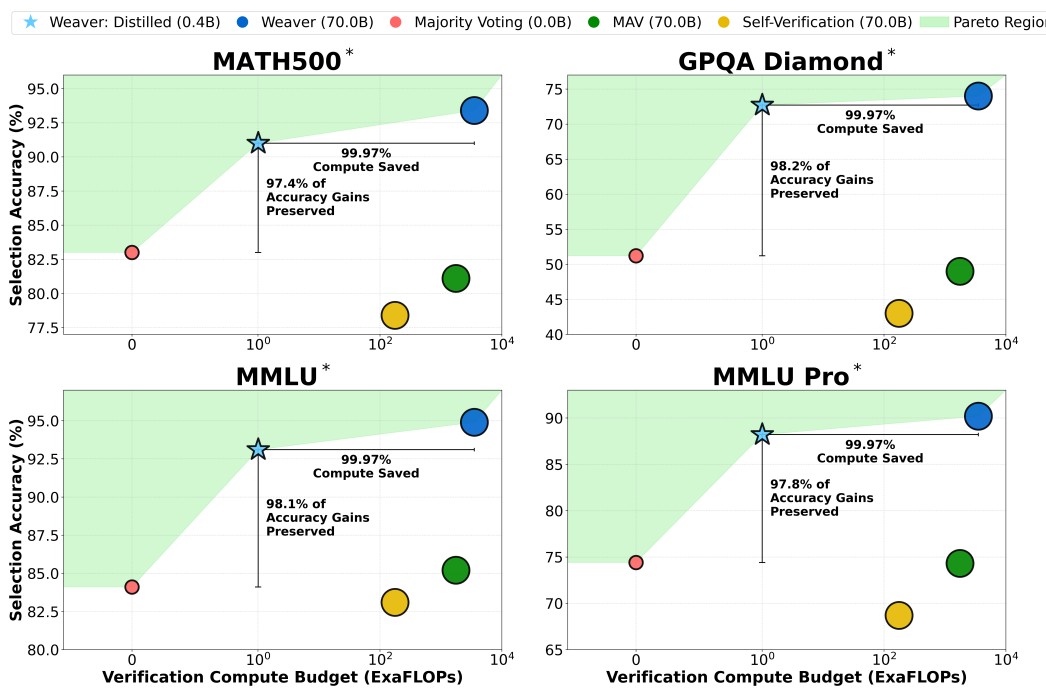

Figure 16: **WEAVER Distilled - Pareto Frontiers**: *We train/evaluate on an 80:20 split.

Table 22: **Distillation Comparison of WEAVER and Naive Ensemble Across Different Training Set Sizes**

| Methodology | Dataset | Training Set as Percentage of Entire Dataset | | | | | Full System |
| --- | --- | --- | --- | --- | --- | --- | --- |
| | | 5% | 10% | 20% | 50% | 80% | |
| WEAVER | MATH500 | 78.4% | 80.7% | 83.9% | 88.2% | 91.4% | 93.4% |
| | GPQA Diamond | 42.6% | 46.8% | 52.7% | 63.1% | 71.8% | 73.2% |
| | MMLU College | 83.5% | 85.2% | 87.6% | 91.0% | 93.1% | 94.9% |
| | MMLU Pro | 69.2% | 72.5% | 76.8% | 83.7% | 87.8% | 90.2% |
| NaiveEnsemble | MATH500 | 77.8% | 79.6% | 82.1% | 86.4% | 89.1% | 92.4% |
| | GPQA Diamond | 42.1% | 44.7% | 48.9% | 56.2% | 62.8% | 66.2% |
| | MMLU College | 84.0% | 85.3% | 87.2% | 90.8% | 93.5% | 95.1% |
| | MMLU Pro | 69.5% | 71.8% | 74.9% | 80.3% | 84.7% | 87.4% |

## C.7 Individual Verifier Optimization

While WEAVER primarily focuses on aggregating multiple weak verifiers to improve overall verification quality, this appendix explores complementary techniques for optimizing individual verifiers. As mentioned earlier in the paper, existing weak verifiers often suffer from high false positive rates [73], which can limit their effectiveness, even within an ensemble.

As we scale the number of repeated samples and employ multiple verifiers, the precision of each individual verifier becomes increasingly important relative to recall. When many candidate solutions are available, a verifier can afford to miss some correct solutions (false negatives) as long as its positive predictions are highly reliable (high precision).

This observation motivates exploring methods to enhance individual verifier quality through methods such as prompt optimization — tailoring verifier prompts to maximize performance, particularly precision, with minimal or no labeled data.

### C.7.1 LM Judge Prompt Optimization

LM judges often suffer from biases such as position bias (favoring answers in certain positions), verbosity bias (preferring longer answers), and self-enhancement bias (preferring answers similar to their own generation patterns) [40, 95], suggesting sensitivity to system and input prompt design.

Throughout our WEAVER experiments, we used fixed, manually engineered prompts for our LM judge verifiers. However, optimizing these prompts could potentially improve individual verifier precision and reliability. Multi-Agent Verification [41] demonstrates this by crafting specialized prompts for specific verification aspects.

We explored systematically optimizing verifier prompts using DSPy [34], an open-source library that provides algorithms for optimizing language model prompts through discrete search over prompt candidates guided by a metric function. DSPy optimization works by generating, evaluating, and refining prompts that maximize task performance on a small labeled dataset.

**Experimental Setup**: We investigate two dimensions of prompt optimization: (1) **optimization space scaling**, where we progressively expand what the optimizer can modify from system instruction only (0-shot) to including 3 demonstrations (3-shot) and 5 demonstrations (5-shot); and (2) **training data size scaling**, where we vary labeled data from 1% to 16% to determine how much data is necessary for effective prompt optimization.

Our experimental setup uses training examples containing instruction-generation pairs. Since our datasets have multiple generations per instruction (up to 100), we group examples by instruction before splitting to prevent data leakage between train and validation sets. We hold out 50% of the dataset instructions (each paired with 100 candidate generations) for evaluation. For the optimization space scaling experiment, we randomly select $n$ generations such that $n \times \text{len}(\text{dataset})/2 = 250$, maximizing training set diversity while maintaining a fixed training set size. For the data scaling experiment, we train on different percentages (1%, 2%, 4%, and 16%) of the dataset by calculating the number of instructions as $\lceil \text{num\_problems\_in\_dataset} \times (\text{train\_percentage}/100) \rceil$ and selecting repeated samples for each instruction with samples $= \min(\max(4, \text{num\_problems} \times 2), 20)$ to avoid overfitting. We use a consistent random seed to ensure identical dataset splits between optimization runs.

**Results**: Figure 17 shows results across different datasets and optimization configurations. While we don't observe clear scaling relationships across all datasets (possibly due to the increased stochasticity of LLM-based optimization), we observe an average precision gain of 3.8% of the best judge over the chain-of-thought (CoT) baseline judge. MATH500 shows the largest jump in precision of 9% and shows clear improvement in precision as the optimization space is scaled.

The scaling behavior with training data size (Figure 18) shows slight log-linear improvements in precision as we increase training data, though gains differ by dataset. MMLU-College shows minimal benefit from additional data, while the remaining datasets see an average boost of 3.2% in precision when scaling the training data size from 1% to 16% of the original dataset.

(Figure 19) reveals that optimized prompts often improve both precision and accuracy by reducing false positive rates - essentially making judges more conservative in their correctness assessments. This is particularly valuable in the repeated sampling regime, where higher precision improves overall verification quality.

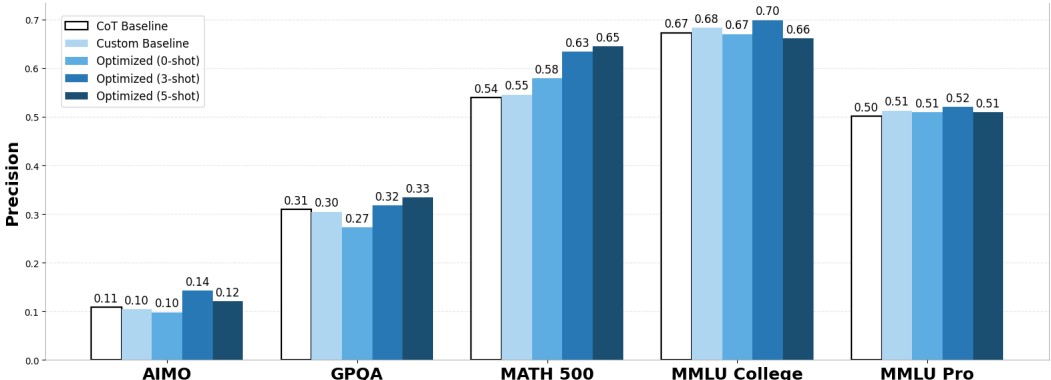

Figure 17: **LM judge prompt optimization using 250 labeled examples consistently yields precision gains**. Baseline methods (CoT and Custom) are compared against DSPy-optimized prompts with varying numbers of demonstrations (0-shot, 3-shot, and 5-shot).

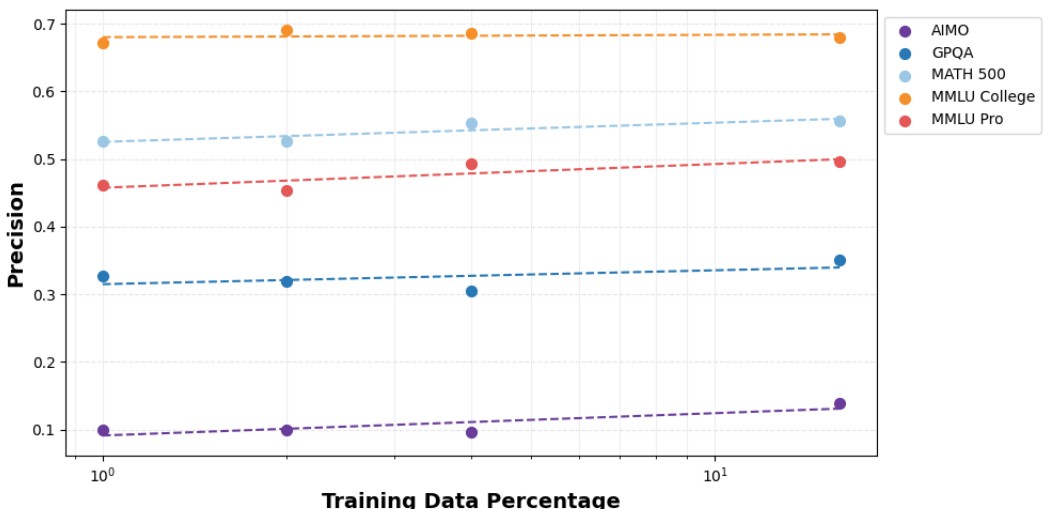

Figure 18: **Scaling LM judge prompt optimization training data leads to modest precision gains.** The x-axis shows the percentage of training data used (log scale), and the y-axis shows precision.

These findings suggest that prompt optimization can be a valuable complement to WEAVER's aggregation approach. Even with limited labeled data, targeted prompt engineering can enhance individual verifier quality, benefiting the ensemble as a whole. Further research is needed to define a more systematic recipe for verifier prompt optimization. Additionally, it remains a question of whether we can extend prompt optimization to discriminative reward models to enjoy similar gains in performance.

## D    Miscellaneous

### D.1    Compute Requirements

**Hardware Infrastructure.** Our experiments were conducted using 4 compute nodes, each equipped with 8 NVIDIA H100 GPUs (80GB HBM3 memory per GPU), for a total of 32 H100 GPUs. Each node was configured with high-bandwidth NVLink connections between GPUs and inter-node communication was facilitated via NVIDIA NVLink Switch System to minimize communication overhead during distributed training and inference.

**Model Parallelism and Distribution.** For our 72B parameter language models, we employed a hybrid parallelism strategy combining tensor parallelism, pipeline parallelism, and data parallelism:

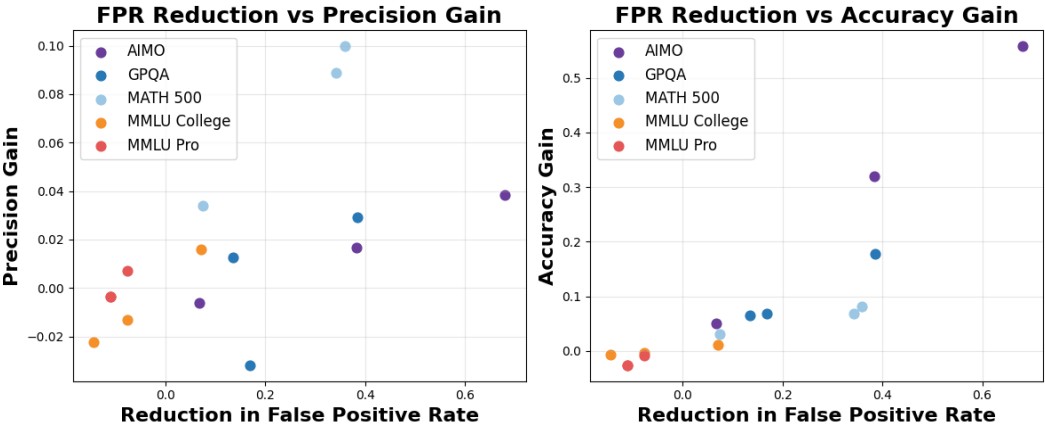

Figure 19: **Optimized prompts often improve LM judge performance by reducing false positive rates.**

- 8-way tensor parallelism across GPUs within each node
- 4-way pipeline parallelism across nodes
- Data parallelism for batch processing

**Storage Requirements.** Processing datasets of 100GB+ required significant storage infrastructure:

- 4TB NVMe SSDs per node for dataset caching and checkpoints
- 100TB shared network storage for full dataset repository

**Software Stack.** Our experiments were powered by:

- NVIDIA CUDA 12.2
- PyTorch 2.1 with NVIDIA NCCL for distributed communication
- DeepSpeed ZeRO Stage 3 for memory optimization
- Distributed data loading with webdataset format for efficient streaming

