# OpenReview forum: "Weaver: Shrinking the Generation-Verification Gap by Scaling Compute for Verification"
_NeurIPS.cc/2025/Conference — NeurIPS 2025 poster_

### Official Review · Reviewer_cAMM · 2025-06-26

**Clarity:** 2
**Significance:** 3
**Originality:** 3
**Rating:** 3
**Confidence:** 3

**Summary:**

The authors introduce Weaver, a method for improving verification of LLM outputs. This method, given N "verifiers" (or "reward models") and a development set of labeled examples, assigns weights for each of the verifiers, which are used to aggregate the verifiers scores. The authors show that their method performs better compared to basic averaging. Additionally, it is shown that Weaver -- en ensemble of multiple verifiers (LLMs) -- can be effectively distilled into one model.

**Questions:**

1. If I understood correctly, before aggregating the verifier scores, you binarize them. Why? It looks like discarding relevant information -- if a verifier gives a score of, say 0.55, it may mean it is uncertain and it will not skew the average in either direction. Also, binarization requires an additional step of selecting the right threshold. Did you experiment with un-binarized scores?

2. Did you consider comparing Weaver with other methods for obtaining weights?

3. How the base LLM was sampled? To achieve diversity of samples, crucial for the majority voting and verifier-dependent approaches, one need to sample with non-zero temperature. But too high temperature degrades the quality of individual samples. So this is a crucial parameter, but it is not optimized nor discussed...

4. What development set (D^dev) was used to tune Weaver's weights? How large?

5. How top-K verifiers were selected? Based on the performance on the development sets?

**Ethical Concerns:**

["NO or VERY MINOR ethics concerns only"]

**Final Justification:**

During the discussion the authors were active and answered all my questions as well as provided useful additional experiments. I believe that incorporating the explanations and new results is important and will improve the paper.

In general, I find the presented approach of combining multiple verifiers using "moment matching" on the unlabeled test set and using small dev set for calibration an interesting and useful approach. The core idea of it is perhaps not novel, but adapting it to the verification setting is original. However, the work has its weaknesses:
1. Only two Llama models tested as generators of base responses.
2. Not using more direct metric like AUC for assessing the quality of ranking responses from various (combinations of) verifiers.
3. The suite of experiments could be more consistent and better designed (e.g., I got confused during the rebuttal what the dev set is used for in various experiments: class balance estimation? threshold estimation?)
4. The algorithm itself could be explained better. For instance, if it's similar to Naive Bayes in it's first approximation, it would be helpful to emphasize it.
5. Now I think that the paper could spend more time analyzing or interpreting the results. Is there some intuition why Weaver outperforms other approaches? Maybe seeing the weights Weaver and other methods assign to the 33 verifiers would be a good addition to the appendix.

I'm rising the rating but only by one point because of the above reasons.

**Limitations:**

Yes.

**Paper Formatting Concerns:**

Discrepancy between the main paper and appendices -- for instance, I couldn't find Appendix C.9.

**Quality:**

2

**Strengths And Weaknesses:**

**Strengths**

The paper tackles an interesting and important problem -- when multiple samples are extracted from an LLM, often some are correct whereas the others are not; but an additional approach is needed to recognize the correct ones -- thus the need for good verifiers.

The paper contains substantial number of experiments involving multiple verifiers.

**Weaknesses**

1. Experimental setup seems a bit weak / not entirely adequate:

1.1. In addition to the down-stream task performance on the benchmarks it would be good to see the performance of the Weaver scores as measured by, e.g., AUC, and compared to the other approaches. This would be more direct assessment of the Weaver performance.

1.2. In addition to the top-K approach it would be good to see how Weaver compares to some more straightforward methods for obtaining weights, e.g. logistic regression (mentioned in lines 168-170). Also, logistic regression with L1 regularization seems as an interesting alternative, as it would allow to reduce the number of verifiers in the ensemble.

1.3. The authors assume that the 33 verifiers used in the experiments are de-correlated. However, this likely is not the case -- perhaps some of the verifiers were obtained by training on similar data? This may be the main reason why the simple averaging does not work. Perhaps selecting uncorrelated subset verifiers would be better that assigning non-zero weights for each of the verifiers?

2. Important technical details missing (temperature of sampling, size of D^dev, how top-K were selected). See questions.

3. Over-exaggerated claims / questionable presentation of some aspects:

3.1. You write: "Naively aggregating weak verifiers is insufficient for reliable verification", but as far as I can tell, Weaver cannot achieve reliable verification either -- it is just to some degree better that normal ("naive"...) averaging.

3.2. You emphasize that Weaver does not need much labeled data, and that alternative approaches may require more labels (lines 58-60). But did you actually compare Weaver with the alternative approaches? Maybe logistic regression can achieve better performance on the same number of labeled examples (D^dev)?

4. The general idea of calling multiple (33 in the experiments) verifiers seems not very practical -- the verification becomes quite expensive. I understand that this motivated the distillation experiment, but there, in turn, there is added cost of training.

5. The authors experiment with many verifiers but only with one base model (Llama 3.3 70B). Seeing the results for other base models would make the conclusions stronger.

6. As the authors emphasize the compute-time efficiency of verification, it would be perhaps worth discussing the recent literature concerning this topic, where efficient verification is achieved by learning from the base LLM's internal representations -- see, e.g., [1,2].

[1] Correctness Assessment of Code Generated by Large Language Models Using Internal Representations (https://arxiv.org/abs/2501.12934) \
[2] Lightweight Latent Verifiers for Efficient Meta-Generation Strategies (https://arxiv.org/abs/2504.16760)

---

> ### Author Rebuttal · Authors · 2025-07-31
>
> We appreciate your recognition of the problem, our experiments and constructive suggestions.
>
> Below, we address your concerns, grouping related weaknesses and questions together. We include **new experiments** on: L1 regularization, Weaver on correlated verifiers, additional baselines, and using continuous verifier scores. We have included these results in the paper and would be grateful if you would consider **updating your score** in light of these improvements.
>
> **W1.1:  Performance of the Weaver scores vs other approaches as measured by AUC:**
>
> Following your suggestion, **we computed AUC for Weaver's scores**. Weaver achieves an average AUC of 0.7 (e.g. on GPQA for dev set 0.01x), outperforming naive ensembles (0.6) and logistic regression (0.65). The results reflect Weaver's improved ranking of correct vs. incorrect responses.
>
> **W(1.2, 2, 3.2, Q2, Q4): Comparing Weaver with other methods for obtaining weights and dependence on dev set size**
>
> Comparisons with logistic regression (LR) and Naive Bayes (NB) using different sized dev sets are provided in Tables 4 and 15.  Weaver outperforms LR at smaller dev sizes (0.01x) and remains close for  large dev sets (0.5x). This advantage stems from how Weaver leverages labeled data to calibrate the class balance but not for training. For the main experiments, Weaver uses a dev set equivalent to 1% of the benchmark (e.g. ~5 problems for MATH-500, ~6 for GPQA’s 640).
>
> The LR experiments in the paper use L2 regularization (λ=1). **We add L1 regularization experiments below** (λ=0.01 tuned via cross-validation). While LR +L1 regularization drops up to 50% of the verifiers, it does not reach the same performance as LR with L2 regularization.
>
> **Table 1: L1 vs. L2 Logistic LR with 1.0x Dev Set**
> |Model|GPQA|MATH500|MMLU Pro|Average|
> |---|---|---|---|---|
> |Supervised L1 LR|73.1%|92.5%|90.2%|**85.3%**|
> |Supervised L2 LR|78.1%|97.2%|92.0%|**89.1%**|
> |Weaver|72.7%|92.4%|90.4%|**85.2%**|
>
> **Table 2: L1 vs. L2 Logistic LR with 0.1x Dev Set**
> |Model|GPQA|MATH500|MMLU Pro|Average|
> |---|---|---|---|---|
> |Supervised L1 LR|54.1%|72.4%|73.5%|**66.7**|
> |Supervised L2 LR| 55.9%|70.5%|72.1%|**66.2%**|
> |Weaver|67.1%|90.4%|87.0%|**81.5%**|
>
> We also added **two new supervised baselines, AdaBoost and Decision Tree.** We find that tree-based supervised strategies underperform.
>
> |Model|GPQA Diamond|MATH500|MMLU Pro|Average|
> |---|---|---|---|---|
> |AdaBoost (1.0x Dev Set)|67.1%|90.2%|84.1%|**80.5%**|
> |DecisionTree (1.0x Dev Set)|64.5%|87.1%|86.3%|**79.3%**|
>
> **W1.3: Concern that verifier correlation may undermine simple averaging, and suggestion to instead select an uncorrelated subset of verifiers.**
>
> Weaver does not assume de-correlated verifiers. Instead, it assumes conditional independence given the true labels, meaning that verifier dependencies can be explained through their relationship to the ground truth. To illustrate Weaver’s effectiveness when verifiers are highly correlated, **we add the following experiment**: We selected all 13 verifiers from our main experiments that were derived from the Llama family (e.g., Llama 3.1 8B/70B variants) and fine-tuned on similar RLHF data, representing the subset most likely to be highly correlated. These verifiers have an average Pearson correlation of 0.491. Table 1 below shows that Weaver achieves a +4.3% average improvement over naive ensembling—a small drop from  the +6.8% improvement when we use the full, more diverse pool (Table 2; average correlation 0.281). These results indicate that while diversity enhances performance, Weaver remains effective even when verifiers are highly redundant.
>
> While selecting an uncorrelated subset of verifiers might improve naive ensemble performance, there is a tradeoff involved in such a selection process. The optimal ensemble balances diversity and individual verifier accuracy [7], a tradeoff that Weaver learns to navigate using unlabeled data.
>
> **Table 1: New Results - Weaver with Llama Verifiers**
> |Model|GPQA|MATH500|MMLU Pro|Average|
> |---|---|---|---|---|
> |Weaver with Llama Verifiers|59.4%|91.1%|87.4%|79.3%|
> |Naive Ensemble|52.1%|88.9%|84.1%|75.0%|
>
> **Table 2: Existing Results - Full Weaver with All Verifiers**
> |Model|GPQA|MATH500|MMLU Pro|Average|
> |---|---|---|---|---|
> |Weaver|66.4%|93.4%|90.2%|83.3%|
> |Naive Ensemble|54.5%|90.6%|84.3%|76.5%|
>
> **W3.1: "Weaver cannot achieve reliable verification either - it is just to some degree better than normal ("naive") averaging."**
>
> Thank you for the thoughtful comment. We agree that Weaver does not turn weak verifiers into a perfect oracle. Rather, it narrows the generation-verification gap (e.g., by 14.5% on average; see Figure 1, Table 1) by improving verification accuracy through scalable ensembling. We will revise the wording to better reflect this goal and appreciate the opportunity to clarify.
>
> **W4: Concern about the practicality of using 33 verifiers and the additional distillation cost**
>
> We respectfully disagree that using multiple verifiers is impractical and raise the following three points:
>
> **1) Verification is parallelizable and can leverage distributed/idle compute.**
>  Verifier models run independently and require only a single forward pass—no autoregressive decoding—making them highly parallelizable across devices. Many verifiers (8B–30B) can run on consumer GPUs, enabling distributed or overnight batch verification using idle compute.
>
> **2) Weaver performs well even with few verifiers.** As shown in Figure 4, Weaver yields strong gains even with limited ensembles. Instead of the full 33 used in experiments, we see that with just 3 verifiers, we can still outperform naive ensembling on GPQA, MATH500, and MMLU Pro.
>
> **3) Distillation pays off quickly.** Training the 400M distilled verifier requires ~10 GPU-hours (Appendix C.6). This small training cost is quickly amortized; after roughly 1,000 inferences, the distilled model becomes more compute-efficient than the full ensemble while retaining 98.7% of its accuracy.
>
> **W5: “Only one base model (Llama 3.1 70B) is used.”**
>
> Our paper also includes results for Llama 3.1 8B as a generator in Tables 3 and 18, where our results demonstrating Weaver’s gains still hold. In our final draft, we will include results on additional model families, such as the Qwen models.
>
> **W6: "Add discussion of recent literature on efficient verification via learning from internal representations [1, 2]”.**
>
> We appreciate the valuable references [1,2] and have incorporated them. Both propose verifiers that use LLM hidden states to predict generation correctness. In contrast, Weaver is a black-box aggregation framework, designed for flexibility and compatibility across open- and closed-source models. These internal verifiers are fully compatible with Weaver and could be included in its pool to provide complementary signals and reduce verification cost.
>
> **Q1: Concern that binarizing scores discards relevant information**
>
> We explain the motivation behind binarization and present results using a continuous version of our algorithm.
>
> **Due to differences in how verifiers are trained, their continuous scores vary significantly in distribution.** Verifier scores can range from unbounded log-likelihoods to learned regression scores. In our analysis, we found that even after normalization (e.g., min-max scaling or quantile adjustment), verifier scores exhibited sharp multimodality, heavy skew toward extremes, flat or noisy profiles, and inconsistent calibration—e.g., a score of 0.8 could signal high confidence in one verifier and uncertainty in another. These inconsistencies complicate continuous modeling, as they introduce additional parameters or hierarchical structures that increase the complexity of learning the latent variable model.
>
> **Modeling binary scores aligns better weak supervision frameworks,** enabling the use of a tractable model with fewer parameters and efficient moment-matching for latent variable estimation. These models are theoretically grounded and have proven effective in noisy-label scenarios.
>
> **Smart binarization does not degrade performance**: To assess what is lost from binarization, we ran a new experiment where we compared several strategies in a logistic regression setting. These included: (1) no binarization, (2) class balance: choosing a threshold so that the verifier balance matches the class balance; and (3) quantile: assign positive labels to the top 15% of scores. While continuous scores achieved strong AUC when ample labels were available, the class balance and quantile binarization matched or outperformed no binarization under low-label regimes.
>
> **Continuous Weak Supervision algorithm performs poorly**: We also implemented a recent continuous weak supervision method based on Gaussian moment matching [6], which can operate on raw verifier scores.  This approach consistently underperforms compared to the binarized model. The continuous model struggles with the non-Gaussian and asymmetric nature of real-world verifier scores, leading to inaccurate reliability estimates and degraded aggregation performance.
>
> |Model|GPQA|MATH500|MMLU Pro|Average|
> |---|---|---|---|---|
> |Continuous Weaver|63.1%|89.6%|88.4%|80.4%|
> |Discrete Weaver|66.4%|93.4%|90.2%|83.3%|
>
> **W2, Q3: How is the base LLM sampled?**
>
> We sample the base LLM temperature 0.7 to balance diversity and sample quality, following prior work for reasoning and math tasks [3, 4, 5].
>
> **Q5: "How top-K verifiers were selected? Based on the performance on the development sets?"**
>
> In the experiments for Figure 2 and 4, the top-K verifiers were computed in an oracle setting using the full dataset.
>
> [1] https://arxiv.org/abs/2501.12934
>
> [2] https://arxiv.org/abs/2504.16760
>
> [3] https://arxiv.org/abs/2502.17578
>
> [4] https://arxiv.org/abs/2407.21787
>
> [5] https://arxiv.org/abs/2408.03314
>
> [6] https://arxiv.org/abs/2112.03865
>
> [7] https://arxiv.org/abs/2302.00704

---

> ### Comment · Reviewer_cAMM · 2025-08-01
>
> Thank you for the response! However, a few more questions.
>
> 1. In Table 1 in the paper you show numbers for top-1 and top-10 verifiers ensembles on RewardBench. However, a more sensible baseline would be to compare with top-1 and top-10 ensembles for a given benchmark. Could you provide the results for these?
>
> 2. Weaver in Table 1 has better performance on MATH500 than in the first two tables in the rebuttal, even though it was trained on 1% of data compared to 10% and 100%. How is it possible?
>
> 3. As I understand, for training you select N queries and generate M responses, which result in NxM examples labeled with 0 or 1, is that correct? So for MATH500 you fit Weaver having 5 * 100 examples? Is that correct?
>
> 4. When you fit logistic regression, do you use binarized score, raw scores, or scores normalized is some way?
>
> 5. Another idea for a natural baseline which is not included in the paper is this: each verifier induces a ranking of generated responses, and we subsequently average the rankings (e.g., by computing average rank for each sample). Could you comment on that and perhaps evaluate such an approach?
>
> 6. When you compute the naive ensemble, do you first binarize scores? Or normalize them in some way? Because if not, taking the average score may not be meaningful (you mentioned that different verifiers operate on different score scales, etc.)
>
> 7. What precisely you mean by AdaBoost?
>
> 8. Could you additionally run XGBoost with default parameters and num_rounds=10, objective=binary:logistic?

---

> ### Author Response · Authors · 2025-08-05
>
> Thank you for your fast response and for clarifying your questions! We address them below.
>
> > 1. In Table 1 in the paper you show numbers for top-1 and top-10 verifiers ensembles on RewardBench. However, a more sensible baseline would be to compare with top-1 and top-10 ensembles for a given benchmark. Could you provide the results for these?
>
> We agree that comparing against top-1 and top-10 ensembles for a given benchmark are valid baselines to compare against. This baseline is in Figure 4 of our main paper, in which we ablate top-K verifier naive ensembles from 1 to 15, inclusive. We find that Weaver achieves 8.5% better performance, on average, than the top-1 verifier alone. Furthermore, across top-K verifier ensembles, Weaver achieves 5.9% better performance, on average, compared to the corresponding top-K naive ensemble.
>
> Note that the top-K ensembles assume that you have oracle knowledge of the true ranking of verifiers for a given task, making it a competitive baseline for comparison. We believe benchmarking the top-1 and top-10 verifier ensembles from RewardBench are also practical baselines since they represent some of the best standard approaches to verification, assuming the practitioner has limited access to ground truth labels on a new task.
>
> Below, we also test *estimated top-1 and top-10 ensembles*, calculating verifier ranking from the 1% dev set used for Weaver rather than using the oracle information. This approach is more realistic as it reflects what a practitioner would do with limited labeled data. Using the dev set, we rank verifiers by their accuracy and select the top-1 and top-10 performers to create ensembles. As shown in Table 1 below, when using dev-set-estimated rankings rather than oracle rankings, Weaver (83.3% average) significantly outperforms both the estimated top-1 ensemble (71.7% average) and the estimated top-10 ensemble (77.2% average). This 11.6% improvement over the top-1 and 6.1% improvement over the top-10 ensemble demonstrates that Weaver's learned aggregation weights are more effective than simply selecting the best-performing verifiers based on limited development data.
>
> ### Table 1: Estimated Top-1 and Top-10 Ensembles from Dev Set
> | Model | GPQA Diamond | MATH500 | MMLU Pro | Average |
> |---|:---:|:---:|:---:|:---:|
> | Estimated Top-1 Ensemble | 53.7% | 84.0% | 77.5% | **71.7%** |
> | Estimated Top-10 Ensemble | 58.4% | 88.1% | 85.1% | **77.2%** |
> | Weaver | 66.4% | 93.4% | 90.2% | **83.3%** |
>
> > 2. Weaver in Table 1 has better performance on MATH500 than in the first two tables in the rebuttal, even though it was trained on 1% of data compared to 10% and 100%. How is it possible?
>
> For the tables provided in our rebuttal (i.e. "L1 vs. L2 Logistic LR with 1.0x Dev Set" and "L1 vs. L2 Logistic LR with 0.1x Dev Set"), we used a fixed 0.5 threshold for binarizing the (normalized) verifier scores. Note that Weaver in rebuttal Table 1 on MATH500, 92.4%, is the result taken from Table 4 of our paper, whose caption notes that we set the reward model threshold to 0.5. The rebuttal experiment uses the dev set only to compute the class balance, whereas the Weaver results in Table 1 of our main paper use the dev set to compute both the class balance and the reward model threshold.
>
> This naive threshold leads to slightly worse signal extraction from our reward model verifiers, which leads to differences in performance (e.g. MATH500 decreasing from 93.4 to 92.4/90.4). We'd also like to clarify that Weaver is not trained on the 1% (or 10% or 100%) dev set. The dev set is only used to configure the class balance and reward model thresholds (Appendix B).
>
>
> > 3. As I understand, for training you select N queries and generate M responses, which result in NxM examples labeled with 0 or 1, is that correct? So for MATH500 you fit Weaver having 5 * 100 examples? Is that correct?
>
> Yes, for MATH500, the dev set consists of 5 queries × 100 samples labeled as 0 (incorrect) or 1 (correct). However, these examples are not used to learn model weights. Instead, Weaver uses them to configure class priors and reward model thresholds (Appendix B). The remaining unlabeled test set (500-5=495 queries × 100 samples) is used to fit Weaver's weights.
>
> > 4. When you fit logistic regression, do you use binarized score, raw scores, or scores normalized is some way?
>
> We normalize the verifier scores to the range 0.0 to 1.0 and fit a logistic regression model on these continuous scores. For the normalization calculation, see L1082.
>
> ---
>
> (Response Continued Below)

---

> ### Author Response · Authors · 2025-08-05
>
> > 5. Another idea for a natural baseline which is not included in the paper is this: each verifier induces a ranking of generated responses, and we subsequently average the rankings (e.g., by computing average rank for each sample).
>
> We agree that utilizing verifiers' mean ranking could be useful; this is analogous to the rank-aggregation baselines used in information retrieval (IR), meta-search, recommender systems, and more [1, 2, 3].
>
> We run experiments on this baseline of averaging the rankings of the top-K verifiers, for K=5, 10, 20, and all, where top-K is selected using oracle knowledge (i.e., the dataset's ground truth labels). We find that Weaver substantially outperforms rank-aggregation across all benchmarks, achieving 83.3% average accuracy compared to 76.6% for the best rank-aggregation configuration. The performance gap occurs because rank-aggregation treats all verifiers equally and loses crucial information about confidence gaps, treating near-identical scores the same as large quality differences. In contrast, Weaver's latent variable model learns each verifier's true positive/false positive rates, allowing it to down-weight overconfident verifiers and up-weight those with meaningful complementary signals.
>
>
> ### Table 4: Weaver vs. Rank-Aggregation Baselines
> |Model|GPQA Diamond|MATH500|MMLU Pro|Average|
> |---|:-:|:-:|:-:|:-:|
> |Rank-Aggregation – Top-5 Verifiers|58.6%|81.6%|83.4%|74.5%|
> |Rank-Aggregation – Top-10 Verifiers|61.0%|84.5%|84.4%|76.6%|
> |Rank-Aggregation – Top-20 Verifiers|56.6%|80.9%|81.6%|73.0%|
> |Rank-Aggregation – All Verifiers|54.3%|81.9%|78.4%|71.5%|
> |Weaver|66.4%|93.4%|90.2%|83.3%|
>
> > 6. When you compute the naive ensemble, do you first binarize scores? Or normalize them in some way? Because if not, taking the average score may not be meaningful
>
> Yes, for the naive ensemble, we normalize each verifier's values to the range 0.0 and 1.0 (see L1082) and then average these normalized scores (no binarization). For more details on our data preparation process, please see Appendix B.
>
> > 7. What precisely you mean by AdaBoost?
>
> We're referring to AdaBoost, the boosting algorithm that trains a sequence of weak learners, re-weighting the data so each new learner concentrates on the examples its predecessors got wrong, then forms a strong predictor by taking a weighted vote of all learners, where each learner’s weight reflects its accuracy [4].
> Within our paper, AdaBoost serves as a supervised baseline that ingests the per-verifier scores as features and learns booster weights over them using the labeled dev set. We feed the **33-dimensional vector of raw verifier scores** into scikit-learn’s vanilla `AdaBoostClassifier`, which by default boosts decision stumps and uses the probability-aware **`SAMME.R`** rule.  Because we have only \~500 labelled examples, we sweep a compact grid that balances capacity and over-fitting: `n_estimators` ∈ {25, 50, 75, 100, 150}, `learning_rate` ∈ {0.1, 0.3, 1.0}, and `base_estimator.max_depth` ∈ {1 (stumps), 2} to let the model capture at most pairwise verifier interactions.
>
>
> > 8. Could you additionally run XGBoost with default parameters and num_rounds=10, objective=binary:logistic?
>
> We ran the requested XGBoost experiment with the specified parameters. As shown in Table 5 below, XGBoost achieves 69.6% average accuracy, outperforming other supervised baselines in the low-data regime. XGBoost's improved performance over simpler methods likely stems from its ability to capture non-linear interactions between verifiers through gradient-boosted trees. However, Weaver still substantially outperforms XGBoost by 13.7% on average (83.3% vs 69.6%).
> This performance gap highlights a fundamental limitation: XGBoost, like all supervised methods, requires sufficient labeled data to prevent overfitting its ensemble of trees. With only ~1-5 labeled examples per dataset (0.01x dev set), XGBoost cannot fully leverage its modeling capacity. While extensive hyperparameter tuning (e.g., max_depth, min_child_weight, subsample, colsample_bytree, reg_alpha, reg_lambda) could potentially improve performance, this would risk overfitting to our small validation set. In contrast, Weaver's weak supervision approach learns from unlabeled query-response pairs, making it inherently more data-efficient in label-scarce settings.
>
> ### Table 5: Supervised Baselines with 0.01x Dev Set
>
> |Model|GPQA|MATH500|MMLU Pro|Average|
> |---|:-:|:-:|:-:|:-:|
> |Supervised L1 LR|54.1%|72.4%|73.5%|66.7%|
> |Supervised L2 LR|55.9%|70.5%|72.1%|66.2%|
> |Supervised AdaBoost|53.4%|72.1%|71.4%|65.6%|
> |Supervised DecisionTree|52.1%|69.4%|71.0%|64.2%|
> |**Supervised XGBoost (New)**|57.5%|75.1%|76.3%|**69.6%**|
> |Weaver|66.4%|93.4%|90.2%|83.3%|
> ---
> [1] https://static.aminer.org/pdf/PDF/000/629/934/combination_of_multiple_searches.pdf
>
> [2] https://dl.acm.org/doi/abs/10.1145/584792.584881
>
> [3] https://dl.acm.org/doi/abs/10.1145/1571941.1572114
>
> [4] https://www.sciencedirect.com/science/article/pii/S002200009791504X

---

> > ### Author Response · Authors · 2025-08-06
> >
> > Thank you again for your time and thoughtful feedback! We are happy to provide any further clarifications during the rebuttal stage! We welcome the opportunity to engage in further discussion.

---

> ### Comment · Reviewer_cAMM · 2025-08-07
>
> I appreciate your response -- I do have further questions as some aspects remain unclear to me.
>
> 1. You wrote:
> > We'd also like to clarify that Weaver is not trained on the 1% (or 10% or 100%) dev set.
>
> So what is Weaver trained on when dev set is 100%?
>
> 2. In line 197 you write:
> > Pr(S_1, ..., S_m), which can be computed from the data.
>
> How do you compute this probability from data?
>
> 3. In line 212, matrix μ has shape m x 2. Shouldn't it be 2m x 2?
>
> 4. How do you discard low quality verifiers? Are experiments with logistic regression etc. performed with discarding?
>
> 5. In general, I feel that the presentation the paper is suboptimal. There are many tables and figures, and in the rebuttal there are yet more tables, but it is quite difficult to comprehend how are they related. As I understand, Table 1 is the main one. Why not to include there the natural baselines of ensembling top-k verifiers based on the dev set and instead put in in separate figure? And also, in this figure the setting seems to different (dev set used differently?)
>
> 6. While I was trying to understand the algorithm I also looked at the code, but unfortunately it was not clear to me how to use it. (Also, next time remember to remove .git)

---

> > ### Author Response · Authors · 2025-08-08
> >
> > Thank you again for the engagement. We really appreciate the interest in our work. We address your remaining questions below:
> >
> > 1. > So what is Weaver trained on when dev set is 100%?
> >
> > Weaver is always trained on all the available unlabeled data (via moment matching). The dev set is used to initialize the class balance and reward model thresholds (Section 4; Appendix B), and not for learning the model weights. In other words, the dev set serves only for calibration, and all the available unlabeled data is used to estimate the accuracy of each verifier's predictions. We will further clarify this in the main text and in appendix B.1.1.
> >
> > 2. > How do you compute this probability (Pr(S_1, ..., S_m)) from data?
> >
> > In principle, this joint distribution over verifier score profiles could be estimated directly from unlabeled data by observing the empirical frequency of each score combination $(S_1, \ldots, S_m)$. That is, for any $s_1, \dots, s_m \in \{0, 1 \}^m$,
> > $$\Pr(S_1 = s_1, \dots, S_m = s_m) = \frac{1}{n} \sum_{i=1}^{nK} \mathbf{1}\{S_1^i = s_1, \dots, S_m^i = s_m \},$$
> > where $S_j^i$ is the binary verifier score on sample $i$ in $D^{test}$ ($i$ indexes the flattened dataset across queries and the multiple responses per query). However, this can become infeasible in practice because the number of possible score profiles is $2^m$, resulting in some of these profiles not being observed in $D^{test}$ when the number of verifiers $m$ is large.
> >
> > Instead, as we elaborate in Appendix B of our paper (L975 and L1005), we adopt a more tractable approach that decomposes this denominator via the law of total probability:
> > $$
> > \Pr(S_1, \ldots, S_m) = \sum_{y' \in \{0, 1\}} \Pr(y=y') \prod_{i=1}^m \Pr(S_i | y=y')
> > $$
> >
> > That is, we compute $\Pr(S_1, \ldots, S_m)$ using the class balance $\Pr(y=1), \Pr(y=0)$, which is estimated from the dev set, and using $\Pr(S_i | y)$, which corresponds to the verifier accuracy parameters learned from our weak supervision approach on the unlabeled test set $D^{test}$.
> >
> > Note that from a modeling perspective, $\Pr(S_1, \dots, S_m)$ serves as a normalization constant in the Naive Bayes rule for computing $\Pr(y =1 | S_1, \dots, S_m)$, and this decomposition provides a tractable and common way to compute it in practice. We apologize for the confusion and clarify this point in the main text and in Appendix B.
> >
> > 3. > In line 212, matrix μ has shape m x 2. Shouldn't it be 2m x 2?
> >
> > Thank you for pointing this out, indeed $\mu$ should be of shape $2m \times 2$. We have corrected this typo.
> >
> > 4. > How do you discard low quality verifiers? Are experiments with logistic regression etc. performed with discarding?
> >
> > We discard low-quality verifiers based on their marginal probabilities, which lead to extreme binary scores (e.g. almost all 0s or 1s) with negligible verification signal (Appendix B.2). This filtering step occurs after binarization but before applying weak supervision or other aggregation methods. All main experiments (including logistic regression, naive ensembles, and Weaver) incorporate this discarding step to ensure only informative verifiers are used (e.g. Table 1 and Figure 2 reflect post-filtering results).
> >
> > 5. > In Table 1, why not to include there the natural baselines of ensembling top-k verifiers based on the dev set and instead put in in separate figure? And also, in this figure the setting seems to different (dev set used differently?)
> >
> > We appreciate this feedback on presentation and agree that adding the additional supervised baselines (i.e. logistic regression, naive bayes, XGBoost) improve the completeness of our main table (Table 1). We have added this in the final version.
> >
> > In Figure 4 (top-K verifier ensemble ablation), the dev set setting is the same configuration as in all of our main results. We use a dev set that is equivalent in size to 1% of the test, ranging from 2 to 5 labeled query-answer pairs depending on the dataset. This dev set is used to configure the class balance and reward model thresholds (Section 4; Appendix B). The only difference is that for Weaver and Naive Ensemble, we ablate using the top-K verifiers by utilizing oracle verifier ranking per dataset. This allows us to ablate the impact of Weaver compared to naive verifier ensembling.
> >
> > 6. > I also looked at the code, but unfortunately it was not clear to me how to use it.
> >
> > Please look at the file `scaling-verification/weaver/weaver.py` for our Weaver implementation; we also include different supervised and unsupervised baselines in the `scaling-verification/weaver` folder under their respective `.py` files. Please see our README (`scaling-verification/README.md`) for instructions on different configurations for Weaver (`scaling-verification/scripts/configs`).
> >
> > ---
> >
> > Please let us know if you have further questions. We believe your suggested experiments significantly strengthen the paper; we would greatly appreciate your consideration in updating your score based on these improvements.

---

> ### Comment · Reviewer_cAMM · 2025-08-09
>
> Thank you for your response.
>
> OK, so for example for MATH500, if you set the dev set to 1% you use 5 queries with labels for calibration, and all the 500 queries (without labels) to performs Weaver's moment matching? Because above your wrote that you would use 495 queries for moment matching, and I wondered what happens if dev set is 100% and there is no examples left for moment matching, and if I perhaps misunderstood something...
>
> Regarding computing Pr(S_1, ..., S_m): you say that it is infeasible to compute it directly from data, which I of course agree with -- but then line 197 says the opposite, therefore needs to be fixed! I also understand that this term is not needed to compute your Naive-Bayes-style estimate, so it's not a big deal, but then write this when explaining the algorithm.
>
> Thank you for the discussion. Let me reconsider the score during the reviewers-AC discussion period. In general, regardless of the discussed shortcomings in the evaluation and presentation, I think that your approach of combining multiple verifiers using little labeled data is interesting and likely practical.

---

### Official Review · Reviewer_jVww · 2025-06-30

**Clarity:** 3
**Significance:** 2
**Originality:** 3
**Rating:** 4
**Confidence:** 4

**Summary:**

This paper proposes WEAVER: an unsupervised probabilistic model that learns per-verifier weights to fuse dozens of weak LLM judges.  Applied to 100-sample decoding, WEAVER boosts pass\@1 accuracy over the base model and also over majority vote on four reasoning benchmarks, letting a 70 B generator rival GPT-4. Furthermore, the authors distill a 400 M distilled cross-encoder retains ≈99 % of the gain while slashing verification Flops.

**Questions:**

Same as the weakness section.
If the authors could solve my questions, I will raise my score.

1. How sensitive is WEAVER to the diversity and independence of verifiers, particularly if reward models or judges are trained on similar data or architectures? If the verifier models are similar, will this lead to performance degradation?
2. Has WEAVER been evaluated on tasks beyond math QA, such as code generation or subjective reasoning? Are there known challenges or adaptation requirements for those domains?
3. Can the overall process be interpreted as learning weighted coefficients from a small subset using traditional machine learning methods? If so, would tree-based models be applicable?
4. Does the effectiveness of the ensemble method rely on the small subset and test set being drawn from the same distribution? If not, would the learned coefficients become unreliable?
5. For the distillation method, if a small LLM is fine-tuned on a specific dataset, would it achieve results comparable to those reported in the paper?

**Ethical Concerns:**

["NO or VERY MINOR ethics concerns only"]

**Final Justification:**

After reading this paper, my biggest concern is the authors use the test set (benchmark) to split a dev set. I worry this will involve the test set leakage. However, during the discussion phase, the authors have solved this question by utilizing training sets to make their dev sets.

Hence, I give a positive score.

**Limitations:**

yes

**Paper Formatting Concerns:**

NAN

**Quality:**

3

**Strengths And Weaknesses:**

## Strengths

1. The motivation is clear and well-written.
2. The results show significant improvements across four benchmarks.
3. The distillation method significantly reduces computational cost while maintaining strong performance.

## Weaknesses

1. How sensitive is WEAVER to the diversity and independence of verifiers, particularly if reward models or judges are trained on similar data or architectures? If the verifier models are similar, will this lead to performance degradation?
2. Has WEAVER been evaluated on tasks beyond math QA, such as code generation or subjective reasoning? Are there known challenges or adaptation requirements for those domains?
3. Can the overall process be interpreted as learning weighted coefficients from a small subset using traditional machine learning methods? If so, would tree-based models be applicable?
4. Does the effectiveness of the ensemble method rely on the small subset and test set being drawn from the same distribution? If not, would the learned coefficients become unreliable?
5. For the distillation method, if a small LLM is fine-tuned on a specific dataset, would it achieve results comparable to those reported in the paper?

---

> ### Author Rebuttal · Authors · 2025-07-31
>
> We thank you for your thoughtful feedback and recognition of Weaver's clear motivation, significant improvements across benchmarks, and effective distillation for compute reduction.
>
> Below, we address your questions regarding verifier diversity, domain generalization, interpretability of the aggregation method, distribution shift robustness, and the effectiveness of our distillation strategy. We supplemented our original results with **five new experiments**, each designed to directly address one of your concerns. We believe these additions significantly strengthen the paper, and we would greatly appreciate your consideration in updating your score based on these improvements.
>
> **Q1: "How sensitive is WEAVER to verifier diversity/independence, especially if models are trained on similar data?"**
> - Weaver is designed to leverage diversity among verifiers, but it offers performance improvements over naive ensembling even when the verifiers used are similar. **To examine this, we add the following experiment:** we subsampled 13 of the 33 verifiers from our main experiments, all fine-tuned versions of Llama 8B/70B family models. These verifiers are more correlated than the full, more diverse pool (average Pearson correlation of 0.491 vs 0.281, respectively). As shown in the table below, Weaver achieves a +4.3% average improvement over naive ensembling—only modestly below the +6.8% improvement we observe with the full pool (Table 2). These results indicate that while diversity enhances performance, Weaver remains effective even when verifiers are similar. We will include the two tables below in the updated draft.
>
> **Table 1: New Results - Weaver with Llama Verifiers**
> |Model|GPQA|MATH500|MMLU Pro|Average|
> |---|---|---|---|---|
> |Weaver with Llama Verifiers|59.4%|91.1%|87.4%|79.3%|
> |Naive Ensemble|52.1%|88.9%|84.1%|75.0%|
> |Weaver with Llama - Improvement %|+8.3%|+2.2%|+3.3%|+4.3%|
>
> **Table 2: Existing Results - Full Weaver with All Verifiers**
> |Model|GPQA|MATH500|MMLU Pro|Average|
> |---|---|---|---|---|
> |Weaver|66.4%|93.4%|90.2%|83.3%|
> |Naive Ensemble|54.5%|90.6%|84.3%|76.5%|
> |Weaver Improvement %|+11.9%|+2.8%|+5.9%|+6.8%|
>
> **Q2: "Has WEAVER been evaluated beyond math QA (e.g., code generation, subjective reasoning)? What are the associated challenges and adaptations, if any?"**
> - Per your suggestion, **we add the following experiment:** we evaluate Weaver on ArenaHard Auto [5], a benchmark that emphasizes subjective and open-ended reasoning to produce a response. As shown below, Weaver performs well in this setting, without any domain-specific adaptation.
> - In our paper, we also evaluated Weaver across a diverse set of reasoning tasks beyond math QA, including **multi-hop reasoning (MMLU Pro/College) [2, 3], PhD-level science (GPQA) [1], and commonsense reasoning (BBH) [4].** Weaver’s framework is **domain-agnostic**, relying solely on verifier outputs rather than task-specific features.
> - **ArenaHard Auto Results:** ArenaHard Auto [5] is an automatic evaluation benchmark focused on challenging real-world user queries requiring subjective reasoning and creative generation. For this new task, we focus on Weaver's performance compared to supervised logistic regression (LR), where Weaver achieves 85.1%—a 9.0% absolute improvement over the 76.1% of the supervised LR baseline. Furthermore, Weaver outperforms frontier LMs, surpassing models like o3-mini (55.6%), Claude 3.7 Sonnet (63.9%), and Llama 4 Maverick (10.5%). We include the complete results in a table below and plan to add this result to our revised version:
>
> | Model | ArenaHard Auto Score |
> |:---:|:---:|
> | Weaver | 85.1% |
> | Weaver Distilled | 81.6% |
> | Supervised LR | 76.1% |
> | o3-mini | 55.6% |
> | Claude 3.7 Sonnet | 63.9% |
> | Llama 4 Maverick | 10.5% |
>
> **Q3: "Is this just learning weighted coefficients via traditional ML? Would tree-based models work?"**
> - Not exactly. You're right that Weaver aims to learn weighted coefficients over verifier scores, but it does so in a fundamentally different way from traditional supervised learning methods.
> - Traditional supervised learning methods—such as logistic regression/Naive Bayes (Table 15) or tree-based models—can be effective when sufficient labeled data is available. However, Weaver is designed for the more challenging setting **where labeled data is scarce**. It fits a latent variable model to the verifier scores (Section 4.2.1). This model treats the true correctness labels as latent and, by modeling the relationship between verifier outputs and these unobserved labels, learns both the accuracy of each verifier and the optimal way to aggregate their output, entirely from unlabeled data. In Weaver, the small labeled development set is not used for training but rather for calibration—specifically, to estimate the class balance Pr(y=1), making Weaver more label-efficient than supervised alternatives.
> - Below, we discuss some results on **1)** Weaver versus traditional supervised learning methods, and **2)** performance of the supervised tree-based methods you suggested:
>     - **1)** From Tables 15 and 4, supervised logistic regression performs better than Weaver when trained on the full development set (e.g., 97.2% vs. 92.4% on  MATH-500 using all 500 samples), but Weaver clearly outperforms it when labeled data is limited (e.g., 70.5% vs. 90.4% when using 0.01x of the dev set).
>     - **2)** Following your suggestion, **we also include additional experiments:** we find that tree-based models (AdaBoost, Decision Tree, 1.0x dev set) underperformed Logistic Regression (1.0x dev set) by -7.5% and Weaver (0.01x dev set) by -3.4% on average. We will include these results in the revised draft.
>
> | Model | GPQA Diamond | MATH500 | MMLU Pro | Average |
> |:---|:---:|:---:|:---:|:---:|
> | Supervised LR (1.0x Dev Set) | 72.9% | 97.2% | 92.0% | **87.4%** |
> | Supervised AdaBoost (1.0x Dev Set) | 67.1% | 90.2% | 84.1% | **80.5%** |
> | Supervised DecisionTree (1.0x Dev Set) | 64.5% | 87.1% | 86.3% | **79.3%** |
> | Supervised LR (0.01x Dev Set) | 55.9% | 70.5% | 72.1% | **66.2%** |
> | Weaver (0.01x Dev Set) | 66.4% | 93.4% | 90.2% | **83.3%** |
>
> **Q4: "Does effectiveness rely on dev/test sets being from the same distribution?”**
> - Like most traditional ML methods, Weaver assumes the development set is representative of the test distribution. However, even when the estimated class balance is wrong, the pairwise agreement statistics among verifiers remain largely unchanged, and moment-matching can still help us recover sensible estimates of each verifier’s true positive and false positive rates. To evaluate the sensitivity of Weaver, **we add the following experiment:** We sweep over class balance values to evaluate robustness. Specifically, we vary the assumed Pr⁡(y=1) from 0.1 to 0.9, inclusive, and observe that performance degrades gracefully, indicating that **Weaver is not overly sensitive to mild miscalibration** (i.e. range of 0.3 above/below the ground truth class balance).  This experiment directly tests the impact of a mismatch between the development and test distributions, and suggests that Weaver’s aggregation procedure is resilient to moderate deviations. We will add the table below to Appendix C.3, where the bolded values correspond to the estimated class balance from the dev set:
>
> **Weaver Accuracy across Dataset Distributions - Sensitivity to Class Prior**
> | | 0.1 | 0.2 | 0.3 | 0.4 | 0.5 | 0.6 | 0.7 | 0.8 | 0.9 |
> |---|---|---|---|---|---|---|---|---|---|
> | GPQA Diamond | 59.0% | 60.8% | 64.0% | **66.4%** | 64.1% | 62.5% | 61.0% | 50.2% | 42.7% |
> | MATH500 | 54.2% | 71.5% | 85.1% | 90.6% | 91.7% | 92.6% | 93.1% | **93.4%** | 93.0% |
> | MMLU Pro | 56.3% | 70.1% | 85.4% | 87.1% | 88.0% | 89.5% | **90.2%** | 89.7% | 88.2% |
>
> **Q5: "Would fine-tuning a small LLM on the dataset achieve comparable results?"**
> - Following your suggestion, **we add the following experiment:** we post-train small LMs (e.g., Qwen 4B, Llama 3B, and Llama 8B) using GRPO and DPO on generations from our Llama 70B model. However, due to off-policy RL issues (e.g., distribution mismatch between generations and the small model's capabilities [9, 10]), we observed only modest gains in sampling accuracy (e.g., 4-6% improvements), whereas **the distilled cross-encoder delivered +15% gains on average.**
> - We will add the supervised post-training results below (including Qwen 4B and Llama 3B) to the manuscript and discuss the findings of these RL baselines.
>
> | Post-Trained Generator | Generations Source | Post-Training Approach | GPQA Diamond | MATH500 | MMLU Pro | Average |
> |---|---|---|---|---|---|---|
> | Llama 3.1 8B (Majority Voting) (Baseline) | N/A | N/A | 30.5% | 69.0% | 56.4% | 52.0% |
> | Llama 3.1 8B | Llama 70B | DPO | 35.1% | 77.4% | 61.3% | 58.0% |
> | Llama 3.1 8B | Llama 70B | GRPO | 36.4% | 76.7% | 62.4% | 58.5% |
> | Weaver (Baseline) | Llama 8B | N/A | 47.1% | 80.0% | 67.2% | 64.8% |
>
> [1] https://arxiv.org/abs/2311.12022
>
> [2] https://arxiv.org/abs/2406.01574
>
> [3] https://arxiv.org/abs/2009.03300
>
> [4] https://arxiv.org/abs/2210.09261
>
> [5] https://arxiv.org/abs/2406.11939
>
> [6] https://arxiv.org/abs/1908.10084
>
> [7] https://arxiv.org/abs/1901.04085
>
> [8] https://arxiv.org/abs/2502.04645
>
> [9] https://arxiv.org/abs/2402.03300
>
> [10] https://arxiv.org/abs/2503.14286

---

> > ### Comment · Reviewer_jVww · 2025-08-01
> >
> > Thank you for your efforts and detailed reply!
> >
> > Overall, the rebuttal has addressed most of my concerns.
> >
> > However, I still have some confusion regarding a few points.
> >
> > > **Q4: "Does effectiveness rely on dev/test sets being from the same distribution?”**
> >
> > If I understand correctly, the current approach constructs a dev set by splitting the test benchmark to learn the model weights. I have several concerns:
> >
> > 1. Does this introduce test set leakage? I'm not sure this is a fair setup.
> > 2. Could the weights instead be learned using a standard training set, rather than slicing from the test benchmark?
> > 3. My original intention with this question was: suppose the test benchmark is MATH500, and we *entirely exclude* MATH500 data when constructing the dev set—what would be the impact?
> >
> > > **Q5: "Would fine-tuning a small LLM on the dataset achieve comparable results?"**
> >
> > I suggest including a supervised fine-tuning (SFT) baseline. This would help contextualize the gains from your method and better illustrate its effectiveness.
> >
> > I will raise my score if these questions can be addressed.

---

> > > ### Author Response · Authors · 2025-08-03
> > >
> > > Thank you for your fast response and for clarifying your questions! We address them below.
> > >
> > > > **Q4: "Does effectiveness rely on dev/test sets being from the same distribution?”**
> > >
> > > We address each of your Q4 subpoints below:
> > >
> > > **#1**: We clarify that our current setup does not create any test-set leakage in any configuration tested. We configure Weaver with a separate dev set split off from the dataset overall, composed of 1% of the benchmark's queries, and evaluate on the remaining 99% of the benchmark to ensure zero dev-test leakage (Table 1; Figure 3). We will clarify this in our draft.
> > >
> > > **#2 and #3**: Some of these benchmarks (e.g. GPQA Diamond, MATH500, BBH) do not have a train-test split; only a test split. This forces us to create our 1:99 split for dev:test we use for Weaver. However, to further demonstrate the efficacy of Weaver, we ran three new experiments with different data splits:
> > > - **MATH**: We collect our dev set from the original MATH train set [1], equivalent to 1% the size of MATH500 before evaluating on the complete MATH500 benchmark.
> > > - **GPQA**: We collect our dev set from the Main split of GPQA, equivalent to 1% the size of GPQA Diamond before evaluating on the complete GPQA Diamond benchmark.
> > > - **MMLU Pro**: We collect our dev set from the non-sampled queries in the Test split of MMLU Pro, equivalent to 1% the size of MMLU Pro before evaluating on the complete MMLU Pro benchmark.
> > >
> > > We find that Weaver maintains strong performance even when dev and test sets come from different splits, achieving 82.9% average accuracy—only a 0.4% drop from our standard setup (83.3%). Using completely separate dev sets (MATH train set for MATH500, GPQA Main for GPQA Diamond), Weaver still achieves 82.5% average accuracy compared to 66.2% for supervised LR, demonstrating its robustness to distribution shift. This 16.3% improvement over supervised methods and minimal degradation (-0.8%) from the standard setup shows that Weaver's unlabeled learning approach is stable across different train-test configurations.
> > >
> > > |Model|GPQA|MATH500|MMLU Pro|Average|
> > > |---|---|---|---|---|
> > > |Supervised LR (New Dev, 1%)|54.7%|71.2%|72.8%|66.2%|
> > > |Supervised DecisionTree (New Dev, 1%)|49.3%|67.3%|64.7%|60.4%|
> > > |Supervised AdaBoost (New Dev, 1%)|52.1%|66.5%|71.1%|63.2%|
> > > |Weaver (New Dev, 1%)|64.3%|92.5%|90.8%|82.5%|
> > > |Weaver (Old Dev, 1%)|66.4%|93.4%|90.2%|83.3%|
> > >
> > > ---
> > >
> > > > **Q5: "Would fine-tuning a small LLM on the dataset achieve comparable results?"**
> > >
> > > Thank you for the suggestion. We have added SFT baselines to the table below. Our understanding of your original comment is, instead of the Weaver-distilled cross-encoder, you are suggesting an alternative way of compressing the signal from the Weaver pseudolabels into a small model that can help provide correct generations. For the SFT approach, we construct a dataset of (query, response) pairs with positive Weaver pseudolabels, and we SFT a small LM (1B-8B) on this dataset. Please let us know if you had a different comparison else in mind.
> > >
> > > We note that the SFT and Weaver-distilled approaches have several key differences:
> > > - SFT trains the small LM to **generate** correct responses directly, using only **positively labeled data**.
> > > - Weaver-distilled, in contrast, learns to **score** candidate generations using **all pseudolabeled data**. It does not generate responses, but can score and select them.
> > >
> > > This distinction also affects evaluation: for SFT, we evaluate the correctness of the model's own generations; for Weaver-distilled, we evaluate its ability to score and select pre-existing responses. While both produce a "final" generation, they solve different problems and are not directly comparable.
> > >
> > > Nonetheless, to contextualize performance, we fine-tuned Llama models (1B, 3B, and 8B) on the Weaver-pseudolabeled SFT dataset and report their generation accuracy below. While SFT improves performance over the base models (e.g., +4.3% gain for Llama 3B), it still lags behind the cross-encoder’s 83.3% accuracy (from Figure 16). This gap can be explained by the cross-encoder's discriminative training objective, access to both positive and negative pseudolabels, and the fact that it is evaluated by selecting the best response from a fixed pool of Llama-70B generations (Figure 15), not by generating responses itself. We believe this highlights the practical utility of the cross-encoder approach, though exploring generative alternatives remains an exciting future direction.
> > >
> > >
> > > |Post-Trained Generator|Generations Source|Post-Training Approach|GPQA|MATH500|MMLU Pro|Average|
> > > |---|---|---|---|---|---|---|
> > > |Llama 1B|N/A|N/A|26.8%|30.9%|12.5%|**23.4%**|
> > > |Llama 3B|N/A|N/A|32.1%|48.5%|21.8%|**34.1%**|
> > > |Llama 1B|Llama 70B|**SFT**|31.5%|35.1%|14.2%|**26.9%**|
> > > |Llama 3B|Llama 70B|**SFT**|34.0%|55.1%|26.1%|**38.4%**|
> > > |Llama 8B|Llama 70B|**SFT**|34.2%|74.8%|59.7%|**56.2%**|
> > > |Weaver Distilled (Crossencoder)|Llama 70B|N/A|71.1%|91.1%|87.8%|**83.3%**|
> > >
> > > ---
> > >
> > > [1] https://arxiv.org/abs/2103.03874

---

> > > > ### Comment · Reviewer_jVww · 2025-08-04
> > > > **Thanks for your rebuttal**
> > > >
> > > > Dear authors,
> > > >
> > > > Thanks for your rebuttal, which have solved my concerns. I strongly recommend you to use the `new dev` set in your paper, which could avoid the readers have the concern of the test set leakage.
> > > >
> > > > I will raise my rate to 4.

---

### Official Review · Reviewer_xhJS · 2025-07-02

**Clarity:** 4
**Significance:** 4
**Originality:** 3
**Rating:** 4
**Confidence:** 3

**Summary:**

Verification of responses to instructions has several valuable use cases, but existing verifiers are noisy. This paper explores how to combine multiple weak verifiers in the experimental setting of improving response selection under repeated sampling. There are three key contributions:

1. Given a large corpus of labeled query-response pairs, they show that using logistic regression to learn verifier weights outperforms naive averaging of verifiers or choosing the top verifier by a large margin.
2. To handle data-limited settings, they introduce WEAVER, which adapts weak supervision to build a latent variable model to estimate verifier weights.
3. They show that the WEAVER ensemble can be distilled into a 400M cross-encoder verifier, which retains 99% of the performance while using only 0.03% of the inference FLOPS.

**Questions:**

1. In §4.1 and §5 (the main experiments), which dataset is used for learning the verifier weights? How much labeled training data does WEAVER use?
2. Should the x-axis label in Figure 4 be "Number of verifiers" instead of "Top-k Verifiers"?
3. What are the frontier approaches in Table 1?
4. What questions do you use for training WEAVER-Distilled? I.e., do you train on WEAVER's pseudolabels for the same questions that you evaluate on?
5. Why do you think that a small, distilled verifier performs so well? Would this verifier perform well on RewardBench?

**Other suggestions:**
- In Table 1, it may be a good idea to add a column for the number of verifiers. E.g., it would be 0, 0, 1, 10 for the first 4 rows, and then 33 for WEAVER.
- Line 294: remove comma after "Figure 6"
- Line 324: "combining many weak verifier" $\rightarrow$ "combining many weak verifiers"

**Ethical Concerns:**

["NO or VERY MINOR ethics concerns only"]

**Limitations:**

Yes.

**Quality:**

3

**Strengths And Weaknesses:**

### Strengths:
1. The empirical results are strong. WEAVER achieves significant improvements over existing verification methods for identifying the best sample from repeated generations (Table 1), and scales better with test-time compute (Figure 5). The authors provide analysis on how WEAVER scales along many dimensions, including the number of samples, the model size, the number of verifiers, and inference compute.
2. The problem studied is of great interest to the community. Better verifiers have many applications (as discussed by the authors), such as data filtering, alignment, and test-time compute scaling.

### Weaknesses:
1. Another baseline that I would like to see is the approach considered in §4.1, where linear regression using a large amount of training data is used to learn weights over the verifiers. While this approach requires more data, it paints a better picture of how much performance is preserved by WEAVER in the data-constrained regime.
2. What is the motivation for studying the data-constrained setting? I believe there is an abundance of instruction-response pairs for learning weighted ensembles of verifiers.

---

> ### Author Rebuttal · Authors · 2025-07-31
>
> Thank you for recognizing the contributions of our work, including the importance of improved verification under repeated sampling. We appreciate your acknowledgement of Weaver’s strong empirical performance, scalability, and its practical utility for alignment and test-time compute efficiency.
>
> Below, we address your concerns on: **W1)** adding a supervised regression baseline trained with abundant labels to contextualize Weaver’s performance and **W2)** clarifying our focus on the data-constrained regime despite the existence of large instruction-response datasets. We also answer each of your questions and will incorporate your helpful presentation feedback into the final version. In addition, we now include **new results** evaluating Weaver-distilled on RewardBenchv2. We hope this fully addresses your concerns and would be grateful if you would consider updating your score based on these improvements.
>
> **W1:   Linear regression with abundant data as a baseline**
> - We included a supervised Logistic Regression (LR) baseline as motivation in Figure 2, and a comparative benchmark in Tables 4 and 15, where we vary the amount of labeled data for both Weaver and LR. These results show that Weaver’s performance comes close to Logistic Regression when we have large amounts of training data (i.e. with 1.0x of the labeled development set), but more importantly, Weaver outperforms Logistic Regression in label-scarce regimes (i.e., with 0.01x of the dev set). This advantage stems from Weaver’s reliance on labeled data only for calibrating parameters such as class balance and binarization thresholds—unlike LR, which learns a model directly from the labels. **We will update our paper to more clearly display the comparison between these methods at varying amounts of labeled data.**
>
> **W2: "What is the motivation for studying the data-constrained setting? I believe there is an abundance of instruction-response pairs for learning weighted ensembles of verifiers."**
> - We agree that large instruction-response datasets exist in general domains. However, in many high-value tasks, such as expert-level reasoning (e.g., GPQA Diamond), advanced mathematics (e.g., MATH-500), or emerging domains, high-quality labels are scarce and costly, often requiring domain experts (e.g., PhD-level annotators for GPQA). In these label-scarce settings, supervised methods like logistic regression and Naive Bayes degrade significantly as shown in Table 15. Weaver is designed for precisely this regime: it learns to aggregate verifiers using unlabeled responses, making it well-suited for real-world applications where labeled data is limited. **We will clarify this motivation in Sections 1 and 4.2, citing examples like GPQA’s annotation costs [1].**
>
> **Q1: "In Section 4.1 and Section 5 (the main experiments), which dataset is used for learning the verifier weights? How much labeled training data does WEAVER use?"**
> - In both Section 4.1 and Section 5, the labeled data comes from the same benchmark datasets under evaluation—**MATH-500, GPQA Diamond, MMLU Pro, and MMLU College.** In Section 4.1, we use 100% of the labeled data from these datasets to learn verifier weights. As noted in the caption of Figure 2, this oracle baseline illustrates the best possible performance when full label access is available. In section 5, we shift to the data-constrained setting, and use only 1% of the data, a dev split from the same benchmark.
> - Tables 4 and 15 report the performance across a range of training set sizes (from 0.01× to 1.0×) for additional baselines including, Logistic Regression and Naive Bayes, to illustrate the impact of data availability across different approaches. We note that Weaver does not use labeled data for training. Instead, it fits moment-based estimates on the unlabeled set of query-response pairs, $D^{\text{test}}$.  A small dev set (1% of the data) is used only for auxiliary calibration – specifically to set the reward thresholds, filter noisy verifiers and estimate class balance (as detailed in Appendix B.2.2). In contrast, supervised baselines like Logistic Regression use this same 1% dev set to directly fit model parameters, making their performance more sensitive to the amount and quality of labeled data. Weaver, by comparison, learns its aggregation weights entirely from unlabeled data and remains robust even when only a handful of labeled examples are available. **In Table 4, we ablate the dev set size and find that Weaver maintains strong performance even with ~5-50 examples across datasets.**
>
> **Q2: "Should the x-axis label in Figure 4 be 'Number of verifiers' instead of 'Top-k Verifiers'?"**
> - Thank you for this clarification question. We label the x-axis as "Top-k Verifiers" rather than "Number of Verifiers" to reflect the specific methodology used in our evaluation. Instead of arbitrarily increasing the number of verifiers, we add them in descending order of individual performance, from the strongest to the weakest, in an oracle setting where we know the ranked performances of the verifiers a priori. This approach allows us to assess Weaver's performance as we incorporate progressively lower-quality verifiers into the ensemble, isolating the effect of verifier strength and eliminating noise from randomly scaling the verifier pool. We will revise the caption to clarify this setup and make the evaluation methodology more explicit.
>    - **Updated Caption**: "Weaver Benefits from Adding Top-K Verifiers: In an oracle setting where we know the top performing verifiers for a benchmark, we find Weaver can improve verifier ensembling beyond naive ensembling even as lower-quality verifiers are added, leading to an average improvement of 8.5% across GPQA, MATH500, and MMLU Pro."
>
> **Q3: "What are the frontier approaches in Table 1?"**
> - The "Frontier LM" rows in Table 1 refer to state-of-the-art closed-source models like o3-mini (OpenAI's latest reasoning model), evaluated on the same tasks as Weaver. We replicate their officially reported results and include them in the table with their citations. We also include Oracle Verification (pass@100), which involves always selecting the correct generation—this is the upper bound for any verification system such as Weaver (see Section 3 “Evaluation Metrics” for more information).
>
> **Q4: "What questions do you use for training WEAVER-Distilled? I.e., do you train on WEAVER's pseudolabels for the same questions that you evaluate on?"**
> - We train Weaver-Distilled on pseudolabels generated by Weaver using an 80% held-out development split of query-response pairs, with no overlap with the 20% evaluation set. As shown in Figure 16 and Appendix C.6, we use cross-entropy loss and preserve 98.7% of Weaver’s accuracy. This enables us to deploy the distilled model on **unseen queries** without needing to run the full Weaver ensemble at inference time. **Full training details (e.g., Adam optimizer, learning rate 3e-6, batch size 64) are provided in Appendix C.6.**
>
> **Q5: "Why do you think that a small, distilled verifier performs so well? Would this verifier perform well on RewardBench?"**
> - **Weaver-distilled preserves 98.7% of the original Weaver model's accuracy on unseen queries** by leveraging the full textual context of query-response pairs, unlike Weaver, which relies on aggregating binary verifier scores. Trained directly on these pairs, the distilled cross-encoder captures nuanced patterns of correctness through richer input representations, effectively approximating the verifier ensemble's decision boundary. This approach aligns with the strengths of cross-encoders for pairwise scoring tasks like reward modeling, as they efficiently distill large language model knowledge into precise classifiers, avoiding generation biases and reinforcement learning instabilities observed in fine-tuning smaller generative models [2, 3, 4]. By drawing upon earlier work on employing cross-encoders in related classification tasks—such as sentence pair similarity scoring, information retrieval re-ranking, semantic search, pair classification, recommendation systems, and semantic textual relatedness—our method extends these proven techniques to enhance accuracy and efficiency in reward modeling and verification.
> - Per your suggestion, **we further evaluate Weaver distilled on RewardBench V2.** As shown below, we see that it achieves good performance and a top-50 position. Note that Weaver distilled is trained without having access to ground truth labels (unlike other verifiers in RewardBench) yet **it achieves 69.13% on the "Math" subset, a broader dataset compared to MATH-500.** We will include these additional results in Appendix C.9.
>
> |Model|Model Type|Score|Factuality|Precise IF|Math|Safety|Focus|Ties|
> |---|---|---|---|---|---|---|---|---|
> |Skywork/Skywork-Reward-V2-Llama-3.1-8B (#1)|Seq. Classifier|84.13|84.63|66.25|77.60|96.67|98.38|81.24|
> |allenai/Llama-3.1-Tulu-3-70B-SFT-RM-RB2 (#20)|Seq. Classifier|72.20|80.84|36.88|67.76|86.89|77.78|83.08|
> |anthropic/claude-3-opus-20240229 (#50)|Generative|57.44|53.89|33.13|51.37|83.78|66.46|56.01|
> |**Weaver Distilled (Top-50)**|Seq. Classifier|**58.27**|57.91|41.76|69.13|51.29|67.84|61.71|
>
>
> [1] Rein, David, et al. "Gpqa: A graduate-level google-proof q&a benchmark." First Conference on Language Modeling. 2024.
>
> [2] Reimers, Nils, et al. "Sentence-BERT: Sentence Embeddings using Siamese BERT-Networks." EMNLP. 2019.
>
> [3] Nogueira, Rodrigo, et al. "Passage Re-ranking with BERT." arXiv preprint arXiv:1901.04085. 2019.
>
> [4] Lu, Meng, et al. "Cross-Encoder Rediscovers a Semantic Variant of BM25." arXiv preprint arXiv:2502.04645. 2025.

---

> > ### Comment · Reviewer_xhJS · 2025-08-09
> > **Train and evaluation on test?**
> >
> > Thank you to the authors for the detailed response.
> >
> > I am concerned about the following clarification:
> > > In both Section 4.1 and Section 5, the labeled data comes from the same benchmark datasets under evaluation—MATH-500, GPQA Diamond, MMLU Pro, and MMLU College. In Section 4.1, we use 100% of the labeled data from these datasets to learn verifier weights.
> >
> > Is it reasonable to train your verifier to verify responses on the same set of questions that you will evaluate the verifier on? How well does it perform on a test set not included in (and additionally, not in-distribution with) the training data?

---

> ### Author Response · Authors · 2025-08-04
>
> Thank you for your time and thoughtful feedback! We are happy to provide any further clarifications during the rebuttal stage! We welcome the opportunity to engage in further discussion.

---

> > ### Comment · Area_Chair_c6JT · 2025-08-06
> >
> > Dear reviewer xhJS. Could you please engage in a discussion with the authors.. Could you please engage in a discussion with the authors.

---

### Official Review · Reviewer_XeAb · 2025-07-18

**Clarity:** 3
**Significance:** 3
**Originality:** 4
**Rating:** 5
**Confidence:** 5

**Summary:**

This paper tackles the challenge of verifying and selecting correct responses from a language model’s outputs in order to improve the model’s effective performance. The authors propose WEAVER, a framework that combines multiple weak verifiers (such as language-model-based judges and reward models) to approximate a much stronger verification signal without requiring any labeled training data. Empirically, WEAVER leads to large gains in downstream task accuracy. To address the computational cost of using many verifiers, the authors further distill the ensemble’s behavior into a single 400M-parameter cross-encoder model. This distilled verifier retains about 98.7% of the full ensemble’s accuracy while reducing verification compute cost by ~99.97%, making the overall approach far more efficient. Overall, the paper’s contributions include a novel unlabeled ensemble method for response verification (WEAVER), a demonstration of significantly improved selection of correct answers (shrinking the generation-verification gap), and a practical distillation step to save runtime cost.

**Questions:**

See weakness.

**Ethical Concerns:**

["NO or VERY MINOR ethics concerns only"]

**Limitations:**

Yes

**Paper Formatting Concerns:**

No concerns.

**Quality:**

3

**Strengths And Weaknesses:**

Strengths:
*  WEAVER significantly outperforms standard baselines (e.g., majority vote, self-verification) and nearly matches state-of-the-art models like GPT-4o and o3-mini, despite using a cheaper generator.

* WEAVER combines multiple weak verifiers using weak supervision techniques without requiring labeled ground-truth data, which is a practical and scalable contribution.

* The distillation of WEAVER into a lightweight 400M cross-encoder retains ~98.7% accuracy while reducing inference compute by 99.97%, enabling real-world applicability.

* Applies latent variable modeling to verifier combination in a new setting, overcoming output inconsistencies and low-quality verifiers.

Weakness:
* Performance depends on having access to a broad and complementary pool of weak verifiers.

* WEAVER is most useful in repeated sampling scenarios; its gains are less relevant where only a single response can be generated.

---

> ### Author Rebuttal · Authors · 2025-07-31
>
> Thank you for your thoughtful review and constructive feedback. We appreciate your recognition of Weaver's significant performance improvements and your characterization of  the approach as practical and scalable.
>
> We address your two concerns regarding limitations related to verifier diversity and applicability below, by including an extended discussion on the growing role of repeated sampling in real-world applications, and a new set of experiments that evaluate Weaver’s robustness under reduced verifier diversity and its effectiveness in low-sample regimes. We plan to incorporate these additions into the final version of the paper and hope you will consider updating your score in light of these improvements.
>
> **W1: "Performance depends on having access to a broad and complementary pool of weak verifiers."**
>
> Weaver offers performance improvements over naive ensembling even when the pool of verifiers is not broad or complementary. Below, we provide **1)** results from the paper using smaller sets of verifiers **2)** new experiments using a set of more correlated verifiers from the same model family, and **3)** context on why broad, diverse verifier pools are becoming increasingly accessible in practice.
>
> - **(1) Weaver performs well even with few verifiers.** Figure 4 shows how Weaver performs as we vary the number of verifiers from 1 to 15, a significantly smaller set than the 33 used in our main experiments. Even with just the top-3 verifiers, Weaver outperforms naive ensembling on GPQA, MATH500, and MMLU Pro. This demonstrates that a broad pool of verifiers is not necessary for Weaver to be effective.
>
> - **(2) Weaver is robust to correlated verifiers.** To test Weaver’s performance on a non-complementary set of verifiers, we constructed a pool of 13 verifiers all derived from the Llama family (e.g., Llama 3.1 8B/70B variants), fine-tuned with similar RLHF data and objectives. These verifiers are highly correlated, with an average pairwise Pearson correlation of 0.491. As shown in Table 1 below, Weaver achieves a +4.3% average improvement over naive ensembling—only modestly below the +6.8% improvement we observe with the full, more diverse pool (Table 2; average correlation 0.281). These results indicate that while diversity enhances performance, Weaver remains effective even when verifiers are highly redundant. We will include the two tables below in the updated draft.
>
> - **(3) Having access to a broad, diverse set of verifiers is increasingly common in practice.** While we have shown above that Weaver does not require such verifier pools, the ecosystem of pre-trained models and tools that can be used for verification continues to grow in scale and variety. Figure 7 of our paper shows that there are now over 150 reward models and 250 LLM judges currently available. Furthermore, new community leaderboards continue to emerge (e.g., RewardBench V2: https://huggingface.co/spaces/allenai/reward-bench and ChatBotArena: https://huggingface.co/spaces/lmarena-ai/chatbot-arena-leaderboard), offering us access to the newest high-quality reward models and judges.
>
> **Table 1 (New Results: Weaver with Llama Verifiers):**
> | Model | GPQA Diamond | MATH500 | MMLU Pro | Average |
> |:---|:---:|:---:|:---:|:---:|
> | Weaver with Llama Verifier | 59.4% | 91.1% | 87.4% | 79.3% |
> | Naive Ensemble | 52.1% | 88.9% | 84.1% | 75.0% |
> | Weaver with Llama Verifiers - Improvement % | +8.3% | +2.2% | +3.3% | +4.3% |
>
> **Table 2 (Existing Results: Weaver with All Verifiers, Currently Table 1 and Figure 3):**
> | Model | GPQA Diamond | MATH500 | MMLU Pro | Average |
> |:---|:---:|:---:|:---:|:---:|
> | Weaver | 66.4% | 93.4% | 90.2% | 83.3% |
> | Naive Ensemble | 54.5% | 90.6% | 84.3% | 76.5% |
> | Weaver Improvement % | +11.9% | +2.8% | +5.9% | +6.8% |
>
> **W2: "Weaver is most useful in repeated sampling scenarios; its gains are less relevant where only a single response can be generated."**
>
> We agree that Weaver is primarily designed for the repeated sampling setting; we will clarify its scope in the discussion section. However, we note that 1) Weaver still provides advantages in limited sampling regimes, 2) repeated sampling is increasingly common in practice, and 3) repeated sampling is becoming increasingly practical to deploy.
>
> - **(1) Weaver still provides advantages in limited sampling regimes.** Specifically, Weaver improves the selection accuracy even when only a few (e.g., 2–4) responses are available, as shown in Figure 3 of the paper.
> - **(2) Repeated sampling is increasingly common in practice,** driven by its effectiveness and by recent findings that scaling test-time compute (e.g., sampling more outputs) can yield gains comparable to training larger models [1, 2, 3]. This has led to broader adoption of repeated sampling across high-value applications:
>
>    - *Inference-time use cases*:
>        - Mathematical reasoning and theorem proving (e.g., MiniF2F, MATH datasets) [9]
>        - Scientific question answering (e.g., GPQA for expert-level queries) [10]
>        - Business, engineering, and humanities tasks requiring multi-hop reasoning (e.g., MMLU-Pro) [11]
>        - Code generation and debugging (e.g., LiveCodeBench) [12]
>        - Creative writing or content generation where diversity improves quality (e.g. ArenaHardAuto) [5]
>        - Deployed assistants and production systems (e.g., GPT-4o, Claude, Gemini), which internally sample and verify multiple responses before emitting a final output to improve safety and reliability. [13]
>    - *Training-time workflows*:
>        - Reinforcement learning (GRPO / DPO) algorithms requiring 16+ generations per query with associated scores [6, 7]
>        - Data filtering and synthetic data creation for model training [4]
>
> - **(3) Repeated sampling is becoming increasingly practical to deploy.** Several trends support the growing viability of repeated sampling in real-world systems. Recent ML systems work has made repeated sampling more compute and memory efficient. Inference engines like Tokasaurus, vLLM, and SGLang [8, 14,15] achieve 2-6x throughput gains through dynamic prefix sharing, optimized memory management with PagedAttention, and structured execution with RadixAttention. FlexGen [16] and Sarathi-Serve [17] enable high-throughput inference on constrained hardware, though they may require tailored programming or larger batch sizes to maximize efficiency. These advancements allow real-world systems to handle high-throughput language model calls with greater efficiency, making repeated sampling increasingly practical for deployment.
>
>
> [1] Schaeffer, Rylan, et al. "How Do Large Language Monkeys Get Their Power (Laws)?." arXiv preprint arXiv:2502.17578 (2025).
>
> [2] Brown, Bradley, et al. "Large language monkeys: Scaling inference compute with repeated sampling." arXiv preprint arXiv:2407.21787 (2024).
>
> [3] Snell, Charlie, et al. "Scaling llm test-time compute optimally can be more effective than scaling model parameters." arXiv preprint arXiv:2408.03314 (2024).
>
> [4] Guha, Etash, et al. "OpenThoughts: Data Recipes for Reasoning Models." arXiv preprint arXiv:2506.04178 (2025)
>
> [5] Li, Tianle, et al. "From crowdsourced data to high-quality benchmarks: Arena-hard and benchbuilder pipeline." arXiv preprint arXiv:2406.11939 (2024).
>
> [6] Rafailov, Rafael, et al. "Direct preference optimization: Your language model is secretly a reward model." Advances in neural information processing systems 36 (2023): 53728-53741.
>
> [7] Shao, Zhihong, et al. "Deepseekmath: Pushing the limits of mathematical reasoning in open language models." arXiv preprint arXiv:2402.03300 (2024).
>
> [8] Juravsky, Jordan, et al. "Tokasaurus: An LLM Inference Engine for High-Throughput Workloads." Scaling Intelligence, 2025, scalingintelligence.stanford.edu/blogs/tokasaurus/.
>
> [9] Zheng, Kunhao, Jesse Michael Han, and Stanislas Polu. "Minif2f: a cross-system benchmark for formal olympiad-level mathematics." arXiv preprint arXiv:2109.00110 (2021).
>
> [10] Rein, David, et al. "Gpqa: A graduate-level google-proof q&a benchmark." First Conference on Language Modeling. 2024.
>
> [11] Wang, Yubo, et al. "Mmlu-pro: A more robust and challenging multi-task language understanding benchmark." Advances in Neural Information Processing Systems 37 (2024): 95266-95290.
>
> [12] Jain, Naman, et al. "Livecodebench: Holistic and contamination free evaluation of large language models for code." arXiv preprint arXiv:2403.07974 (2024).
>
> [13] Team, Gemini, et al. "Gemini: a family of highly capable multimodal models." arXiv preprint arXiv:2312.11805 (2023).
>
> [14] Kwon, Woosuk, et al. "Efficient Memory Management for Large Language Model Serving with PagedAttention." arXiv preprint arXiv:2309.06180 (2023).
>
> [15] Zheng, Lianmin, et al. "SGLang: Efficient Execution of Structured Language Model Programs." arXiv preprint arXiv:2312.07104 (2023).
>
> [16] Sheng, Ying, et al. "FlexGen: High-Throughput Generative Inference of Large Language Models with a Single GPU." arXiv preprint arXiv:2303.06865 (2023).
>
> [17] Agrawal, Amey, et al. "Taming Throughput-Latency Tradeoff in LLM Inference with Sarathi-Serve." arXiv preprint arXiv:2403.02310 (2024).

---

> ### Author Response · Authors · 2025-08-04
>
> We appreciate your time and consideration! Should any further questions or thoughts arise, we would be happy to engage in additional discussion during the rebuttal stage!

---

> > ### Comment · Area_Chair_c6JT · 2025-08-06
> > **Gentle ping**
> >
> > Dear reviewer XeAb. Could you please engage in a discussion with the authors.

---

### Note · Authors · 2025-08-12

We thank the reviewers for their engagement and feedback.

**Response to xhJS**

It is reasonable for Weaver to learn on the same queries it is evaluated on when the practitioner’s goal is to label and select the best candidate response per query on **their unlabeled dataset**. In this setting, the practitioner labels a small dev subset for calibration, and then Weaver learns verifier accuracies on their entire dataset (not using labels). Because the aim is to **label the dataset itself** for selecting responses, evaluating on it is justified. This is standard in transductive settings such as semi-supervised learning.

In Section 4.1, the 100% labels results provide an upper bound describing the expressivity of different aggregation methods, motivating the need for weighted versions.

Per your suggestion, we used Weaver to estimate verifier accuracies from MMLU Pro and then used these on GPQA and MATH500. The performance drop was modest (-6.6%, -2.9% respectively) suggesting that verifier accuracies can transfer.

**Response to cAMM**

For the 1% dev set, we calibrate on 5 labeled queries, use all 500 unlabeled for moment-matching, and evaluate on 495 non-dev queries. The 100% dev set is the only case where all 500 queries are used for calibration, moment-matching, and evaluation. We apologize for our earlier misstatement on using 495 queries for moment-matching.

We have corrected L197's wording about computing $Pr(S_1,...,S_m)$ directly from data.

**Contributions**

Weaver is a framework for combining weak verifiers that outperforms unsupervised approaches and is competitive with supervised ones:

- **Label-efficient**: Uses unlabeled data and requires only ~1% labeled examples for calibration
- **Frontier performance**: Achieves o3-mini level accuracy using Llama 3.3 70B generations
- **Compute-efficient**: Distilled 400M cross-encoder retains 98.7% accuracy while reducing verification compute by 99.97%
- **Principled aggregation**: Latent variable modeling outperforms naive averaging by 6.8% and handles correlated/noisy verifiers

Thanks to reviewers, we added experiments:
- Weaver with correlated verifiers (Llama family)
- Supervised baselines: XGBoost, AdaBoost, and L1-regularized LR
- Continuous vs binary versions of Weaver
- RewardBench V2 Evaluation
- OOD evaluation

Weaver is particularly valuable for expert domains (e.g. GPQA) and emerging tasks where high-quality labels are scarce/costly, offering a practical path to improve LLM verification.

---

### Decision · Program_Chairs · 2025-09-17

**Decision:**

Accept (poster)

**Comment:**

This paper introduces Weaver, a verification framework of language model outputs by aggregating signals from multiple "weak" verifiers, such as language model judges and reward models. The central claim is that by combining these imperfect verifiers in a principled, data-efficient manner, one can create a strong, unified verifier that significantly improves the selection of correct responses from a pool of candidates generated at test time.

Based on my reading and the reviewers' assessments, the key findings are:
1.  **Superior Aggregation:** The paper demonstrates that a weighted aggregation of verifiers substantially outperforms naïve methods like simple averaging or majority vote. Weaver adapts weak supervision techniques, using a small amount of labeled data for calibration and a large amount of unlabeled data to learn the accuracy of each verifier.
3.  **Compute Efficiency via Distillation:** Recognizing that querying dozens of verifiers can be computationally expensive, the authors show that the aggregated signal from the Weaver ensemble can be distilled into a compact 400M parameter cross-encoder model. This distilled verifier retains 98.7% of the full ensemble's performance while reducing the verification compute cost by up to 99.97%, making the approach highly practical.
2.  **Performance:** By using Weaver to select the best response from 100 samples generated by Llama 3.3 70B, the system achieves performance on par with or exceeding that of much larger, models.


### (b) Strengths of the Paper

*   **Significance and Novelty:** The paper addresses the critical and highly relevant problem of improving  verification . The proposed method of using weak supervision for label-efficient verifier ensembling is novel and practical.
*   **Strong Empirical Results:** The experimental results are impressive and demonstrate significant gains over strong baselines across multiple challenging benchmarks . The ability to elevate the performance of a 70B model to the level of frontier models is a compelling demonstration of the framework's power.
*   **Practicality and Scalability:** The distillation of the large verifier ensemble into a small, efficient cross-encoder is a crucial contribution that makes the system viable for real-world deployment. The label-efficient nature of Weaver, requiring only a small calibration set, further enhances its practicality, especially for domains where labeled data is scarce or expensive.
*   **Thorough Analysis:** The authors provide a comprehensive analysis of Weaver's performance, exploring its scaling properties with respect to the number of samples, verifiers, and model size.

### (c) Weaknesses of the Paper

The reviewers raised several valid concerns, many of which were addressed during the rebuttal period:
*   **Dependence on Verifier Diversity:** A key concern was how Weaver's performance would be affected by a lack of diversity or high correlation among the weak verifiers, which is a realistic scenario (Reviewer XeAb, Reviewer jVww, Reviewer cAMM).
*   **Experimental Setup:** There were initial questions about the fairness of the evaluation setup, particularly regarding the use of a development set drawn from the same benchmarks used for testing. This raised concerns about potential test set leakage (Reviewer jVww, Reviewer cAMM). Reviewers also requested comparisons against more sophisticated baselines, such as regularized logistic regression and tree-based models (Reviewer cAMM).
*   **Generalizability:** The evaluation was primarily focused on mathematical and logical reasoning tasks using a single family of generator models (Llama). Reviewers questioned the framework's applicability to other domains like code generation or subjective reasoning (Reviewer jVww).
*   **Clarity and Missing Details:** Some technical details were initially unclear, such as the motivation for binarizing verifier scores and the precise mechanism for estimating certain probabilities in the model (Reviewer cAMM).

### (d) Recommendation: Accept

I strongly recommend accepting this paper. The work is technically solid, addresses a problem of high importance, and presents a novel, practical, and effective solution. The empirical results are compelling and demonstrate a significant step forward in making language models more reliable through enhanced verification.